# Breaking Correlation Shift via Conditional Invariant Regularizer

**Mingyang Yi**[1,2,3], **Ruoyu Wang**[1,2], **Jiacheng Sun**[3], **Zhenguo Li**[3], **Zhi-Ming Ma**[1,2]
[1]University of Chinese Academy of Sciences
{yimingyang17,wangruoyu17}@mails.ucas.edu.cn
[2]Academy of Mathematics and Systems Science, Chinese Academy of Sciences
mazm@amt.ac.cn
[3]Huawei Noah's Ark Lab
{sunjiacheng1,li.zhenguo}@huawei.com

## Abstract

Recently, generalization on out-of-distribution (OOD) data with correlation shift has attracted great attentions. The correlation shift is caused by the spurious attributes that correlate to the class label, as the correlation between them may vary in training and test data. For such a problem, we show that given the class label, the models that are conditionally independent of spurious attributes are OOD generalizable. Based on this, a metric Conditional Spurious Variation (CSV) which controls the OOD generalization error, is proposed to measure such conditional independence. To improve the OOD generalization, we regularize the training process with the proposed CSV. Under mild assumptions, our training objective can be formulated as a nonconvex-concave mini-max problem. An algorithm with a provable convergence rate is proposed to solve the problem. Extensive empirical results verify our algorithm's efficacy in improving OOD generalization.

## 1 Introduction

The success of standard learning algorithms rely heavily on the identically distributed assumption of training and test data. However, in real-world, such assumption is often violated due to the varying circumstances, selection bias, and other reasons (Meinshausen & Bühlmann, 2015). Thus, learning a model that generalizes on out-of-distribution (OOD) data has attracted great attentions. The OOD data (Ye et al., 2022) can be categorized into data with *diversity shift* or *correlation shift*. Roughly speaking, there is a mismatch of the spectrum and a spurious correlation between training and test distributions under the two shifts, respectively. Compared with diversity shift, correlation shift is less explored (Ye et al., 2022), while the misleading spurious correlation works for training data may deteriorate model's performance on test data (Beery et al., 2018).

The correlation shift says, for the spurious attributes in data, there exists variation of (spurious) correlation between class label and such spurious attributes from training to test data (Figure 1). Based on a theoretical characterization of it, we show that given the class label, the model which is conditionally independent of spurious attributes has stable performance across training and OOD test data. Then, a metric *Conditional Spurious Variation* (CSV, Definition 2) is proposed to measure such conditional independence. Notably, in contrast to the existing metrics related to OOD generalization (Hu et al., 2020; Mahajan et al., 2021), our CSV can control the OOD generalization error.

To improve OOD generalization, we regularize the training process with estimated CSV. With observable spurious attributes, we propose an estimator to CSV. However, such observable condition may be violated. In this case, we propose another estimator, which approximates a sharp upper bound of CSV. We regularize the training process with one of them, depending on whether the spurious attributes are observable. Our method improves the observable condition in (Sagawa et al., 2019).

Under mild smoothness assumptions, the regularized training objective can be formulated as a specific non-convex concave minimax problem. A stochastic gradient descent based algorithm with a provable convergence rate of order $\mathcal{O}(T^{-2/5})$ is proposed to solve it, where $T$ is the number of iterations.

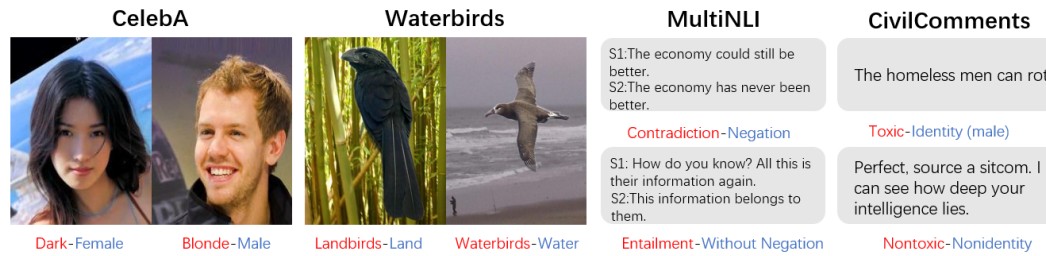

Figure 1: Examples of `CelebA` (Liu et al., 2015), `Waterbirds` (Sagawa et al., 2019), `MultiNLI` (Williams et al., 2018), and `CivilComments` (Borkan et al., 2019) involved in this paper. The class labels and spurious attributes are respectively colored with red and blue. Their correlation may vary from training set to test set. More details are shown in Section 6.

Finally, extensive experiments are conducted to empirically verify the effectiveness of our methods on the OOD data with spurious correlation. Concretely, we conduct experiments on benchmark classification datasets `CelebA` (Liu et al., 2015), `Waterbirds` (Sagawa et al., 2019), `MultiNLI` (Williams et al., 2018), and `CivilComments` (Borkan et al., 2019). Empirical results show that our algorithm consistently improves the model's generalization on OOD data with correlation shifts.

## 2 RELATED WORKS AND PRELIMINARIES

### 2.1 RELATED WORKS

**OOD Generalization.** The appearance of OOD data (Hendrycks & Dietterich, 2018) has been widely observed in machine learning community (Recht et al., 2019; Schneider et al., 2020; Salman et al., 2020; Tu et al., 2020; Lohn, 2020). To tackle this, researchers have proposed various algorithms from different perspectives, e.g., distributional robust optimization (Sinha et al., 2018; Volpi et al., 2018; Sagawa et al., 2019; Yi et al., 2021b; Levy et al., 2020) or causal inference (Arjovsky et al., 2019; He et al., 2021; Liu et al., 2021b; Mahajan et al., 2021; Wang et al., 2022; Ye et al., 2021). Ye et al. (2022) points out that the OOD data can be categorized into data with *diversity shift* (e.g., `PACS` (Li et al., 2018)) and *correlation shift* (e.g., `Waterbirds` (Sagawa et al., 2019)), and we focus on the latter in this paper, as we have clarified that it deteriorates the performance of the model on OOD test data (Geirhos et al., 2018; Beery et al., 2018; Xie et al., 2020; Wald et al., 2021).

**Domain Generalization.** To goal of domain generalization is extrapolating model to test data from unseen domains to capture OOD generalization. The problem we explored can be treated by domain generalization methods as data with different spurious attributes can be regarded as from different domains. The core idea in domain generalization is to learn a domain-invariant model. To this end, Arjovsky et al. (2019); Hu et al. (2020); Li et al. (2018); Mahajan et al. (2021); Heinze-Deml & Meinshausen (2021); Krueger et al. (2021); Wald et al. (2021); Seo et al. (2022) propose plenty of invariant metrics as training regularizer. However, unlike our CSV, none of these metrics controls the OOD generalization error. Moreover, none of these methods capture the invariance corresponds to the correlation shift we discussed (see Section 4.1). This motivates us to reconsider the effectiveness of these methods. Finally, in contrast to ours, these methods require observable domain labels, and it is usually impractical. The techniques in (Liu et al., 2021b; Devansh Arpit, 2019; Sohoni et al., 2020; Creager et al., 2021) are also applicable without domain information, but they are built on strong assumptions (mixture Gaussian data (Liu et al., 2021b) and linear model (Devansh Arpit, 2019)) or require a high-quality spurious attribute classifier (Sohoni et al., 2020; Creager et al., 2021).

**Distributional Robustness.** The distributional robustness (Ben-Tal et al., 2013) based methods minimize the worst-case loss over different groups of data (Sagawa et al., 2019; Liu et al., 2021a; Zhou et al., 2022). The groups are decided via certain rules, e.g., data with same spurious attributes (Sagawa et al., 2019) or annotated via validation sets with observable spurious attributes (Liu et al., 2021a; Zhou et al., 2022). However, Sagawa et al. (2019) finds that directly minimizing the worst-group loss results in unstable training processes. In contrast, our method has stable training process as it balances the objectives of accuracy and robustness over spurious attributes (see Section 5).

### 2.2 PROBLEM SETUP

We collect the notations in this paper. $\|\cdot\|$ is the $\ell_2$-norm of vectors. $\mathcal{O}(\cdot)$ is the order of a number. The sample $(X, Y) \in \mathcal{X} \times \mathcal{Y}$, where $X$ and $Y$ are respectively input data and its label. The integer set

from 1 to $K$ is $[K]$. The cardinal of a set $A$ is $|A|$. The loss function is $\mathcal{L}(\cdot, \cdot) : \mathbb{R}^K \times \mathcal{Y} \to \mathbb{R}^+$ with $0 \leq \mathcal{L}(\cdot, \cdot) \leq M$ for positive $K, M$. For any distribution $P$, let $R_{pop}(P, f) = \mathbb{E}_P[\mathcal{L}(f(X), Y)]$ and $R_{emp}(P, f) = n^{-1} \sum_{i=1}^n \mathcal{L}(f(\boldsymbol{x}_i), y_i)$ respectively be the population risk under $P$ and its empirical counterpart. Here $\{(\boldsymbol{x}_i, y_i)\}_{i=1}^n$ are $n$ i.i.d. samples from distribution $P$, and $f(\cdot) : \mathcal{X} \to \mathbb{R}^K$ is the model potentially with parameter space $\Theta \subset \mathbb{R}^d$ ($f(\cdot)$ becomes $f_{\boldsymbol{\theta}}(\cdot)$). For random variables $V_1, V_2$ with joint distribution $P_{V_1, V_2}$, $P_{V_1}$ and $P_{V_2|V_1}$ are the marginal distribution of $V_1$ and the conditional distribution of $V_2$ under $V_1$. $P_{V_1}(v_1)$ and $P_{V_2|V_1}(v_1, v_2)$ are their probability measures.

Suppose the training and OOD test data are respectively from distributions $P_{X,Y,Z}$ and $Q_{X,Y,Z}$. We may neglect the subscript if there is no obfuscation. There usually exists similarities between $P$ and $Q$ that guarantee the possibility of OOD generalization (Kpotufe & Martinet, 2021). The similarity we explored is that there only exists a correlation shift in the OOD test data formulated as follows. For each input $X$, there exists spurious attributes $Z \in \mathcal{Z}$ that are not causal to predict class label, but $Z$ is potentially related to class label $Y$. The $Z$ can be some features of $X$ e.g., gender of celebrity in Figure 1. The correlation between $Z$ and $Y$ (i.e., spurious correlation) can vary from training to test data, and the one in training distribution may become a misleading signal for the model to make predictions on test data. For example, in the celebrity's face in Figure 1, if most males in the training set have dark hair, the model may overfit such spurious correlation and mispredict the male with blond hair. Thus we should learn a model that is robust to correlation shift defined as follows.

**Definition 1.** *Given training distribution $P$, the test distribution $Q \in \mathcal{P}$ has correlation shift, where*

$$\mathcal{P} = \{Q_{X,Y,Z} : Q_Y = P_Y, Q_{X|Y,Z} = P_{X|Y,Z}\}. \tag{1}$$

Our definition characterizes the distributions with correlation shift. The first equality in (1) obviates the mismatching caused by label shift (i.e., $P_Y \neq Q_Y$) which is unrelated to spurious correlation. More discussions of it are in Appendix A. The second equality in (1) states the invariance of conditional distribution of data, given the class label and spurious attributes, which is reasonable as the unstable spurious correlations are decided by the joint distribution of $Y$ and $Z$. Finally, since

$$
\begin{aligned}
Q_{X,Y}(\boldsymbol{x}, y) &= Q_{X,Y}(\boldsymbol{x} \mid y) Q_Y(y) = Q_Y(y) \int_{\mathcal{Z}} Q_{X|Y,Z}(\boldsymbol{x} \mid y, z) dQ_{Z|Y}(z \mid y) \\
&= P_Y(y) \int_{\mathcal{Z}} P_{X|Y,Z}(\boldsymbol{x} \mid y, z) dQ_{Z|Y}(z \mid y),
\end{aligned}
\tag{2}
$$

the two constraints in (1) together implies the correlation shift of $Q \in \mathcal{P}$ is from the variety of conditional distributions $Q_{Z|Y}$, which is consistent with intuition. Our definition is different from the ones in (Mahajan et al., 2021; Makar & D'Amour, 2022), as they rely on a causal directed acyclic graph and the existence of a sufficient statistic such that $Y$ only affects $X$ through it.

## 3 GENERALIZING ON OOD DATA

In this section, we show that misleading spurious correlation can hurt the OOD generalization. Then we give a condition under which the model is OOD generalizable.

### 3.1 SPURIOUS CORRELATION MISLEADS MODELS

The common way to train a model is empirical risk minimization (ERM Vapnik, 1999), i.e., approximating the minimizer of $R_{pop}(P, f)$ which generalizes well on in-distribution samples via minimizing its empirical counterpart $R_{emp}(P, f)$. However, the following proposition shows that the minimizer of $R_{pop}(P, f)$ may not generalize well on the OOD data from the other distributions in $\mathcal{P}$.

**Proposition 1.** *There exists a population risk $R_{pop}(P, f)$ whose minimizer has nearly perfect performance on the data from $P$, while it fails to generalize to OOD data drawn from another $Q \in \mathcal{P}$.*

Similar results also appear in (Xie et al., 2020; Krueger et al., 2021), while they are not obtained on the minimizer of population risk. The proof of this proposition is in Appendix B which indicates that the spurious correlation in training data can become a misleading supervision signal that deteriorates the model's performance on OOD data. Hence, it is crucial to learn a model that is independent of such spurious correlation, even if it sometimes can be helpful in the training set (Xie et al., 2020).

## 3.2 CONDITIONAL INDEPENDENCE ENABLES OOD GENERALIZATION

Next, we give a sufficient condition proved in Appendix B to make the model OOD generalizable. The condition is also necessary under some specific data generating structures (Veitch et al., 2021).

**Theorem 1.** *For model $f(\cdot)$ satisfying $f(X) \perp Z \,|\, Y$, the conditional distribution $Y \mid f(X)$ and population risk $\mathbb{E}_Q[\mathcal{L}(f(X), Y)]$ are invariant with $(X, Y) \sim Q_{X,Y}$ such that $Q \in \mathcal{P}$.*

Here $f(X) \perp Z \,|\, Y$ means given $Y$, $f(X)$ is conditionally independent of $Z$. Theorem 1 shows our conditional independence obviates the impact of correlation shift, as the prediction error (gap between $Y$ and $f(X)$, decided by $Y \mid f(X)$) and population risk of model are invariant over test distributions $Q \in \mathcal{P}$. Thus we propose to obtain a model that is conditional independent of spurious attributes.

**Remark 1.** *If the spurious attributes are domain labels, the conditional independence in Theorem 1 becomes the ones in (Liu et al., 2015; Hu et al., 2020; Mahajan et al., 2021), while they do not explore its correlation with the OOD generalization. Besides, the counterexample in Mahajan et al. (2021) violates our conditional invariant assumption in (1) and hence is not contrary to our theorem.*

## 4 LEARNING OOD GENERALIZABLE MODEL

In Theorem 1 we propose a independence condition to break correlation shift. In this section, a metric Conditional Spurious Variation (CSV) is proposed to quantitatively measure the independence. As our CSV can control the OOD generalization error, smaller CSV leads to improved OOD generalization. Finally, two estimators of CSV are proposed, depending on whether spurious attributes are observable.

### 4.1 GUARANTEED OOD GENERALIZATION ERROR

Theorem 1 shows that the conditional independence between the model and spurious attributes guarantees the OOD generalization. We propose the following metric to measure such independence.

**Definition 2** (Conditional Spurious Variation). *The conditional spurious variation of model $f(\cdot)$ is*

$$\text{CSV}(f) = \mathbb{E}_Y \left[ \sup_{z_1, z_2} \left( \mathbb{E}_X[\mathcal{L}(f(X), Y) \mid Y, Z = z_1] - \mathbb{E}_X[\mathcal{L}(f(X), Y) \mid Y, Z = z_2] \right) \right]. \quad (3)$$

As can be seen, CSV is a functional of $f(\cdot)$ which measures the intra-class conditional variation of the model over spurious attributes, given the class label $Y$. It can be computed via training distribution and is invariant across $Q \in \mathcal{P}$ due to (1). It is worth noting that the model satisfies the conditional independence in Theorem 1 has zero CSV but not vice versa. [1] However, the following theorem proved in Appendix C.1 shows that $\text{CSV}(f)$ is sufficient to control the OOD generalization error.

**Theorem 2.** *For any $Q \in \mathcal{P}$, we have*

$$\sup_{Q \in \mathcal{P}} |R_{emp}(f, P) - R_{pop}(f, Q)| \leq |R_{emp}(f, P) - R_{pop}(f, P)| + \text{CSV}(f) \quad (4)$$

The $|R_{emp}(f,P) - R_{pop}(f,P)|$ is in-distribution generalization error, which is well explored (Vershynin, 2018). Thus, we upper bound the OOD generalization error via the in-distributional one and CSV. The OOD generalization error is also connected to many other metrics e.g., (Hu et al., 2020; Mahajan et al., 2021; Ben-David et al., 2007; 2010; Muandet et al., 2013; Ganin et al., 2016), but none of them directly control the OOD generalization error. Besides, these metrics are proposed to obtain the invariance over $Z$ as a condition, i.e., invariant $P_{f(X),Y|Z}$ or $P_{f(X)|Z}$, while the invariances can not handle correlation shift (Definition. 1). As, 1): invariant $P_{f(X),Y|Z}$ implies invariant $P_{Y|Z}$ which is incompatible with correlation shift, 2): invariant $P_{f(X)|Z} = \int_{\mathcal{Y}} P_{f(X)|Z,Y}(f(x) \mid z, y) dP_{Y|Z}(y \mid z)$ does not imply invariant $P_{f(X)|Y,Z}$ which guarantees OOD generalization.

As our bound (4) involves both CSV and in-distribution generalization error, it motivates us to explore whether the conditional independence is contradicted by the in-distribution generalization.

---

[1] Conditional independence is a strong sufficient condition to make model OOD generalizable. However, the proof of Theorem 1 shows the model that is invariant with spurious correlation $Q(Z \mid Y)$ is sufficient to be OOD generalizable, while the invariance can be characterized by both zero CSV and conditional independence.

The following information-theoretic bound proved in Appendix C.1 presents a positive answer. Let $I(V_1, V_2)$ be the mutual information between variables $V_1, V_2$, we have the following result.

**Theorem 3.** *Let model $f_{\boldsymbol{\theta}}(\cdot)$ parameterized by $\boldsymbol{\theta} \in \Theta \subset \mathbb{R}^d$, and is trained on $\mathcal{S} = \{(\boldsymbol{x}_i, y_i)\}_{i=1}^n$ from distribution $P$, with the spurious attributes of $\boldsymbol{x}_i$ is $z_i$. If the learned model $f_{\boldsymbol{\theta}_{\mathcal{S}}}(\cdot) \perp \mathcal{S}_{\boldsymbol{z}} \mid \mathcal{S}_y$* [2]

$$\mathcal{E}_{gen}(f_{\boldsymbol{\theta}_{\mathcal{S}}}, P) \leq \inf_g \sqrt{\frac{M^2}{4n} \left( I(\mathcal{S}_{\boldsymbol{x}-g(z)}, \mathcal{S}_y; f_{\boldsymbol{\theta}_{\mathcal{S}}} \mid \mathcal{S}_y, \mathcal{S}_{g(z)}) + I(\mathcal{S}_y; f_{\boldsymbol{\theta}_{\mathcal{S}}}) \right)}, \tag{5}$$

*where $\mathcal{E}_{gen}(f_{\boldsymbol{\theta}_{\mathcal{S}}}, P) = |\mathbb{E}[R_{emp}(f_{\boldsymbol{\theta}_{\mathcal{S}}}, P)] - R_{pop}(f_{\boldsymbol{\theta}_{\mathcal{S}}}, P)|$, $g(\cdot)$ is any measurable function, $\mathcal{S}_{\boldsymbol{x}-g(z)} = \{\boldsymbol{x}_i - g(z_i)\}_{i=1}^n$, $\mathcal{S}_y = \{y_i\}_{i=1}^n$.*

Our bound improves the classical result $\mathcal{E}_{gen}(f_{\boldsymbol{\theta}_{\mathcal{S}}}, P) \leq \sqrt{M^2 I(\mathcal{S}, \boldsymbol{\theta}_{\mathcal{S}})/4n}$ without conditional independence (Steinke & Zakynthinou, 2020), because taking $g(\cdot)$ as a constant function, then applying data processing inequality (Xu & Raginsky, 2017) in r.h.s. of (5), it becomes $\sqrt{M^2 I(\mathcal{S}, \boldsymbol{\theta}_{\mathcal{S}})/4n}$. The bound indicates the model $f_{\boldsymbol{\theta}_{\mathcal{S}}}(\cdot)$ that is conditional independent of spurious attribute (we aim to capture) is not in contradiction with in-distribution generalization. Thus, due to Theorem 2, less conditional independence with spurious attributes of model improves the OOD generalization bound.

## 4.2 ESTIMATING CSV WITH OBSERVABLE SPURIOUS ATTRIBUTES

As smaller CSV enables improved OOD generalization, we propose to regularize the training process with it. Suppose we have $n$ i.i.d. $\{(\boldsymbol{x}_i, y_i)\}_{i=1}^n$ training samples with spurious attributes $\{z_i\}_{i=1}^n$ from $P$. Before our analysis, we need the following two mild assumptions in the sequel.

**Assumption 1.** *The class label and spurious attributes are discrete, i.e., the $\mathcal{Y} = [K_y]$ and $\mathcal{Z} = [K_z]$ for some positive integers $K_y, K_z$. Besides that, the number of observations $A_{kz} = \{i : y_i = k, z_i = z\}$ from each pair of $(k, j) \in [K_y] \times [K_z]$ is $|A_{kz}| = n_{kz} > 0$.*

**Assumption 2.** *The model is parameterized by $\boldsymbol{\theta} \in \Theta \subset \mathbb{R}^d$. The loss function $\mathcal{L}(f_{\boldsymbol{\theta}}(\boldsymbol{x}), y)$ is Lipschitz continuous and smooth w.r.t. $\boldsymbol{\theta}$ with coefficient $L_0$ and $L_1$, i.e., for any $(\boldsymbol{x}, y) \in \mathcal{X} \times \mathcal{Y}$, and $\boldsymbol{\theta}_1, \boldsymbol{\theta}_2 \in \Theta$,*

$$\begin{aligned} |\mathcal{L}(f_{\boldsymbol{\theta}_1}(\boldsymbol{x}), y) - \mathcal{L}(f_{\boldsymbol{\theta}_2}(\boldsymbol{x}), y)| &\leq L_0 \|\boldsymbol{\theta}_1 - \boldsymbol{\theta}_2\|; \\ \|\nabla_{\boldsymbol{\theta}} \mathcal{L}(f_{\boldsymbol{\theta}_1}(\boldsymbol{x}), y) - \nabla_{\boldsymbol{\theta}} \mathcal{L}(f_{\boldsymbol{\theta}_2}(\boldsymbol{x}), y)\| &\leq L_1 \|\boldsymbol{\theta}_1 - \boldsymbol{\theta}_2\|. \end{aligned} \tag{6}$$

In Assumption 1, we require the spurious attributes space is finite. This is explained as $Z$ is a "label" of spurious attributes, e.g., the gender label "male" or "female" in `CelebA` dataset (Figure 1) when classifying hair color. Assumption 1 also requires the data with all the possible combinations of the label and spurious attributes are collected in the training set. This is a mild condition since we do not restrict the magnitude of $n_{kz}$. For example, to satisfy this, we can synthetic some of the missing data by generative models as in (Wang et al., 2022; Zhu et al., 2017).

Let $\mathcal{L}_{kz}(f_{\boldsymbol{\theta}}) = \mathbb{E}[\mathcal{L}(f_{\boldsymbol{\theta}}(X), k) \mid Y = k, Z = z]$, $\hat{\mathcal{L}}_{kz}(f_{\boldsymbol{\theta}}) = (1/n_{kz}) \sum_{i \in A_{kz}} \mathcal{L}(f_{\boldsymbol{\theta}}(\boldsymbol{x}_i), k)$, and $\hat{p}_k = n_k/n$ with $n_k = \sum_{z \in [K_z]} n_{kz}$. Then the following empirical counterpart of CSV

$$\widehat{\mathrm{CSV}}(f_{\boldsymbol{\theta}}) = \sum_{k=1}^{K_y} \sup_{z_1, z_2 \in [K_z]} \left( \hat{\mathcal{L}}_{kz_1}(f_{\boldsymbol{\theta}}) - \hat{\mathcal{L}}_{kz_2}(f_{\boldsymbol{\theta}}) \right) \hat{p}_k \tag{7}$$

is a natural estimator to CSV. The following theorem quantify its approximation error.

**Theorem 4.** *Under Assumption 1 and 2, if $\inf_{k \in [K_y], z \in [K_z]} n_{kz}/n_k = \mathcal{O}(1)$, then*

$$\mathrm{CSV}(f_{\boldsymbol{\theta}}) \leq \widehat{\mathrm{CSV}}(f_{\boldsymbol{\theta}}) + \mathcal{O}\left( \frac{\log(1/\delta)}{\sqrt{n}} \right) \tag{8}$$

*holds with probability at least $1 - \delta$ for any $\boldsymbol{\theta} \in \Theta, \delta > 0$.*

This theorem implies that CSV is upper bounded by $\widehat{\mathrm{CSV}}(f_{\boldsymbol{\theta}})$. As shown in the proof in Appendix C.2, we hide a factor related to covering number (Vershynin, 2018) of the hypothesis space in the numerator of $\mathcal{O}(\log(1/\delta)/\sqrt{n})$ in (8). The hidden factor is of order $\sqrt{d}$ (Wainwright, 2019). Thus, more samples are required to estimate CSV in high-dimensional space. Besides that, if the condition $\inf_{k,z} n_{kz}/n_k = \mathcal{O}(1)$ does not hold, the order of error is $\mathcal{O}(1/\sqrt{\min_{k,z} n_{kz}})$ (see Appendix C.2 for details).

---

[2] $\boldsymbol{\theta}_{\mathcal{S}}$ is the learned parameters depends on training set $\mathcal{S}$. $f_{\boldsymbol{\theta}_{\mathcal{S}}}(\cdot)$ is a random element that takes values in a functional space (i.e., model space), details can be referred to (Shiryaev, 2016).

---

**Algorithm 1** Regularize training with CSV.

---

**Input:** Training set $\{(\boldsymbol{x}_i, y_i)\}_{i=1}^n$, number of labels $K_y$ and spurious attributes $K_z$, training steps $T$, model $f_{\boldsymbol{\theta}}(\cdot)$ parameterized by $\boldsymbol{\theta}$. Initialized $\boldsymbol{\theta}_0, \{\boldsymbol{F}_0^k\}$. Positive regularization constant $\lambda$, surrogate constant $\rho$, and correction constant $\gamma$. Estimators $\hat{R}_{emp}(f_{\boldsymbol{\theta}}, P)$ to $R_{emp}(f_{\boldsymbol{\theta}}, P)$, $\hat{\boldsymbol{F}}^k(\boldsymbol{\theta})$ to $\boldsymbol{F}^k(\boldsymbol{\theta})$.

1: **for** $t = 0, \cdots, T$ **do**
2:     **Solve the maximization problem:**
3:     **for** $k = 1, \cdots, K_y$ **do**
4:         $\boldsymbol{F}_{t+1}^k = (1 - \gamma)\boldsymbol{F}_t^k + \gamma\hat{\boldsymbol{F}}^k(\boldsymbol{\theta}(t))$;
5:         $\boldsymbol{u}_k(t+1) = \text{Softmax}(\boldsymbol{F}_{t+1}^k / \rho)$.
6:     **end for**
7:     **Minimization step via SGD:**
8:     $\boldsymbol{\theta}(t+1) = \boldsymbol{\theta}(t) - \eta_{\boldsymbol{\theta}} \sum_{k=1}^{K_y} \hat{p}_k \nabla_{\boldsymbol{\theta}}(\hat{R}_{emp}(f_{\boldsymbol{\theta}(t)}, P) + \lambda \boldsymbol{u}_k(t+1)^\top \boldsymbol{F}_{t+1}^k)$.
9: **end for**

---

### 4.3 ESTIMATING CSV WITH UNOBSERVABLE SPURIOUS ATTRIBUTES

Computing the empirical CSV (7) requires observable spurious attributes which may not be available in practice (Liu et al., 2021a). Thus, we need to estimate CSV in the absence of spurious attributes.

Let $P_{kz} = P_{X|Y=k, Z=z}$ be the conditional distributions of $X$ with $Y, Z$ given. The core difficulty of estimating CSV with unobservable spurious attributes is to estimate $\sup_z \mathbb{E}_{P_{kz}}[\mathcal{L}(f_{\boldsymbol{\theta}}(X), k)] - \inf_z \mathbb{E}_{P_{kz}}[\mathcal{L}(f_{\boldsymbol{\theta}}(X), k)]$ via $n_k$ independent samples $\{(x_i, y_i)\}_{i \in A_k}$ drawn from a mixture distribution $P_k = \sum_{z \in [K_z]} \pi_{kz} P_{kz}$. Here $A_k = \bigcup_{z \in [K_z]} A_{kz}$ for $k \in [K_y]$, $P_k = P_{X|Y=k}$, $\pi_{kz} = P_{Z|Y}(Z = z \mid Y = k)$, and we can not specify the data in $A_k$ is from which of $A_{kz}$. To proceed, suppose $\pi_{kz} \geq c > 0$, which is a necessary condition for Assumption 1 to hold. We show in Appendix C.3 that the quantile conditional expectation

$$\mathbb{E}_{P_k}[\mathcal{L}(f_{\boldsymbol{\theta}}(X), k) \mid \mathcal{L}(f_{\boldsymbol{\theta}}(X), k) \geq q_{P_k}(1 - c)] - \mathbb{E}_{P_k}[\mathcal{L}(f_{\boldsymbol{\theta}}(X), k) \mid \mathcal{L}(f_{\boldsymbol{\theta}}(X), k) \leq q_{P_k}(c)] \quad (9)$$

is an upper bound (which is sharp for $K \geq 3$) of $\sup_z \mathbb{E}_{P_{kz}}[\mathcal{L}(f_{\boldsymbol{\theta}}(X), k)] - \inf_z \mathbb{E}_{P_{kz}}[\mathcal{L}(f_{\boldsymbol{\theta}}(X), k)]$. Here $q_{P_k}(\cdot)$ is the quantile function defined as $q_{P_k}(s) = \inf\{p : P_k(\mathcal{L}(f_{\boldsymbol{\theta}}(X), k) \leq p) \geq s\}$. For large $n_k$, we must have $\pi_{kz} \geq 1/n_k = c$ for each $z \in [K_z]$. Thus by substituting the expectation on $P_k$ in (9) with its empirical counterpart for $c = 1/n_k$, we get the the following estimator

$$\widehat{\text{CSV}}_{\text{U}}(f_{\boldsymbol{\theta}}) = \sum_{k=1}^{K_y} \left( \max_{i \in A_k} \mathcal{L}(f_{\boldsymbol{\theta}}(\boldsymbol{x}_i), k) - \min_{i \in A_k} \mathcal{L}(f_{\boldsymbol{\theta}}(\boldsymbol{x}_i), k) \right) \hat{p}_k. \quad (10)$$

The subscript "U" means "unobservable spurious attributes". Besides that, the $\widehat{\text{CSV}}_{\text{U}}(f_{\boldsymbol{\theta}})$ is an upper bound to the estimator $\widehat{\text{CSV}}(f_{\boldsymbol{\theta}})$, which is another straightfoward way to obtain it.

## 5 REGULARIZING TRAINING WITH CSV (RCSV)

The previous results have claimed that the model with small CSV generalizes well on OOD data. On the other hand, Theorem 4 and discussion in Section 4.3 have approximated the CSV via $\widehat{\text{CSV}}(f_{\boldsymbol{\theta}})$ and $\widehat{\text{CSV}}_{\text{U}}(f_{\boldsymbol{\theta}})$, respectively. Thus we can regularize the training process with one or the other to improve the OOD generalization, depending on whether the spurious attributes are observable.

It is notable that both of the regularized training objectives can be formulated as the following minimax problem for positive constants $m$ and $\lambda$

$$\min_{\boldsymbol{\theta} \in \Theta} \sum_{k=1}^{K_y} \hat{p}_k \left( R_{emp}(f_{\boldsymbol{\theta}}, P) + \lambda \max_{\boldsymbol{u} \in \Delta_m} \boldsymbol{u}^\top \boldsymbol{F}^k(\boldsymbol{\theta}) \right). \quad (11)$$

Here $\Delta_m = \{\boldsymbol{u} = (u_1, \ldots, u_m) \in \mathbb{R}_+^m : \sum_i u_i = 1\}$, $\boldsymbol{F}^k(\boldsymbol{\theta}) \in \mathbb{R}^m$, and each dimension of $\boldsymbol{F}^k(\boldsymbol{\theta})$ is Lipschitz continuous function with Lipschitz gradient. Under Assumption 1 and 2, the training

process of empirical risk minimization regularized with $\widehat{\mathrm{CSV}}(f_{\boldsymbol{\theta}})$ or $\widehat{\mathrm{CSV}}_{\mathrm{U}}(f_{\boldsymbol{\theta}})$ can be formulated as the above problem by respectively setting $\boldsymbol{F}^k(\boldsymbol{\theta})$ as the vectorization of the two following matrices. The $m$ of $\widehat{\mathrm{CSV}}(f_{\boldsymbol{\theta}})$ and $\widehat{\mathrm{CSV}}_{\mathrm{U}}(f_{\boldsymbol{\theta}})$ are respectively $K_z^2$ and $|A_k|^2$.

$$(\hat{\mathcal{L}}_{kz_1}(f_{\boldsymbol{\theta}}) - \hat{\mathcal{L}}_{kz_2}(f_{\boldsymbol{\theta}}))_{z_1,z_2 \in [K_z]}, \quad (\mathcal{L}(f_{\boldsymbol{\theta}}(\boldsymbol{x}_{i_1}), k) - \mathcal{L}(f_{\boldsymbol{\theta}}(\boldsymbol{x}_{i_2}), k))_{i_1, i_2 \in A_k}. \tag{12}$$

Before solving (11), we clarify the difference between regularizing training with CSV and distributional robustness optimization (DRO) based methods which minimize the worst-case expected loss over data with same spurious attributes, e.g., GroupDRO (Sagawa et al., 2019) minimizes $\max_{k,z} \mathbb{E}_{P_{kz}}[\mathcal{L}(f_{\boldsymbol{\theta}}(X), k)]$. Theoretically, the OOD generalizable model has perfect in-distribution test accuracy and robustness over different spurious attributes as in (4). Our regularized training objective split such two goals, while DRO based methods mix them into one objective. Though both objectives theoretically upper bound the loss on OOD data, we empirically observe that the two goals are in contradiction with each other (see Section 6). We also observe that splitting the two goals (our objective) enables us easily take a balance between them, which guarantees a stable training process. In contrast, the mixed training objective can be easily dominated by one of the two goals, which results in an unstable training process. Similar phenomena are also observed in (Sagawa et al., 2019). This motivates the early stopping or large weight decay regularizer used in GroupDRO.

## 5.1 SOLVING THE MINIMAX PROBLEM

Let $\phi^k(\boldsymbol{\theta}, \boldsymbol{u}) = R_{emp}(f_{\boldsymbol{\theta}}, P) + \lambda \boldsymbol{u}^\top \boldsymbol{F}^k(\boldsymbol{\theta})$, $\Phi^k(\boldsymbol{\theta}) = \max_{\boldsymbol{u} \in \Delta_m} \phi^k(\boldsymbol{\theta}, \boldsymbol{u})$. Under Assumption 1 and 2, (11) is a nonconvex-concave minimax problem. As explored in (Lin et al., 2020), the nonconvex-strongly concave minimax problem is much easier than the nonconvex-concave one. Thus we consider the surrogate of $\phi^k(\boldsymbol{\theta}, \boldsymbol{u})$ defined as $\phi_\rho^k(\boldsymbol{\theta}, \boldsymbol{u}) = \phi^k(\boldsymbol{\theta}, \boldsymbol{u}) - \lambda \rho \sum_{j=1}^m \boldsymbol{u}(j) \log(m\boldsymbol{u}(j))$ for $\lambda_\rho > 0$, which is strongly concave w.r.t. $\boldsymbol{u}$, and $\phi^k(\boldsymbol{\theta}, \boldsymbol{u})$ is well approximated by it for small $\rho$. Next, we consider the following nonconvex-strongly concave problem

$$\min_{\boldsymbol{\theta} \in \Theta} \sum_{k=1}^{K_y} \hat{p}_k \max_{\boldsymbol{u} \in \Delta_m} \phi_\rho^k(\boldsymbol{\theta}, \boldsymbol{u}) = \min_{\boldsymbol{\theta} \in \Theta} \sum_{k=1}^{K_y} \hat{p}_k \Phi_\rho^k(\boldsymbol{\theta}), \tag{13}$$

instead of (11), where $\Phi_\rho^k(\boldsymbol{\theta}) = \max_{\boldsymbol{u} \in \Delta_m} \phi_\rho^k(\boldsymbol{\theta}, \boldsymbol{u})$. To solve (13), we propose the Algorithm 1.

In Algorithm 1, lines 3-6 solve the maximization problem in (13), which has close-formed solution $\boldsymbol{u}_k^*(t+1) = \mathrm{Softmax}(\boldsymbol{F}^k(\boldsymbol{\theta}(t))/\rho)$ (Yi et al., 2021a), where $\mathrm{Softmax}(\cdot)$ is the softmax function (Epasto et al., 2020). As the estimator $\hat{\boldsymbol{F}}^k(\boldsymbol{\theta})$ may have large variance, substituting the $\boldsymbol{F}^k(\boldsymbol{\theta})$ in $\boldsymbol{u}_k^*(t+1)$ with it in Line 8 (the minimization step) will induce a large deviation. Thus we use the moving average correction $\boldsymbol{F}_{t+1}^k$ (Line 4) to estimate $\boldsymbol{F}^k(\boldsymbol{\theta}(t))$, which guarantees our convergence result in Theorem 5. The convergence rate of Algorithm 1 is evaluated via approximating first-order stationary point, which is standard in non-convex problems (Ghadimi & Lan, 2013; Lin et al., 2020).

**Theorem 5.** *Under Assumption 1 and 2, if $\hat{R}_{emp}(f_{\boldsymbol{\theta}}, P)$ and $\hat{\boldsymbol{F}}^k(\boldsymbol{\theta})$ are all unbiased estimators with bounded variance, $\boldsymbol{\theta}(t)$ is updated by Algorithm 1 with $\eta_{\boldsymbol{\theta}} = \mathcal{O}\left(T^{-\frac{3}{5}}\right)$ and $\gamma = T^{-\frac{2}{5}}$, then*

$$\min_{1 \le t \le T} \mathbb{E}\left[\left\|\sum_{k=1}^{K_y} \hat{p}_k \nabla \Phi_\rho^k(\boldsymbol{\theta}(t))\right\|^2\right] \le \mathcal{O}\left(T^{-\frac{2}{5}}\right). \tag{14}$$

*Besides that, for any $\boldsymbol{\theta}(t)$ and $\rho$, we have $|\sum_{k=1}^{K_y} \hat{p}_k(\Phi_\rho^k(\boldsymbol{\theta}(t)) - \Phi^k(\boldsymbol{\theta}(t)))| \le \lambda\rho(1/me + 2\log m)$.*

The theorem is proved in Appendix D, and it says the first-order stationary point of the surrogate loss $\Phi_\rho^k(\cdot)$ is approximated by $\boldsymbol{\theta}(t)$ in Algorithm 1, in the order of $\mathcal{O}(T^{-2/5})$ (can be improved to $\mathcal{O}(T^{-1/2})$ when $\sigma^2 = \mathcal{O}(T^{-1/2})$). As the gap between $\Phi^k(\cdot)$ and $\Phi_\rho^k(\cdot)$ is $\mathcal{O}(\rho)$, taking small $\rho$ yields small $\Phi^k(\boldsymbol{\theta}(T))$. The unbiased estimators in our theorem are constructed in the next section.

## 6 EXPERIMENTS

In this section, we empirically evaluate the efficacy of the proposed Algorithm 1 in terms of breaking the spurious correlation. More experiments are shown in Appendix E.

Table 1: Test accuracy (%) of ResNet50 on each group of CelebA and Waterbirds.

| Dataset | Method / Group | D-F | D-M | B-F | B-M | Avg | Total | Worst | SA |
|---------|---------------|-----|-----|-----|-----|-----|-------|-------|-----|
| CelebA | RCSV | 92.1 | 94.0 | 92.3 | 91.8 | **92.6** | 92.9 | **91.8** | √ |
| | IRM | 92.8 | 93.1 | 88.9 | 89.3 | 91.0 | 92.3 | 88.9 | √ |
| | GroupDRO | 94.4 | 94.6 | 88.9 | 88.3 | 91.6 | **93.7** | 88.3 | √ |
| | ERMRS$_{YZ}$ | 88.8 | 94.7 | 95.8 | 85.6 | 91.2 | 91.9 | 85.6 | √ |
| | RCSV$_U$ | 88.3 | 97.8 | 96.1 | 76.9 | 89.8 | 93.3 | 76.9 | × |
| | Correlation | 87.6 | 96.3 | 96.8 | 69.4 | 87.5 | 91.9 | 69.4 | × |
| | ERMRS$_Y$ | 91.3 | 97.5 | 91.0 | 62.2 | 85.5 | 93.3 | 62.2 | × |

| Dataset | Method / Group | L-L | L-W | W-L | W-W | Avg | Total | Worst | SA |
|---------|---------------|-----|-----|-----|-----|-----|-------|-------|-----|
| Waterbirds | RCSV | 93.4 | 90.8 | 88.6 | 89.4 | **90.5** | **91.4** | **88.6** | √ |
| | IRM | 93.1 | 87.7 | 87.4 | 89.7 | 89.5 | 90.4 | 87.4 | √ |
| | GroupDRO | 92.0 | 87.8 | 87.9 | 90.3 | 89.5 | 89.7 | 87.8 | √ |
| | ERMRS$_{YZ}$ | 93.2 | 88.5 | 87.9 | 91.3 | 90.2 | 90.6 | 87.9 | √ |
| | RCSV$_U$ | 98.5 | 81.2 | 81.3 | 95.5 | 89.1 | 89.5 | 81.2 | × |
| | Correlation | 98.8 | 74.4 | 70.9 | 93.0 | 84.3 | 85.5 | 70.9 | × |
| | ERMRS$_Y$ | 99.3 | 76.7 | 68.8 | 94.9 | 84.9 | 86.6 | 68.8 | × |

Table 2: Test accuracy (%) of BERT on each group of MultiNLI.

| Dataset | Method / Group | C-WN | C-N | E-WN | E-N | N-WN | N-N | Avg | Total | Worst | SA |
|---------|---------------|------|-----|------|-----|------|-----|-----|-------|-------|-----|
| MultiNLI | RCSV | 77.1 | 95.3 | 83.5 | 79.5 | 81.3 | 78.6 | **82.6** | **81.5** | **78.6** | √ |
| | IRM | 79.7 | 96.9 | 80.4 | 71.7 | 77.2 | 71.7 | 79.6 | 79.9 | 71.7 | √ |
| | GroupDRO | 77.4 | 93.5 | 82.5 | 79.7 | 81.5 | 77.6 | 82.0 | 81.3 | 77.6 | √ |
| | ERMRS$_{YZ}$ | 74.7 | 89.5 | 79.1 | 72.3 | 82.0 | 71.8 | 78.2 | 79.3 | 71.8 | √ |
| | RCSV$_U$ | 79.3 | 95.5 | 82.0 | 74.3 | 78.1 | 70.8 | 80.0 | 80.6 | 70.8 | × |
| | Correlation | 76.2 | 94.7 | 76.7 | 67.9 | 75.5 | 67.9 | 76.5 | 77.0 | 67.9 | × |
| | ERMRS$_Y$ | 82.7 | 95.5 | 79.4 | 72.6 | 79.7 | 67.3 | 79.5 | 81.1 | 67.3 | × |

**Implementation.** The Algorithm 1 can be applied to the regularized training process with either $\widehat{\mathrm{CSV}}(f_{\boldsymbol{\theta}})$ (RCSV) or $\widehat{\mathrm{CSV_U}}(f_{\boldsymbol{\theta}})$ (RCSV$_U$) depending on whether spurious attributes are observable. The implementations is clear after estimating $R_{emp}(f_{\boldsymbol{\theta}(t)}, P)$ and $F^k(\boldsymbol{\theta})$ in (12).

For RCSV, in each step $t$, we let $\hat{R}_{emp}(f_{\boldsymbol{\theta}(t)}, P)$ be the empirical risk over a uniformly drawn batch (size $S$) of data. Then we randomly sample another batch (size $S$) of data with replacement. The probability of each data with class label $k$ and spurious attribute $z$ be sampled is $1/(K_y K_z n_{kz})$. Then the $\hat{\mathcal{L}}_{kz}(f_{\boldsymbol{\theta}(t)})$ in (12) is estimated as the conditional expected risk over this batch of data.

For RCSV$_U$, in each step $t$, the $\hat{R}_{emp}(f_{\boldsymbol{\theta}(t)}, P)$ is as in RCSV. We also randomly sample another mini-batch (batch size $S$) of data with replacement but the probability of data with label $k$ be sampled is $1/(K_y n_k)$. We estimate $F^k(\boldsymbol{\theta})$ in (12) via its empirical counterpart over these sampled data.

As can be seen, all of the resulting $\hat{R}_{emp}(f_{\boldsymbol{\theta}(t)}, P)$ and $\hat{\boldsymbol{F}}^k(\boldsymbol{\theta}(t))$ are unbiased estimators with variance of order $\mathcal{O}(1/S)$ as in Theorem 5. Besides that, our RCSV (resp. RCSV$_U$) can be implemented with (resp. without) observable spurious attributes. The complete implementation of RCSV and RCSV$_U$ are shown in Appendix G.1. When estimating CSV, the data are sampled with weights that are inversely proportional to $n_{kz}$ or $n_k$. The sampling strategy also appears in (Sagawa et al., 2019; Idrissi et al., 2021; Arjovsky et al., 2019) which significantly improves the OOD generalization according to our ablation study in Appendix F.

**Data.** We use the following benchmark datasets with correlation shift (see details in Appendix G.2).

**CelebA (Liu et al., 2015).** An image classification task to recognize a celebrity's hair color ("dark" or "blond"), which is spuriously correlated with the celebrity's gender ("male" or "female"). The data are categorized as 4 groups via the combination of hair color and gender, e.g., "dark-female" (D-M).

**Waterbirds (Sagawa et al., 2019).** An image classification task to recognize a bird as "waterbird" or "landbird", while the bird is spuriously correlated with background "land" or "water". The data are categorized into 4 groups, e.g., "landbird-water" (L-W).

**MultiNLI (Williams et al., 2018).** Given a sentence-pair, the task aims to recognize the relationship between the two sentences, i.e., "entailment", "neutrality", "contradiction". The relationship is spuriously correlated with the presence of negation words. The data are categorized into 6 groups, e.g., "entailment-without negation" (E-W), "contradiction-negation" (C-N).

**CivilComments (Borkan et al., 2019).** A textual classification task to check whether a sentence is toxic or not with the label spuriously correlated with whether any of 8 certain demographic identities are mentioned. The data have 4 groups, e.g., "nontoxic-identity" (N-I), "toxic-nonidentity" (T-N).

Table 3: Test accuracy (%) of BERT on each group of `CivilComments`.

| Dateset | Method / Group | N-N | N-I | T-N | T-I | Avg | Total | Worst | SA |
|---|---|---|---|---|---|---|---|---|---|
| | RCSV | 93.1 | 87.7 | 82.4 | 71.7 | **83.7** | 89.3 | **71.7** | $\sqrt{}$ |
| | IRM | 96.2 | 88.5 | 68.0 | 67.5 | 80.1 | 90.3 | 67.5 | $\sqrt{}$ |
| | GroupDRO | 94.5 | 88.7 | 76.3 | 69.4 | 82.2 | 90.0 | 69.4 | $\sqrt{}$ |
| `CivilComments` | ERMRS$_{YZ}$ | 94.3 | 88.8 | 79.1 | 70.0 | 83.1 | 90.0 | 70.0 | $\sqrt{}$ |
| | RCSV$_U$ | 96.2 | 89.5 | 72.6 | 68.7 | 81.7 | 90.9 | 68.7 | $\times$ |
| | Correlation | 94.1 | 89.2 | 85.2 | 65.5 | 83.5 | 89.6 | 65.5 | $\times$ |
| | ERMRS$_Y$ | 98.0 | 94.4 | 61.0 | 57.2 | 77.7 | **92.3** | 57.2 | $\times$ |

**Setup.** We compare our methods RCSV and RCSV$_U$ with four baseline methods (see Appendix G.3 for details) i.e., ERM with reweighted sampling (ERMRS) (Idrissi et al., 2021), IRM (Arjovsky et al., 2019), GroupDRO (Sagawa et al., 2019), and Correlation (Devansh Arpit, 2019).

The GroupDRO and IRM use the reweighted sampling strategy as in RCSV, while the Correlation uses same one with RCSV$_U$. As these sampling strategies improve the OOD generalization (Idrissi et al., 2021), to make a fair comparison, we also conduct ERMRS with the two sampling strategies. The two ERMRS are respectively denoted as ERMRS$_Y$ and ERMRS$_{YZ}$. The involved 7 methods are categorized as 2 groups, i.e., conducted with observable spurious attributes (RCSV, IRM, GroupDRO, ERMRS$_{YZ}$) and with unobservable spurious attributes (RCSV$_U$, Correlation, ERMRS$_Y$, Correlation).

The backbone models of image (`CelebA`, `Waterbirds`) and textual datasets (`MultiNLI`, `CivilComments`) are respectively ResNet-50 (He et al., 2016) pre-trained on `ImageNet` (Deng et al., 2009) and pre-trained BERT (Devlin et al., 2019). The hyperparameters are in Appendix G.4.

**Main Results.** Our goal is to verify whether all these methods can break the spurious correlation in data. Thus for each dataset, we report the test accuracies on each group of it, as the groups are divided via the combination of class label and spurious attribute. We also report the averaged test accuracies over groups ("Avg"), the test accuracies on the whole test set ("Total", which is in-distribution test accuracy expected for `Waterbirds`), and the worst test accuracies over groups ("Worst"). The results are in Table 1, 2, 3. The "$\sqrt{}$" and "$\times$" for SA (spurious attributes) respectively mean whether the method requires observable spurious attributes. The observations from these tables are as follows.

To check the OOD generalization, a direct way is comparing the column of "Avg" and "Worst" to summarize the results in each group. As can be seen, in terms of the two criteria, the proposed RCSV (resp. RCSV$_U$) consistently achieves better performances, compared to baseline methods with observable (resp. unobservable) spurious attributes. This verifies our methods can improve the OOD generalization. On the other hand, leveraging the observable spurious attributes benefits the OOD generalization since the methods with them consistently exhibits better performances than the ones without them. For example, the discussion in Section 4.3 shows that the estimator of CSV in RCSV with observable spurious attributes is more accurate than the one in RCSV$_U$.

There is a trade-off between the robustness of the model over spurious attributes and the test accuracies on the groups with the same class label, especially for `CelebA` and `Waterbirds`, see "D-F" v.s. "D-M" in `CelebA` for example. The phenomenon illustrates that some spurious correlations are captured for all methods. However, compared to the other methods, our methods have better averaged test accuracies and a smaller gap between the test accuracies over groups with the same spurious attributes. The robustness and test accuracies here respectively correspond to the goals of "robustness" and "in-distribution test accuracy" in Section 5, the improvements support our discussion in Section 5 that splitting the goals of accuracy and robustness enables us easily take a balance between them.

## 7 CONCLUSION

In this paper, we explore the OOD generalization for data with correlation shift. After a formal characterization, we give a sufficient condition to make the model OOD generalizable. The condition is the conditional independence of the model, given the class label. Conditional Spurious Variation, which controls the OOD generalization error, is proposed to measure such independence. Based on this metric, we propose an algorithm with a provable convergence rate to regularize the training process with two estimators of CSV (i.e., RCSV and RCSV$_U$), depending on whether the spurious attributes are observable. Finally, the experiments conducted on the datasets `CelebA`, `Waterbirds`, `MultiNLI`, `CivilComments` verify the efficacy of our methods.

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

# A   LABEL SHIFT

In the sequel, we may omit the subscribe if no obfuscation. Our discussions in the main body of this paper are built upon the assumption that marginal distribution of label $Y$ is invariant i.e., $P_Y = Q_Y$. In this section, we explore OOD generalization without such invariant assumption. Before presenting our discussion, we give the definition of total variation distance.

**Definition 3.** *The total variation distance between two distributions $P, Q$ defined on the same measurable space $\mathcal{X}$ is*

$$\mathrm{TV}(P, Q) = \frac{1}{2} \int_{\mathcal{X}} |dP(x) - dQ(x)|. \tag{15}$$

In Theorem 1, we show that the gap between the performances of the model on training and OOD test data disappears if the model satisfies conditional independence such that $f(X) \perp Z \mid Y$. However, we show by the following example that the gap will not disappear if the marginal distribution of $Y$ also varies across training and test data.

**Example 1.** Suppose $Y, Z \in \{-1, 1\}$ and a specialized loss function

$$\mathcal{L}(f(X), Y) = 1_{\{Y=1\}}(5 - f(X)) + 1_{\{Y=-1\}}(2 + f(X)), \tag{16}$$

where $f(\cdot)$ is any classifier whose output is in $\{-1, 1\}$. Let $P, Q$ be two distributions such that $P_{X|Y,Z} = Q_{X|Y,Z}$ but $P_Y \neq Q_Y$. We suppose $X \perp Z \mid Y$, and thus $f(X) \perp Z \mid Y$. Thus,

$$P_{X|Y}(\boldsymbol{x} \mid y) = \sum_{z \in \mathcal{Z}} P_{X,Z|Y}(\boldsymbol{x}, z \mid y) = P_{X|Y}(\boldsymbol{x} \mid y) \sum_{z \in \mathcal{Z}} P_{Z|Y}(z \mid y), \tag{17}$$

is unrelated to $P_{Z|Y}$. Then we have $P_{X|Y} = Q_{X|Y}$. Thus

$$\mathbb{E}_P[\mathcal{L}(f(X), Y)] = P_Y(Y = 1)\mathbb{E}_P[\mathcal{L}(f(X), Y) \mid Y = 1] + P_Y(Y = -1)\mathbb{E}_P[\mathcal{L}(f(X), Y) \mid Y = -1],$$
$$\mathbb{E}_Q[\mathcal{L}(f(X), Y)] = Q_Y(Y = 1)\mathbb{E}_Q[\mathcal{L}(f(X), Y) \mid Y = 1] + Q_Y(Y = -1)\mathbb{E}_Q[\mathcal{L}(f(X), Y) \mid Y = -1]. \tag{18}$$

Since $P_{X|Y} = Q_{X|Y}$,

$$\begin{aligned} &|\mathbb{E}_P[\mathcal{L}(f(X), Y)] - \mathbb{E}_Q[\mathcal{L}(f(X), Y)]| \\ &= |(P_Y(Y = 1) - Q_Y(Y = 1))(\mathbb{E}_P[\mathcal{L}(f(X), Y) \mid Y = 1] - \mathbb{E}_Q[\mathcal{L}(f(X), Y) \mid Y = -1])| \\ &\geq |4 - 3| \times |P(Y = 1) - Q(Y = 1)| \\ &= |P_Y(Y = 1) - Q_Y(Y = 1)| \\ &= \mathrm{TV}(P_Y, Q_Y), \end{aligned} \tag{19}$$

where $\mathrm{TV}(P_Y, Q_Y)$ is the total variation distance of the marginal distributions $P_Y, Q_Y$. This inequality holds for any $f(X)$, and hence the gap can never be removed by representation learning like what we do in Theorem 1.

The example indicates that under shifted label distribution, the conditional independent model can not generalize on OOD data. Thus, we consider the reweighted loss to fix the bias brought by the shifted label distribution. The formal result is stated as follows.

**Theorem 6.** *Let $P, Q$ be two distributions such that $P_{X|Y,Z} = Q_{X|Y,Z}$ but $P_Y$ does not necessary equals to $Q_Y$. $w(y) : \mathcal{Y} \to \mathbb{R}^+$ is a weighting function satisfies $E_P[w(Y)] = 1$. Then if $f(X) \perp Z \mid Y$,*

$$|\mathbb{E}_P[w(Y)\mathcal{L}(f(X), Y)] - \mathbb{E}_Q[\mathcal{L}(f(X), Y)]| \leq 2M\mathrm{TV}(P_Y^w, Q) \tag{20}$$

*where $P_Y^w$ is the reweighted label distribution defined as $P_Y^w(A) = \int_A w(y)dP_Y(y)$ for any measurable set $A \subset \mathcal{Y}$.*

*Proof.* Because $P_{X|Y,Z} = Q_{X|Y,Z}$ and $f(X) \perp Z \mid Y$, as in Appendix C.1, we have $P(f(X) \mid Y) = Q(f(X) \mid Y)$. Thus

$$\begin{aligned} &|\mathbb{E}_P[w(Y)\mathcal{L}(f(X), Y)] - \mathbb{E}_Q[\mathcal{L}(f(X), Y)]| \\ &= \left| \int_{\mathcal{Y}} w(y)\mathbb{E}_P[\mathcal{L}(f(X), Y) \mid Y = y]dP_Y(y) - \int_{y \in \mathcal{Y}} \mathbb{E}_Q[\mathcal{L}(f(X), Y) \mid Y = y]dQ_Y(y) \right| \\ &\leq \int_{\mathcal{Y}} M |w(y)dP_Y(y) - dQ_Y(y)| \\ &= 2M\mathrm{TV}(P_Y^w, Q_Y). \end{aligned} \tag{21}$$

$\square$

**Remark 2.** *The total variation distance* $\mathrm{TV}(P_Y^w, Q_Y)$ *appears in the upper bound to the gap between the two population risk in* (21). *Moreover, this terms seems to be inevitable since it also appears in the lower bound in* (19).

According to Theorem 6, we have invariance relationship $\mathbb{E}_P[w(Y)\mathcal{L}(f(X), Y)] = \mathbb{E}_Q[\mathcal{L}(f(X), Y)]$ if we can take $w(y) = dQ_Y(y)/dP_Y(y)$. Thus if the label distribution in the test data is available, minimizing the reweighted loss with its weights decided by the ration of two label distributions can guarantee the OOD generalization capability of the model.

However, the label distribution of test data are usually unavailable in practical. Thus for unknown test label distribution, we alternatively chose the weight $w(\cdot)$ to minimize the worst-case upper bound

$$\sup_Q \mathrm{TV}(P_Y^w, Q_Y) = \frac{1}{2} \sup_Q \int_{\mathcal{Y}} |w(y)dP_Y(y) - dQ_Y(y)|, \tag{22}$$

given the training distribution $P$, where the supremum is taken over all distributions $Q$ such that $Q_{X|Y,Z} = P_{X|Y,Z}$. Then by minimizing the reweighted loss under such weight $w(\cdot)$, we get a model with minimized worst-case risk over different distributions.

**Proposition 2.** *Suppose that $\mathcal{Y}$ is a discrete space, then if $P_Y(Y = y) > 0$ for all $y \in \mathcal{Y}$ and $w^*(y) = \frac{1}{|\mathcal{Y}|P_Y(Y=y)}$, we then have*

$$w^*(\cdot) \in \underset{w(\cdot):\mathbb{E}_P[w(Y)]=1}{\arg\min} \left\{ \sup_{Q \in \mathcal{P}} \mathrm{TV}(P_Y^w, Q_Y) \right\}, \tag{23}$$

*where $|\mathcal{Y}|$ is the cardinal of $\mathcal{Y}$.*

*Proof.* From Section A.6.2 in (van der Vaart & Wellner, 2000), we know that $\mathrm{TV}(P_Y^w, Q_Y) = \sup_{A \subset \mathcal{Y}} |\sum_{y \in A} w(y)P_Y(Y = y) - Q_Y(Y = y)|$. Thus

$$\sup_{Q \in \mathcal{P}} \mathrm{TV}(P_Y^w, Q_Y) = \sup_{Q \in \mathcal{P}} \sup_{A \subset \mathcal{Y}} \left| \sum_{y \in A} w(y)P_Y(Y = y) - Q_Y(Y = y) \right|$$

$$= \sup_{Q \in \mathcal{P}} \sum_{y \in \{y':w(y')P_Y(Y=y') \geq Q_Y(Y=y')\}} (w(y)P_Y(Y = y) - Q_Y(Y = y)) \tag{24}$$

$$= 1 - \min_{y \in \mathcal{Y}} w(y)P_Y(Y = y)$$

due to $w(y)P_Y(Y = y) \geq 0$ and $\mathbb{E}_P[w(Y)] = 1$. Then, we have

$$\min_{w(\cdot):\mathbb{E}_P[w(Y)]=1} \sup_{Q \in \mathcal{P}} \mathrm{TV}(P_Y^w, Q_Y) = \min_{w(\cdot):\mathbb{E}_P[w(Y)]=1} \left\{ 1 - \min_{y \in \mathcal{Y}} w(y)P_Y(Y = y) \right\}$$

$$= 1 - \max_{w(\cdot):\mathbb{E}_P[w(Y)]=1} \left\{ \min_{y \in \mathcal{Y}} w(y)P_Y(Y = y) \right\} \tag{25}$$

$$= 1 - w^*(y)P_Y(Y = y)$$

$$= \frac{|\mathcal{Y}| - 1}{|\mathcal{Y}|}.$$

The third equality is due to $|\mathcal{Y}| \min_{y \in \mathcal{Y}} w(y)P_Y(Y = y) \leq \sum_{y \in \mathcal{Y}} w(y)P_Y(Y = y) = 1$, and the equality is taken when $w(\cdot) = w^*(\cdot)$. □

# B  PROOFS IN SECTION 3

In this section, we present the proofs of results in Section 3.

**Proposition 1.** *There exists a population risk $R_{pop}(P, f)$ whose minimizer has nearly perfect performance on the data from $P$, while it fails to generalize to OOD data drawn from another $Q \in \mathcal{P}$.*

*Proof.* Let us consider the following example that

$$X = \begin{pmatrix} Y \cdot \boldsymbol{\mu}_1 \\ Z \cdot \boldsymbol{\mu}_2 \end{pmatrix} + \boldsymbol{\xi}, \tag{26}$$

where $\boldsymbol{\xi} \sim \mathcal{N}(\boldsymbol{0}, I_{d_1+d_2})$, $Y, Z \in \{-1, 1\}$ and follow the standard binomial distribution. Denote the training distribution as $P$. In this example, $Z$ is the spurious attributes. The correlation coefficient between $Y$ and $Z$ is denoted as $\sigma_{YZ}(Q)$ for $Q \in \mathcal{P}$. One can verify that

$$\sigma_{YZ}(Q) = \mathbb{E}_Q[YZ] = Q(Y = Z) - Q(Y \neq Z) = 2Q(Y = Z) - 1. \tag{27}$$

Let us consider the linear classifier $f_{\boldsymbol{\theta}}(\boldsymbol{x}) = \boldsymbol{\theta}^\top \boldsymbol{x}$ and its loss on data $(X, Y)$ is the exponential loss Soudry et al. (2018)

$$\mathcal{L}(f_{\boldsymbol{\theta}}(X), Y) = e^{-Y f_{\boldsymbol{\theta}}(X)}. \tag{28}$$

Thus we can compute the population risk

$$
\begin{aligned}
R_{pop}(P, f_{\boldsymbol{\theta}}) &= \mathbb{E}_P\left[\exp(-Y f_{\boldsymbol{\theta}}(X))\right] \\
&= \mathbb{E}\left[\exp\left(-\boldsymbol{\theta}_1^\top \boldsymbol{\mu}_1 - YZ\boldsymbol{\theta}_2^\top \boldsymbol{\mu}_2 - \boldsymbol{\theta}^\top \boldsymbol{\xi}\right)\right] \\
&= \mathbb{E}\left[\exp\left(-\boldsymbol{\theta}_1^\top \boldsymbol{\mu}_1 - \boldsymbol{\theta}_2^\top \boldsymbol{\mu}_2 - \boldsymbol{\theta}^\top \boldsymbol{\xi}\right) \mid Y = Z\right] P(Y = Z) \\
&\quad + \mathbb{E}\left[\exp\left(-\boldsymbol{\theta}_1^\top \boldsymbol{\mu}_1 + \boldsymbol{\theta}_2^\top \boldsymbol{\mu}_2 + \boldsymbol{\theta}^\top \boldsymbol{\xi}\right) \mid Y \neq Z\right] P(Y \neq Z) \\
&= \left(\frac{1 + \sigma_{YZ}(P)}{2}\right) \exp\left(-\boldsymbol{\theta}_1^\top \boldsymbol{\mu}_1 - \boldsymbol{\theta}_2^\top \boldsymbol{\mu}_2 + \frac{\|\boldsymbol{\theta}\|^2}{2}\right) \\
&\quad + \left(\frac{1 - \sigma_{YZ}(P)}{2}\right) \exp\left(-\boldsymbol{\theta}_1^\top \boldsymbol{\mu}_1 + \boldsymbol{\theta}_2^\top \boldsymbol{\mu}_2 + \frac{\|\boldsymbol{\theta}\|^2}{2}\right),
\end{aligned}
\tag{29}
$$

Since $R_{pop}(P, f_{\boldsymbol{\theta}})$ is continuous w.r.t. to $\sigma_{YZ}(P)$ and $\boldsymbol{\theta}$, we have that $\boldsymbol{\theta}^*(P) = \arg\min_{\boldsymbol{\theta}} R_{pop}(P, f_{\boldsymbol{\theta}})$ is continuous to $\sigma_{YZ}(P)$. Since $\sigma_{YZ}(P) \in [-1, 1]$, we conclude $\|\boldsymbol{\theta}^*(P)\|$ is upper bounded. W.o.l.g. we assume $\|\boldsymbol{\theta}^*(P)\| \leq 1$ for any $\sigma_{YZ}(P) \in [-1, 1]$, then for any $\sigma_{YZ}(P)$ we know $\boldsymbol{\theta}^*(P)$ satisfies the first order optimality condition such that

$$
\begin{aligned}
0 &= \left(\frac{1 + \sigma_{YZ}(P)}{2}\right) (\boldsymbol{\theta}^*(P) - \boldsymbol{\mu}) \exp\left(-\boldsymbol{\theta}^\top \boldsymbol{\mu} + \frac{\|\boldsymbol{\theta}\|^2}{2}\right) \\
&\quad + \left(\frac{1 - \sigma_{YZ}(P)}{2}\right) (\boldsymbol{\theta}^*(P) - \tilde{\boldsymbol{\mu}}) \exp\left(-\boldsymbol{\theta}^\top \tilde{\boldsymbol{\mu}} + \frac{\|\boldsymbol{\theta}\|^2}{2}\right).
\end{aligned}
\tag{30}
$$

where $\tilde{\boldsymbol{\mu}} = (\boldsymbol{\mu}_1^\top, -\boldsymbol{\mu}_2^\top)^\top$. Thus for $\sigma_{YZ}(P) \neq \pm 1$ we have

$$\boldsymbol{\theta}^*(P) - \boldsymbol{\mu} = \left(\frac{1 - \sigma_{YZ}(P)}{1 + \sigma_{YZ}(P)}\right) (\tilde{\boldsymbol{\mu}} - \boldsymbol{\theta}^*(P)) \exp\left(2\boldsymbol{\theta}_2^\top \boldsymbol{\mu}_2\right). \tag{31}$$

Thus we can take a $\sigma_{YZ}(P) \to 1$ to make $\|\boldsymbol{\theta}^*(P) - \boldsymbol{\mu}\| \leq \epsilon\|\boldsymbol{\mu}\|$ for any small $\epsilon$ where the $\epsilon$ can be independent of $\boldsymbol{\mu}$.

Now we show that the linear model $f_{\boldsymbol{\theta}^*(P)}(\cdot)$ with its prediction on $Y$ as $\text{sign}(f_{\boldsymbol{\theta}^*(P)}(\cdot))$ can make correct prediction with high probability. Let us see the error of linear model $f_{\boldsymbol{\theta}^*(P)}(\cdot)$ on the data from training distribution $P$. Simple algebra show that $\boldsymbol{\theta}^{*\top}(P)X \mid Y \sim \mathcal{N}(Y\boldsymbol{\theta}_1^{*\top}(P)\boldsymbol{\mu}_1 + Z\boldsymbol{\theta}_2^{*\top}(P)\boldsymbol{\mu}_2, \|\boldsymbol{\theta}_1^{*\top}(P)\|^2)$. Then under condition of $Y = Z$, we have

$$\boldsymbol{\theta}^{*\top}(P)X - Y\|\boldsymbol{\mu}\|^2 = Y\boldsymbol{\theta}^{*\top}(P)\boldsymbol{\mu} - Y\|\boldsymbol{\mu}\|^2 + \boldsymbol{\theta}^{*\top}(P)\boldsymbol{\xi} = Y(\boldsymbol{\theta}^*(P) - \boldsymbol{\mu})^\top \boldsymbol{\mu} + \boldsymbol{\theta}^{*\top}(P)\boldsymbol{\xi}. \tag{32}$$

Thus from the sub-Gaussian property of Gaussian random variable, for any $\delta > 0$

$$
\begin{aligned}
&P_{X|Y}\left(\left|\boldsymbol{\theta}^{*\top}(P)X - Y\|\boldsymbol{\mu}\|^2\right| \geq \delta \mid Y\right) \\
&= P_{X|Y}\left(\left|\boldsymbol{\theta}^{*\top}(P)X - Y\|\boldsymbol{\mu}\|^2\right| \geq \delta \mid Y, Y = Z\right)\left(\frac{1 + \sigma_{YZ}(P)}{2}\right) \\
&\quad + P_{X|Y}\left(\left|\boldsymbol{\theta}^{*\top}(P)X - Y\|\boldsymbol{\mu}\|^2\right| \geq \delta \mid Y, Y \neq Z\right)\left(\frac{1 - \sigma_{YZ}(P)}{2}\right) \\
&\leq P_{X|Y}\left(\left|\boldsymbol{\theta}^{*\top}(P)\boldsymbol{\xi}\right| \geq \delta - \epsilon\|\boldsymbol{\mu}\|^2 \mid Y, Y = Z\right)\left(\frac{1 + \sigma_{YZ}(P)}{2}\right) + \left(\frac{1 - \sigma_{YZ}(P)}{2}\right) \\
&\leq \exp\left(-\frac{(\delta - \epsilon\|\boldsymbol{\mu}\|^2)^2}{2\|\boldsymbol{\theta}^*(P)\|^2}\right)\left(\frac{1 + \sigma_{YZ}(P)}{2}\right) + \left(\frac{1 - \sigma_{YZ}(P)}{2}\right).
\end{aligned}
\tag{33}
$$

We may take $\delta = \|\boldsymbol{\mu}\|^2/2 + \epsilon\|\boldsymbol{\mu}\|^2$, and $\sigma_{YZ}(P) \to 1$, due to $\|\boldsymbol{\theta}(\sigma_{YZ}(P))\| \leq 1$ and for a large enough $\|\boldsymbol{\mu}\|$, with a high probability, we have

$$Y\|\boldsymbol{\mu}\|^2 - \left(\frac{1}{2} + \epsilon\right)\|\boldsymbol{\mu}\|^2 \leq \boldsymbol{\theta}^{*\top}(P)X \leq Y\|\boldsymbol{\mu}\|^2 + \left(\frac{1}{2} + \epsilon\right)\|\boldsymbol{\mu}\|^2. \tag{34}$$

Since $\epsilon \to 0$ for $\sigma_{YZ}(P) \to 1$, we have proved that the population minimizer $\boldsymbol{\theta}^*(P)$ has nearly perfect performance on the data from training distribution.

However, a similar argument of (33) shows that for data drawn from distribution $Q \in \mathcal{P}$

$$Q_{X|Y}\left(\left|\boldsymbol{\theta}^{*\top}(P)X - Y(\|\boldsymbol{\mu}_1\|^2 - \|\boldsymbol{\mu}_2\|^2)\right| \geq \delta \mid Y\right) \leq \exp\left(-\frac{(\delta - \epsilon\|\boldsymbol{\mu}\|^2)^2}{2\|\boldsymbol{\theta}^*(P)\|^2}\right)\left(\frac{1 - \sigma_{YZ}(Q)}{2}\right) \\ + \left(\frac{1 + \sigma_{YZ}(Q)}{2}\right) \tag{35}$$

for any $\delta > 0$. Again, by taking $\sigma_{YZ}(Q) \to -1$ we get

$$Y(\|\boldsymbol{\mu}_1\|^2 - \|\boldsymbol{\mu}_2\|^2) - \left(\frac{1}{2} + \epsilon\right)\left(\|\boldsymbol{\mu}_1\|^2 + \|\boldsymbol{\mu}_2\|\right) \leq \boldsymbol{\theta}^{*\top}(P)X \\ \leq Y(\|\boldsymbol{\mu}_1\|^2 - \|\boldsymbol{\mu}_2\|^2) + \left(\frac{1}{2} + \epsilon\right)\left(\|\boldsymbol{\mu}_1\|^2 + \|\boldsymbol{\mu}_2\|\right) \tag{36}$$

with high probability. We can take, for example, $\|\boldsymbol{\mu}_2\|^2 > \left(\frac{3+2\epsilon}{1-\epsilon}\right)\|\boldsymbol{\mu}_1\|^2$ for $\epsilon \to 0$, then under $Y = -1$, the inequality becomes

$$0 < \left(\frac{1}{2} - \epsilon\right)\|\boldsymbol{\mu}_2\|^2 - \left(\frac{3}{2} + \epsilon\right)\|\boldsymbol{\mu}_1\|^2 \leq \boldsymbol{\theta}^{*\top}(P)X \leq \left(\frac{3}{2} + \epsilon\right)\|\boldsymbol{\mu}_2\|^2 - \left(\frac{1}{2} - \epsilon\right)\|\boldsymbol{\mu}_1\|^2, \tag{37}$$

which shows the prediction given by $f_{\boldsymbol{\theta}^*(P)}(\cdot)$ for $Y = -1$ is incorrect with high probability. A similar argument can be verified for $Y = 1$. Then we complete our proof. $\square$

**Theorem 1.** *For model $f(\cdot)$ satisfying $f(X) \perp Z \mid Y$, the conditional distribution $Y \mid f(X)$ and population risk $\mathbb{E}_Q[\mathcal{L}(f(X), Y)]$ are invariant with $(X, Y) \sim Q_{X,Y}$ such that $Q \in \mathcal{P}$.*

*Proof.* The difference of $Y \mid f(X)$ for any $(X, Y) \sim Q$ with $Q \in \mathcal{P}$ originates from the different spurious correlation i.e., the different $Q_{Z|Y}$. Thus to obtain our result, it is suffice to prove that the distribution of $Y \mid f(X)$ is independent of $Q_{Z|Y}$. To see this, for any measurable sets $A, B \subset \mathcal{Y}$,

$$Q_{Y|X}(Y \in A \mid f(X) \in B)$$
$$= \frac{Q_{X|Y}(f(X) \in B \mid Y \in A)Q_Y(Y \in A)}{Q_{X|Y}(f(X) \in B \mid Y \in A)Q_Y(Y \in A) + Q_{X|Y}(f(X) \in B \mid Y \notin A)Q_Y(Y \notin A)} \tag{38}$$
$$= \frac{1}{1 + \frac{Q_{X|Y}(f(X) \in B|Y \notin A)Q_Y(Y \notin A)}{Q_{X|Y}(f(X) \in B|Y \in A)Q_Y(Y \in A)}}.$$

As $Q_Y(Y \notin A)/Q_Y(Y \in A)$ is invariant across $Q \in \mathcal{P}$. Then for the $Q_{X|Y}(f(X) \in B \mid Y \notin A)/Q_{X|Y}(f(X) \in B \mid Y \in A)$, we see

$$\frac{Q_{X|Y}(f(X) \in B \mid Y \notin A)}{Q_{X|Y}(f(X) \in B \mid Y \in A)} = \frac{\int_{\mathcal{Z}} Q_{X,Z|Y}(f(X) \in B, z \mid Y \notin A)dz}{\int_{\mathcal{Z}} Q_{X,Z|Y}(f(X) \in B, z \mid Y \in A)dz}$$
$$= \frac{Q_{X|Y}(f(X) \in B \mid Y \notin A)\int_{\mathcal{Z}} Q_{Z|Y}(z \mid Y \notin A)dz}{Q_{X|Y}(f(X) \in B \mid Y \in A)\int_{\mathcal{Z}} Q_{Z|Y}(z \mid Y \in A)dz}, \tag{39}$$

where the second equality is from the independent condition that $f(X) \perp Z \mid Y$. From the calculation, we figure out that the distribution of $Y \mid f(X)$ is independent of spurious correlation $P_{Z|Y}$ due to the arbitrariness of $A, B \in \mathcal{Y}$. Then we prove $Y \mid f(X)$ is invariant over $Q \in \mathcal{P}$..

To provide the invariance of $\mathbb{E}_Q[\mathcal{L}(f(X), Y)]$, it is suffice to show that for the union distribution of $(f(X), Y)$ is invariant w.r.t. $Q$ for $Q \in \mathcal{P}$. Thus for any sets $A, B \subset \mathcal{Y}$ and $(X, Y) \sim Q \in \mathcal{P}$

$$Q_{X,Y}(Y \in A, f(X) \in B) = Q_{X|Y}(f(X) \in B \mid Y \in A)Q_Y(Y \in A)$$
$$= Q_Y(Y \in A)\int_{\mathcal{Z}} Q_{X,Z|Y}(f(X) \in B, z \mid Y \in A)dz \tag{40}$$
$$= Q_Y(Y \in A)Q_{X|Y}(f(X) \in B \mid Y \in A)\int_{\mathcal{Z}} Q_{Z|Y}(z \mid Y \in A)dz.$$

Since $f(X) \perp Z \mid Y$, we figure out the $Q_{X,Y}(Y \in A, f(X) \in B)$ is independent of with spurious correlation $Q_{Z|Y}$. Then due to the arbitrary of $A, B \in \mathcal{Y}$, we summarize that the union distribution of $(f(X), Y)$ is invariant w.r.t. $Q$ for $Q \in \mathcal{P}$. Then the proof is completed. $\qquad\square$

# C   PROOFS IN SECTION 4

## C.1   PROOFS IN SECTION 4.1

**Theorem 2.** *For any $Q \in \mathcal{P}$, we have*

$$\sup_{Q \in \mathcal{P}} |R_{emp}(f, P) - R_{pop}(f, Q)| \leq |R_{emp}(f, P) - R_{pop}(f, P)| + \text{CSV}(f) \tag{4}$$

*Proof.* This theorem can be computed via the assumption in (1). We have

$$\mathbb{E}_P[\mathcal{L}(f(X), Y)] = \mathbb{E}_P\left[\mathbb{E}[\mathcal{L}(f(X), Y) \mid Y, Z]\right]$$
$$= \mathbb{E}_Y\left[\int_{\mathcal{Z}} \mathbb{E}_X[\mathcal{L}(f(X), Y) \mid Y, Z = z]dP(z \mid Y)\right]. \tag{41}$$

Due to (1), the first expectation is invariant with $P, Q \in \mathcal{P}$, while the second expectation is a function of $(Y, z)$ independent the choice of $P$. Thus

$$|\mathbb{E}_P[\mathcal{L}(f(X), Y)] - \mathbb{E}_Q[\mathcal{L}(f(X), Y)]| \leq \mathbb{E}\left[\sup_{z_1, z_2} |\mathbb{E}[\mathcal{L}(f(X), Y) \mid Y, z_1] - \mathbb{E}[\mathcal{L}(f(X), Y) \mid Y, z_2]|\right]$$
$$\leq \text{CSV}(f), \tag{42}$$

where the last inequality is due to the loss function is non-negative. Then due to

$$\sup_{Q \in \mathcal{P}} |R_{emp}(f, P) - R_{pop}(f, Q)| \leq |R_{emp}(f, P) - R_{pop}(f, P)| + \sup_{Q \in \mathcal{P}} |R_{pop}(f, P) - R_{pop}(f, Q)|$$
$$\leq |R_{emp}(f, P) - R_{pop}(f, P)| + \text{CSV}(f), \tag{43}$$

we get the theorem. $\qquad\square$

Next we provide the definitions of KL-divergence, mutual information, and conditional mutual information which are useful to prove Theorem 3.

**Definition 4** (KL-Divergence). *Let $P, Q$ be two distributions with the same support and $P$ is absolutely continuous w.r.t. $Q$. Then the KL divergence from $Q$ to $P$ is*

$$D_{\text{KL}}(P \parallel Q) = \mathbb{E}_{V \sim P}\left[\log \frac{dP}{dQ}(V)\right], \tag{44}$$

*where $\frac{dP}{dQ}$ is the Radon–Nikodym derivative of $P$ w.r.t. $Q$.*

**Definition 5** (Mutual Information). *For random variables $V_1, V_2$ with joint distribution $P_{V_1, V_2}$, the mutual information between them is*

$$I(V_1; V_2) = D_{\text{KL}}(P_{V_1, V_2} \parallel P_{V_1} \times P_{V_2}). \tag{45}$$

**Definition 6** (Conditional Mutual Information). *For three random variables $U, V, W$, the mutual information between $U, V$ conditional on $W$ is*

$$I(U; V \mid W) = \mathbb{E}_{w \sim P_W}\left[I(U \mid W = w; V \mid W = w)\right]. \tag{46}$$

Before presenting the proof of Theorem 3, we need the following lemma.

**Lemma 1.** *Let $U, V, W$ be three random variables such that $U$ and $W$ are independent with each other, then*

$$I(W; U + V) \leq I(W; V \mid U). \tag{47}$$

*Proof.* By Data Processing Inequality (Xu & Raginsky, 2017), we have

$$I(W; U + V) \leq I(W; U, V) = I(W; U) + I(W; V \mid U) = I(W; V \mid U) \tag{48}$$

Thus the proof is completed. $\qquad\square$

Now we are ready to give the proof of Theorem 3.

**Theorem 3.** *Let model $f_{\boldsymbol{\theta}}(\cdot)$ parameterized by $\boldsymbol{\theta} \in \Theta \subset \mathbb{R}^d$, and is trained on $\mathcal{S} = \{(\boldsymbol{x}_i, y_i)\}_{i=1}^n$ from distribution $P$, with the spurious attributes of $\boldsymbol{x}_i$ is $z_i$. If the learned model $f_{\boldsymbol{\theta}_{\mathcal{S}}}(\cdot) \perp \mathcal{S}_{\boldsymbol{z}} \mid \mathcal{S}_y$[3]*

$$\mathcal{E}_{gen}(f_{\boldsymbol{\theta}_{\mathcal{S}}}, P) \leq \inf_g \sqrt{\frac{M^2}{4n} \left( I(\mathcal{S}_{\boldsymbol{x}-g(z)}, \mathcal{S}_y; f_{\boldsymbol{\theta}_{\mathcal{S}}} \mid \mathcal{S}_y, \mathcal{S}_{g(z)}) + I(\mathcal{S}_y; f_{\boldsymbol{\theta}_{\mathcal{S}}}) \right)}, \tag{5}$$

*where $\mathcal{E}_{gen}(f_{\boldsymbol{\theta}_{\mathcal{S}}}, P) = |\mathbb{E}[R_{emp}(f_{\boldsymbol{\theta}_{\mathcal{S}}}, P)] - R_{pop}(f_{\boldsymbol{\theta}_{\mathcal{S}}}, P)|$, $g(\cdot)$ is any measurable function, $\mathcal{S}_{\boldsymbol{x}-g(z)} = \{\boldsymbol{x}_i - g(z_i)\}_{i=1}^n$, $\mathcal{S}_y = \{y_i\}_{i=1}^n$.*

*Proof.* Let $\tilde{\mathcal{S}} = \{(\tilde{x}_i, \tilde{y}_i)\}$ be another $n$ samples drawn from $P$ independent of $\mathcal{S}$. W.o.l.g., we assume $\mathbb{E}_{P \times f_{\boldsymbol{\theta}_{\mathcal{S}}}}[\mathcal{L}(f_{\boldsymbol{\theta}_{\mathcal{S}}}(\boldsymbol{x}), y)] = 0$, otherwise we can replace $\mathcal{L}(f_{\boldsymbol{\theta}_{\mathcal{S}}}(\boldsymbol{x}_i), y_i)$ with $\mathcal{L}(f_{\boldsymbol{\theta}_{\mathcal{S}}}(\boldsymbol{x}_i), y_i) - \mathbb{E}_{P \times f_{\boldsymbol{\theta}_{\mathcal{S}}}}[\mathcal{L}(f_{\boldsymbol{\theta}_{\mathcal{S}}}(\boldsymbol{x}), y)]$. For any $\lambda > 0$ by Donsker-Varadhan's inequality,

$$D_{\mathrm{KL}}(P_{\mathcal{S} \times f_{\boldsymbol{\theta}_{\mathcal{S}}}} \parallel P_{\mathcal{S}} \times P_{f_{\boldsymbol{\theta}_{\mathcal{S}}}}) \geq \mathbb{E}_{\mathcal{S} \times f_{\boldsymbol{\theta}_{\mathcal{S}}}} \left[ \frac{\lambda}{n} \sum_{i=1}^n \mathcal{L}(f_{\boldsymbol{\theta}_{\mathcal{S}}}(x_i), y_i) \right] - \log \mathbb{E}_{\tilde{\mathcal{S}} \times f_{\boldsymbol{\theta}_{\mathcal{S}}}} \left[ \exp \left( \frac{\lambda}{n} \sum_{i=1}^n \mathcal{L}(f_{\boldsymbol{\theta}_{\mathcal{S}}}(\tilde{x}_i), \tilde{y}_i) \right) \right]. \tag{49}$$

Then for any $\boldsymbol{\theta}$, $\lambda > 0$, and Lebesgue measurable function $g(\cdot)$,

$$\begin{aligned}
\lambda \mathbb{E}[R_{emp}(f_{\boldsymbol{\theta}_{\mathcal{S}}}, P)] &\leq D_{\mathrm{KL}}(P_{\mathcal{S} \times f_{\boldsymbol{\theta}_{\mathcal{S}}}} \parallel P_{\mathcal{S}} \times P_{f_{\boldsymbol{\theta}_{\mathcal{S}}}}) + \log \mathbb{E}_{\tilde{\mathcal{S}} \times \boldsymbol{\theta}_{\mathcal{S}}} \left[ \exp \left( \frac{\lambda}{n} \sum_{i=1}^n \mathcal{L}(f_{\boldsymbol{\theta}_{\mathcal{S}}}(\tilde{x}_i), \tilde{y}_i) \right) \right] \\
&\overset{a}{\leq} I(\mathcal{S}_{\boldsymbol{x}}, \mathcal{S}_y; f_{\boldsymbol{\theta}_{\mathcal{S}}}) + \frac{\lambda^2 M^2}{8n} \\
&= I(\mathcal{S}_{\boldsymbol{x}}; f_{\boldsymbol{\theta}_{\mathcal{S}}} \mid \mathcal{S}_y) + I(\mathcal{S}_y; f_{\boldsymbol{\theta}_{\mathcal{S}}}) + \frac{\lambda^2 M^2}{8n} \\
&\overset{b}{\leq} I(\mathcal{S}_{\boldsymbol{x}-g(z)}; f_{\boldsymbol{\theta}_{\mathcal{S}}} \mid \mathcal{S}_y, \mathcal{S}_{g(z)}) + I(\mathcal{S}_y; f_{\boldsymbol{\theta}_{\mathcal{S}}}) + \frac{\lambda^2 M^2}{8n},
\end{aligned} \tag{50}$$

where $a$ is due to the definition of mutual information, $\mathcal{L}(f_{\boldsymbol{\theta}_{\mathcal{S}}}(\tilde{\boldsymbol{x}}_i), y_i)$ is $\frac{M}{2}$-sub Gaussian, $b$ is from Lemma 1, and the last equality is due to the conditional independence of the model. Thus, we conclude that

$$\mathbb{E}[R_{emp}(f_{\boldsymbol{\theta}_{\mathcal{S}}}, P)] \leq \inf_g \sqrt{\frac{M^2}{4n} \left( I(\mathcal{S}_{\boldsymbol{x}-g(z)}; f_{\boldsymbol{\theta}_{\mathcal{S}}} \mid \mathcal{S}_y, \mathcal{S}_{g(z)}) + I(\mathcal{S}_y; f_{\boldsymbol{\theta}_{\mathcal{S}}}) \right)}. \tag{51}$$

Thus, we complete the proof. $\qquad \square$

### C.2 PROOFS IN SECTION 4.2

Let $\mathcal{F}(\Theta)$ and $\|\cdot\|_{L_\infty}$ respectively be the parameterized function class and $L_\infty$-norm on $\mathcal{F}(\Theta)$ defined as

$$\|f_{\boldsymbol{\theta}_1} - f_{\boldsymbol{\theta}_2}\|_{L_\infty} = \sup_{\boldsymbol{x}} |f_{\boldsymbol{\theta}_1}(\boldsymbol{x}) - f_{\boldsymbol{\theta}_2}(\boldsymbol{x})| \tag{52}$$

for any $f_{\boldsymbol{\theta}_1}, f_{\boldsymbol{\theta}_2} \in \mathcal{F}(\Theta)$. To provide the proof of Theorem 4, we need the following definition of covering number.

**Definition 7.** *A $\epsilon$-cover of metric space $(\epsilon, \mathcal{F}(\Theta), \|\cdot\|_{L_\infty})$ is any point set $\{f_{\boldsymbol{\theta}_i}(\cdot)\} \subseteq \mathcal{F}(\Theta)$ such that for any $f_{\boldsymbol{\theta}}(\cdot) \in \mathcal{F}(\Theta)$, there exists $\boldsymbol{\theta}_i$ satisfies $\|f_{\boldsymbol{\theta}} - f_{\boldsymbol{\theta}_i}\|_{L_\infty} \leq \epsilon$. The covering number $N(\epsilon, \mathcal{F}(\Theta), \|\cdot\|_{L_\infty})$ is the cardinality of the smallest $\epsilon$-cover.*

**Theorem 4.** *Under Assumption 1 and 2, if $\inf_{k \in [K_y], z \in [K_z]} n_{kz}/n_k = \mathcal{O}(1)$, then*

$$\mathrm{CSV}(f_{\boldsymbol{\theta}}) \leq \widehat{\mathrm{CSV}}(f_{\boldsymbol{\theta}}) + \mathcal{O}\left( \frac{\log(1/\delta)}{\sqrt{n}} \right) \tag{8}$$

*holds with probability at least $1 - \delta$ for any $\boldsymbol{\theta} \in \Theta, \delta > 0$.*

---

[3] $\boldsymbol{\theta}_{\mathcal{S}}$ is the learned parameters depends on training set $\mathcal{S}$. $f_{\boldsymbol{\theta}_{\mathcal{S}}}(\cdot)$ is a random element that takes values in a functional space (i.e., model space), details can be referred to (Shiryaev, 2016).

*Proof.* First, for any given $\boldsymbol{\theta} \in \Theta$ and given $Y = k, Z = z$, due to $0 \leq \mathcal{L}(f_{\boldsymbol{\theta}}(X), Y) \leq M$, by Azuma-Hoeffding's inequality (Corollary 2.20 in (Wainright, 2019)), we know that the with probability at least $1 - \delta$

$$\hat{\mathcal{L}}_{kz}(f_{\boldsymbol{\theta}}) - M\sqrt{\frac{\log(2/\delta)}{2n_{kz}}} \leq \mathcal{L}_{kz}(f_{\boldsymbol{\theta}}) \leq \hat{\mathcal{L}}_{kz}(f_{\boldsymbol{\theta}}) + M\sqrt{\frac{\log(2/\delta)}{2n_{kz}}}. \tag{53}$$

Then we see

$$\sup_{z_1, z_2} \left( \mathcal{L}_{kz_1}(f_{\boldsymbol{\theta}}) - \mathcal{L}_{kz_2}(f_{\boldsymbol{\theta}}) \right) \leq \sup_{z_1, z_2} \left[ \left( \mathcal{L}_{kz_1}(f_{\boldsymbol{\theta}}) - \hat{\mathcal{L}}_{kz_1}(f_{\boldsymbol{\theta}}) \right) - \left( \mathcal{L}_{kz_2}(f_{\boldsymbol{\theta}}) - \hat{\mathcal{L}}_{kz_2}(f_{\boldsymbol{\theta}}) \right) \right]$$

$$+ \sup_{z_1, z_2} \left( \hat{\mathcal{L}}_{kz_1}(f_{\boldsymbol{\theta}}) - \hat{\mathcal{L}}_{kz_2}(f_{\boldsymbol{\theta}}) \right) \tag{54}$$

$$\leq M \log\left( \frac{2}{K_z \delta} \right) \sup_{z_1, z_2} \left( \sqrt{\frac{1}{2n_{kz_1}}} + \sqrt{\frac{1}{2n_{kz_2}}} \right) + \sup_{z_1, z_2} \left( \hat{\mathcal{L}}_{kz_1}(f_{\boldsymbol{\theta}}) - \hat{\mathcal{L}}_{kz_2}(f_{\boldsymbol{\theta}}) \right)$$

holds with probability at least $1 - \delta$. Since the function class $\mathcal{F}(\Theta)$ is bounded by $M$ under $\|\cdot\|_{L_\infty}$, it has finite covering number $N(\epsilon, \mathcal{F}(\Theta), \|\cdot\|_{L_\infty})$. Let $f_{\boldsymbol{\theta}_1}(\cdot), \cdots, f_{\boldsymbol{\theta}_N}(\cdot) \in \mathcal{F}(\Theta)$ be a $\epsilon$-covering of $\mathcal{F}(\Theta)$ with $N \leq N(\epsilon, \mathcal{F}(\Theta), \|\cdot\|_{L_\infty})$ such that $\forall f_{\boldsymbol{\theta}} \in \mathcal{F}(\Theta), \exists q \in \{1, \cdots, N\}, \|f_{\boldsymbol{\theta}} - f_{\boldsymbol{\theta}_q}\|_{L_\infty} \leq \epsilon$. Thus combining the above inequality, for any $f_{\boldsymbol{\theta}}(\cdot)$ and its corresponded $f_{\boldsymbol{\theta}_q}(\cdot)$, we have

$$\sup_{z_1, z_2} \left( \mathcal{L}_{kz_1}(f_{\boldsymbol{\theta}}) - \mathcal{L}_{kz_2}(f_{\boldsymbol{\theta}}) \right) \leq \sup_{z_1, z_2} \left[ \left( \mathcal{L}_{kz_1}(f_{\boldsymbol{\theta}}) - \mathcal{L}_{kz_1}(f_{\boldsymbol{\theta}_q}) \right) + \left( \mathcal{L}_{kz_2}(f_{\boldsymbol{\theta}_q}) - \mathcal{L}_{kz_2}(f_{\boldsymbol{\theta}}) \right) \right]$$

$$+ \sup_{z_1, z_2} \left( \mathcal{L}_{kz_1}(f_{\boldsymbol{\theta}_q}) - \mathcal{L}_{kz_2}(f_{\boldsymbol{\theta}_q}) \right)$$

$$\overset{a}{\leq} 2\epsilon + M \left( \log\left( \frac{2}{K_z \delta} \right) + N(\mathcal{F}(\Theta), \epsilon, \|\cdot\|_{L_\infty}) \right) \sup_{z_1, z_2} \left( \sqrt{\frac{1}{2n_{kz_1}}} + \sqrt{\frac{1}{2n_{kz_2}}} \right) \tag{55}$$

$$+ \sup_{z_1, z_2} \left( \hat{\mathcal{L}}_{kz_1}(f_{\boldsymbol{\theta}_q}) - \hat{\mathcal{L}}_{kz_2}(f_{\boldsymbol{\theta}_q}) \right)$$

holds with probability at least $1 - \delta$ for any $\epsilon > 0$. Here the inequality $a$ is due to the definition of $L_\infty$-norm on $\mathcal{F}(\Theta)$.

On the other hand, as

$$\text{CSV}(f_{\boldsymbol{\theta}}) = \sum_{k=1}^{K_y} \sup_{z_1, z_2} \left( \mathcal{L}_{kz_1}(f_{\boldsymbol{\theta}}) - \mathcal{L}_{kz_2}(f_{\boldsymbol{\theta}}) \right) P(Y = k), \tag{56}$$

We estimate the $P(Y = k)$ with its empirical counterpart $n_k = \sum_{z=1}^{K_z} n_{kz}/n$. For bounded sub-Gaussian variable $\mathbf{1}_{\{Y=k\}}$, we have

$$\mathbb{E}\left[ \mathbf{1}_{\{Y=k\}} \right] - \frac{1}{n} \sum_{i=1}^{n} \mathbf{1}_{\{y_i=k\}} = P(Y = k) - \hat{p}_k \leq \sqrt{\frac{\log(1/\delta)}{2n}} \tag{57}$$

holds with probability at least $1 - \delta$. Plugging this into (56) and combining (55) we get

$$\text{CSV}(f_{\boldsymbol{\theta}}) \leq \frac{1}{n} \sum_{k=1}^{K_y} \sup_{z_1, z_2} \left( \mathcal{L}_{kz_1}(f_{\boldsymbol{\theta}}) - \mathcal{L}_{kz_2}(f_{\boldsymbol{\theta}}) \right) n_k + K_y \sqrt{\frac{\log(2K_y/\delta)}{2n}}$$

$$\leq \sum_{k=1}^{K_y} \sup_{z_1, z_2} \left( \hat{\mathcal{L}}_{kz_1}(f_{\boldsymbol{\theta}}) - \hat{\mathcal{L}}_{kz_2}(f_{\boldsymbol{\theta}}) \right) \hat{p}_k + K_y \sqrt{\frac{\log(2K_y/\delta)}{2n}} \tag{58}$$

$$+ \inf_{\epsilon} \left\{ 2\epsilon + M \left( \log\left( \frac{2}{K_z \delta} \right) + N(\mathcal{F}(\Theta), \epsilon, \|\cdot\|_{L_\infty}) \right) \right\} \sum_{k=1}^{K_y} \sup_{z_1, z_2} \left( \sqrt{\frac{1}{2n_{kz_1}}} + \sqrt{\frac{1}{2n_{kz_2}}} \right) \hat{p}_k$$

holds with probability at least $1 - \delta$ due to the definition of $\hat{p}_k$. Then suppose $\inf_{k \in [K_y], z \in [K_z]} n_{kz}/n_k \geq \alpha$, we have

$$\sum_{k=1}^{K_y} \sup_{z_1, z_2} \left( \sqrt{\frac{1}{2n_{kz_1}}} + \sqrt{\frac{1}{2n_{kz_2}}} \right) \hat{p}_k \leq \frac{1}{n} \sum_{k \in [K_y]} \frac{\sqrt{2} n_k}{\min_{k \in [K_y], z \in [K_z]} \sqrt{n_{kz}}}$$

$$\leq \sqrt{\frac{2}{\alpha}} \sum_{k \in [K_y]} \frac{\sqrt{n_k}}{n} \tag{59}$$

$$\leq \sqrt{\frac{2K_y}{\alpha n}},$$

where the last inequality is due to the Schwarz's inequality. Combining this with (58), we get

$$\text{CSV}(f_{\boldsymbol{\theta}}) \leq \widehat{\text{CSV}}(f_{\boldsymbol{\theta}}) + \mathcal{O}\left(\frac{N\left(\mathcal{F}(\Theta), \epsilon, \|\cdot\|_{L_\infty}\right) + \log(1/\delta)}{\sqrt{n}}\right). \tag{60}$$

That completes our proof. $\qquad\square$

### C.3 Proofs in Section 4.3

In this section we aim at proving that the (9) is a sharp estimator to the CSV when we know a lower bound $c$ for $\pi_{kz}$. Note that our problem is equivalent to estimate $\sup_{P_k} \mathbb{E}_{P_k}[V] - \inf_{P_k} \mathbb{E}_{P_k}[V]$ via $n$ samples $\{V_i\}$ drawn from mixture distribution $P = \sum_{k\in[K]} \pi_k P_k$ with known $\pi_k > c$. W.o.l.g., suppose $\mathbb{E}_{P_1}[V] \leq \mathbb{E}_{P_2}[V] \leq, \dots, \leq \mathbb{E}_{P_K}[V]$, then we show $\mathbb{E}_P[V \mid V \geq q_P(1-c)] - \mathbb{E}_P[V \mid V \leq q_P(c)]$ is a upper bound to $\mathbb{E}_{P_K}[V] - \mathbb{E}_{P_1}[V]$ and the bound is sharp when $K \geq 3$.

**Proposition 3.** *For $k = 1, \cdots, K$ and $\pi_k \geq c$, suppose $P_k$ are absolutely continuous w.r.t. Lebesgue measure, then we have*

$$\mathbb{E}_P[V \mid V \geq q_P(1-c)] - \mathbb{E}_P[V \mid V \leq q_P(c)] \geq \mathbb{E}_{P_K}[V] - \mathbb{E}_{P_1}[V], \tag{61}$$

*and the equality can be taken form some $P_k$ and $\pi_k$ ($k = 1, \dots, K$) if $K \geq 3$.*

*Proof.* Let $p_k(v)$ be the density function of $P_k$. Then $p(\cdot) = \sum_{k=1}^{K} \pi_k p_k(\cdot)$ is the density function of $P$. One can verify that

$$(v - q_P(1-c))\left(\frac{p(v)}{c}\mathbf{1}_{\{v \geq q_P(1-c)\}} - p_K(v)\right) \geq 0 \tag{62}$$

for any $v$ since $p_2(v) \geq 0$ and $0 < c \leq \pi_K$. Thus taking integral w.r.t. $v$ we get

$$\mathbb{E}_P[V \mid V \geq q_P(1-c)] - \mathbb{E}_{P_K}[V] = \int_{\mathbb{R}} (v - q_P(1-c))\left(\frac{p(v)}{c}\mathbf{1}_{\{v \geq q_P(1-c)\}} - p_K(v)\right) dv \geq 0. \tag{63}$$

We can apply the similar argument to prove $\mathbb{E}_P[V \mid V \leq q_P(c)] \leq \mathbb{E}_{P_1}[V]$. Combining the two inequalities implies (61).

On the other hand, if $K \geq 3$, we take

$$\pi_1 = \pi_K = c;$$
$$\pi_2 =, \dots, = \pi_{K-1} = \frac{1-2c}{K-2}, \tag{64}$$

and

$$p_1(v) = \frac{p(v)}{c}\mathbf{1}_{\{v \leq q_P(c)\}};$$
$$p_K(v) = \frac{p(v)}{c}\mathbf{1}_{\{v \geq q_P(1-c)\}}; \tag{65}$$
$$p_2(v) =, \cdots, = p_{K-1}(V) = \frac{p(v)}{1-2c}\mathbf{1}_{\{q_P(c) \leq v \leq q_P(1-c)\}}.$$

Then it is easy to verify that

$$\mathbb{E}_{P_K}[V \mid V \geq q_P(1-c)] - \mathbb{E}_P[V \mid V \leq q_P(c)] = \mathbb{E}_{P_K}[V] - \mathbb{E}_{P_1}[V] \tag{66}$$

under this distribution. $\qquad\square$

According to this proposition, we can use the quintile conditional expectation to estimate the proposed CSV as we did in the main body of this paper.

## D  Solving the Proposed Minimax Problem (11)

In this section, we provide the convergence of the proposed Algorithm 1 to solve (11). We illustrate it in the regime of regularize training with $\widehat{\text{CSV}}(f_{\boldsymbol{\theta}})$ i.e., $\boldsymbol{F}^k(\boldsymbol{\theta})$ defined in Section 5. Then we have $m = K_z^2$ in this regime. Let us define

$$\Phi_\rho^k(\boldsymbol{\theta}) = R_{emp}(f_{\boldsymbol{\theta}}, P) + \lambda \max_{\boldsymbol{u}\in\Delta_{K_z^2}} \boldsymbol{u}^\top \boldsymbol{F}^k(\boldsymbol{\theta}) - \rho\lambda\boldsymbol{u}^\top \log\left(K_z^2\boldsymbol{u}\right) = \max_{\boldsymbol{u}\in\Delta_{K_z^2}} \phi_\rho^k(\boldsymbol{\theta}, \boldsymbol{u});$$

$$\hat{\Phi}_\rho^k(\boldsymbol{\theta}) = R_{emp}(f_{\boldsymbol{\theta}}, P) + \lambda \max_{\boldsymbol{u}\in\Delta_{K_z^2}} \boldsymbol{u}^\top \hat{\boldsymbol{F}}^k(\boldsymbol{\theta}) - \rho\lambda\boldsymbol{u}^\top \log\left(K_z^2\boldsymbol{u}\right) = \max_{\boldsymbol{u}\in\Delta_{K_z^2}} \hat{\phi}_\rho^k(\boldsymbol{\theta}, \boldsymbol{u}). \tag{67}$$

We have the following lemma to state the some continuities.

**Lemma 2.** *Under Assumption 1-2, we have the following conclusions*

*1. For $\phi_\rho^k(\boldsymbol{\theta}, \boldsymbol{u})$ with any $\rho$ and $k$ we have*

$$
\left\|\nabla_{\boldsymbol{\theta}}\phi_\rho^k(\boldsymbol{\theta}_1, \boldsymbol{u}) - \nabla_{\boldsymbol{\theta}}\phi_\rho^k(\boldsymbol{\theta}_2, \boldsymbol{u})\right\| \leq (1 + 2\lambda K_z L_1)\|\boldsymbol{\theta}_1 - \boldsymbol{\theta}_2\| = L_{11}\|\boldsymbol{\theta}_1 - \boldsymbol{\theta}_2\|;
$$
$$
\left\|\nabla_{\boldsymbol{\theta}}\phi_\rho^k(\boldsymbol{\theta}, \boldsymbol{u}_1) - \nabla_{\boldsymbol{\theta}}\phi_\rho^k(\boldsymbol{\theta}, \boldsymbol{u}_2)\right\| \leq 2\lambda K_z L_0\|\boldsymbol{u}_1 - \boldsymbol{u}_2\| = L_{12}\|\boldsymbol{u}_1 - \boldsymbol{u}_2\|; \tag{68}
$$
$$
\left\|\nabla_{\boldsymbol{u}}\phi_\rho^k(\boldsymbol{\theta}_1, \boldsymbol{u}) - \nabla_{\boldsymbol{u}}\phi_\rho^k(\boldsymbol{\theta}_2, \boldsymbol{u})\right\| \leq 2\lambda K_z L_0\|\boldsymbol{\theta}_1 - \boldsymbol{\theta}_2\| = L_{12}\|\boldsymbol{u}_1 - \boldsymbol{u}_2\|.
$$

*2. Let $\boldsymbol{u}_k^*(\boldsymbol{\theta}, \rho) = \arg\max_{\boldsymbol{u} \in \Delta_{K_z^2}} \phi_\rho^k(\boldsymbol{\theta}, \boldsymbol{u})$ and $\hat{\boldsymbol{u}}_k^*(\boldsymbol{\theta}, \rho) = \arg\max_{\boldsymbol{u} \in \Delta_{K_z^2}} \hat{\phi}_\rho^k(\boldsymbol{\theta}, \boldsymbol{u})$ then*

$$
\|\boldsymbol{u}_k^*(\boldsymbol{\theta}, \rho) - \hat{\boldsymbol{u}}_k^*(\boldsymbol{\theta}, \rho)\| \leq \frac{1}{\rho}\left\|\hat{\boldsymbol{F}}(\boldsymbol{\theta}) - \boldsymbol{F}(\boldsymbol{\theta})\right\|. \tag{69}
$$

*3. $\Phi_\rho^k(\boldsymbol{\theta})$ is $L_1 + \lambda\left(L_{11} + L_{12}^2/\rho\right)$-smoothness*

*Proof.* Let us proof the conclusions by order. For the first conclusion, we have

$$
\nabla_{\boldsymbol{\theta}}\phi_\rho^k(\boldsymbol{\theta}, \boldsymbol{u}) = \nabla_{\boldsymbol{\theta}}R_{emp}(f_{\boldsymbol{\theta}}, P) + \lambda\nabla_{\boldsymbol{\theta}}\boldsymbol{F}^k(\boldsymbol{\theta})^\top\boldsymbol{u}; \qquad \nabla_{\boldsymbol{u}}\phi_\rho^k(\boldsymbol{\theta}, \boldsymbol{u}) = \lambda\boldsymbol{F}^k(\boldsymbol{\theta}) - \rho\lambda\left(\log\left(K_z^2\boldsymbol{u}\right) + \mathbf{e}\right), \tag{70}
$$

where $\mathbf{e} = (1, \cdots, 1)$. Thus, by Schwarz's inequality, one can verify

$$
\left\|\nabla_{\boldsymbol{\theta}}\phi_\rho^k(\boldsymbol{\theta}_1, \boldsymbol{u}) - \nabla_{\boldsymbol{\theta}}\phi_\rho^k(\boldsymbol{\theta}_2, \boldsymbol{u})\right\| \leq \|\nabla_{\boldsymbol{\theta}}R_{emp}(f_{\boldsymbol{\theta}_1}, P) - \nabla_{\boldsymbol{\theta}}R_{emp}(f_{\boldsymbol{\theta}_2}, P)\|
$$
$$
+ \lambda\left\|\boldsymbol{u}^\top\left(\nabla_{\boldsymbol{\theta}}\boldsymbol{F}^k(\boldsymbol{\theta}_1) - \nabla_{\boldsymbol{\theta}}\boldsymbol{F}^k(\boldsymbol{\theta}_2)\right)\right\|
$$
$$
\leq \lambda\left(\sum_{i\in[K_z^2], j\in[K_z^2]}\boldsymbol{u}(i)\boldsymbol{u}(j)\sup_{(z_1, z_2)}\left\|\left(\nabla\hat{\mathcal{L}}_{kz_1}(f_{\boldsymbol{\theta}_1}) - \nabla\hat{\mathcal{L}}_{kz_2}(f_{\boldsymbol{\theta}_1})\right) - \left(\nabla\hat{\mathcal{L}}_{kz_1}(f_{\boldsymbol{\theta}_2}) - \nabla\hat{\mathcal{L}}_{kz_2}(f_{\boldsymbol{\theta}_2})\right)\right\|^2\right)^{\frac{1}{2}}
$$
$$
+ L_1\|\boldsymbol{\theta}_1 - \boldsymbol{\theta}_2\|
$$
$$
\leq (1 + 2\lambda K_z)L_1\|\boldsymbol{\theta}_1 - \boldsymbol{\theta}_2\|, \tag{71}
$$

and

$$
\left\|\nabla_{\boldsymbol{\theta}}\phi_\rho^k(\boldsymbol{\theta}, \boldsymbol{u}_1) - \nabla_{\boldsymbol{\theta}}\phi_\rho^k(\boldsymbol{\theta}, \boldsymbol{u}_2)\right\| \leq \lambda\|\boldsymbol{u}_1 - \boldsymbol{u}_2\|\left\|\nabla_{\boldsymbol{\theta}}\boldsymbol{F}^k(\boldsymbol{\theta})\right\| \leq 2\lambda K_z L_0\|\boldsymbol{u}_1 - \boldsymbol{u}_2\|, \tag{72}
$$

and

$$
\left\|\nabla_{\boldsymbol{u}}\phi_\rho^k(\boldsymbol{\theta}_1, \boldsymbol{u}) - \nabla_{\boldsymbol{u}}\phi_\rho^k(\boldsymbol{\theta}_2, \boldsymbol{u})\right\| \leq \lambda\left\|\boldsymbol{F}^k(\boldsymbol{\theta}_1) - \boldsymbol{F}^k(\boldsymbol{\theta}_2)\right\| \leq 2\lambda K_z L_0\|\boldsymbol{\theta}_1 - \boldsymbol{\theta}_2\|. \tag{73}
$$

Thus we complete the proof to the first conclusion.

For the second conclusion, by the Lagrange's multiplier method or Theorem in (Yi et al., 2021a), we have the unique closed-form solution of $\boldsymbol{u}_k^*(\boldsymbol{\theta}, \rho) \in \Delta_{K_z^2}$ that

$$
\boldsymbol{u}_k^*(\boldsymbol{\theta}, \rho) = \frac{\exp\left(\frac{1}{\rho}\boldsymbol{F}^k(\boldsymbol{\theta})(j)\right)}{\sum_{j=1}^{K_z^2}\exp\left(\frac{1}{\rho}\boldsymbol{F}^k(\boldsymbol{\theta})(j)\right)} = \text{Softmax}\left(\frac{\boldsymbol{F}^k(\boldsymbol{\theta})}{\rho}\right);
$$
$$
\tag{74}
$$
$$
\hat{\boldsymbol{u}}_k^*(\boldsymbol{\theta}, \rho) = \frac{\exp\left(\frac{1}{\rho}\boldsymbol{F}^k(\boldsymbol{\theta})(j)\right)}{\sum_{j=1}^{K_z^2}\exp\left(\frac{1}{\rho}\hat{\boldsymbol{F}}^k(\boldsymbol{\theta})(j)\right)} = \text{Softmax}\left(\frac{\hat{\boldsymbol{F}}^k(\boldsymbol{\theta})}{\rho}\right).
$$

On the other hand, due to $\boldsymbol{u} \in \Delta_{K_z^2}$ we have

$$
\nabla_{\boldsymbol{uu}}^2\phi_\rho^k(\boldsymbol{\theta}, \boldsymbol{u}) = -\rho\lambda\text{diag}\left(\frac{1}{\boldsymbol{u}(i)}, \cdots, \frac{1}{\boldsymbol{u}(K_z^2)}\right) \preceq -\rho\lambda\boldsymbol{I}, \tag{75}
$$

where $\boldsymbol{A} \succeq \boldsymbol{B}$ means that $\boldsymbol{A} - \boldsymbol{B}$ is a semi-positive definite matrix and $\boldsymbol{I}$ is the identity matrix. The similar conclusion holds for $\hat{\phi}_\rho^k(\boldsymbol{\theta}, \boldsymbol{u})$. Thus both $\phi_\rho^k(\boldsymbol{\theta}, \boldsymbol{u})$ and $\hat{\phi}_\rho^k(\boldsymbol{\theta}, \boldsymbol{u})$ are $\rho$-strongly concave w.r.t.

$\boldsymbol{u}$. Then

$$
\begin{aligned}
\phi_\rho^k(\boldsymbol{\theta}, \boldsymbol{u}_k^*(\boldsymbol{\theta}, \rho)) &\le \phi_\rho^k(\boldsymbol{\theta}, \hat{\boldsymbol{u}}_k^*(\boldsymbol{\theta}, \rho)) + \left\langle \nabla_{\boldsymbol{u}} \phi_\rho^k(\boldsymbol{\theta}, \hat{\boldsymbol{u}}_k^*(\boldsymbol{\theta}, \rho)), \boldsymbol{u}_k^*(\boldsymbol{\theta}, \rho) - \hat{\boldsymbol{u}}_k^*(\boldsymbol{\theta}, \rho) \right\rangle \\
&\quad - \frac{\rho\lambda}{2} \left\| \boldsymbol{u}_k^*(\boldsymbol{\theta}, \rho) - \hat{\boldsymbol{u}}_k^*(\boldsymbol{\theta}, \rho) \right\|^2 ; \\
\phi_\rho^k(\boldsymbol{\theta}, \hat{\boldsymbol{u}}_k^*(\boldsymbol{\theta}, \rho)) &\le \phi_\rho^k(\boldsymbol{\theta}, \boldsymbol{u}_k^*(\boldsymbol{\theta}, \rho)) + \left\langle \nabla_{\boldsymbol{u}} \phi_\rho^k(\boldsymbol{\theta}, \boldsymbol{u}_k^*(\boldsymbol{\theta}, \rho)), \hat{\boldsymbol{u}}_k^*(\boldsymbol{\theta}, \rho) - \boldsymbol{u}_k^*(\boldsymbol{\theta}, \rho) \right\rangle \\
&\quad - \frac{\rho\lambda}{2} \left\| \boldsymbol{u}_k^*(\boldsymbol{\theta}, \rho) - \hat{\boldsymbol{u}}_k^*(\boldsymbol{\theta}, \rho) \right\|^2 .
\end{aligned}
\tag{76}
$$

Plugging the two above inequalities, we have that

$$
\left\langle \nabla_{\boldsymbol{u}} \phi_\rho^k(\boldsymbol{\theta}, \hat{\boldsymbol{u}}_k^*(\boldsymbol{\theta}, \rho)), \boldsymbol{u}_k^*(\boldsymbol{\theta}, \rho) - \hat{\boldsymbol{u}}_k^*(\boldsymbol{\theta}, \rho) \right\rangle \ge \rho\lambda \left\| \boldsymbol{u}_k^*(\boldsymbol{\theta}, \rho) - \hat{\boldsymbol{u}}_k^*(\boldsymbol{\theta}, \rho) \right\|^2
\tag{77}
$$

due to the $\left\langle \nabla_{\boldsymbol{u}} \phi_\rho^k(\boldsymbol{\theta}, \hat{\boldsymbol{u}}_k^*(\boldsymbol{\theta}, \rho)), \boldsymbol{u}_k^*(\boldsymbol{\theta}, \rho) - \hat{\boldsymbol{u}}_k^*(\boldsymbol{\theta}, \rho) \right\rangle \le 0$. On the other hand, as

$$
\left\langle \nabla_{\boldsymbol{u}} \hat{\phi}_\rho^k(\boldsymbol{\theta}, \hat{\boldsymbol{u}}_k^*(\boldsymbol{\theta}, \rho)), \boldsymbol{u}_k^*(\boldsymbol{\theta}, \rho) - \hat{\boldsymbol{u}}_k^*(\boldsymbol{\theta}, \rho) \right\rangle \le 0.
\tag{78}
$$

Plugging this into the above inequality, we get

$$
\begin{aligned}
\rho\lambda \left\| \boldsymbol{u}_k^*(\boldsymbol{\theta}, \rho) - \hat{\boldsymbol{u}}_k^*(\boldsymbol{\theta}, \rho) \right\|^2 &\le \left\langle \nabla_{\boldsymbol{u}} \phi_\rho^k(\boldsymbol{\theta}, \hat{\boldsymbol{u}}_k^*(\boldsymbol{\theta}, \rho)) - \nabla_{\boldsymbol{u}} \hat{\phi}_\rho^k(\boldsymbol{\theta}, \hat{\boldsymbol{u}}_k^*(\boldsymbol{\theta}, \rho)), \boldsymbol{u}_k^*(\boldsymbol{\theta}, \rho) - \hat{\boldsymbol{u}}_k^*(\boldsymbol{\theta}, \rho) \right\rangle \\
&= \lambda \left\langle \boldsymbol{F}^k(\boldsymbol{\theta}) - \hat{\boldsymbol{F}}^k(\boldsymbol{\theta}), \boldsymbol{u}_k^*(\boldsymbol{\theta}, \rho) - \hat{\boldsymbol{u}}_k^*(\boldsymbol{\theta}, \rho) \right\rangle \\
&\le \lambda \left\| \boldsymbol{F}^k(\boldsymbol{\theta}) - \hat{\boldsymbol{F}}^k(\boldsymbol{\theta}) \right\| \left\| \boldsymbol{u}_k^*(\boldsymbol{\theta}, \rho) - \hat{\boldsymbol{u}}_k^*(\boldsymbol{\theta}, \rho) \right\| .
\end{aligned}
\tag{79}
$$

Thus the conclusion is proofed.

Finally, we prove the third conclusion. Similar to the proof of the second conclusion, we have

$$
\left\| \boldsymbol{u}_k^*(\boldsymbol{\theta}_1, \rho) - \boldsymbol{u}_k^*(\boldsymbol{\theta}_2, \rho) \right\| \le \frac{1}{\rho} \| \boldsymbol{\theta}_1 - \boldsymbol{\theta}_2 \|.
\tag{80}
$$

Since $\Delta_{K_z^2}$ is convex, bounded, and $\boldsymbol{u}_k^*(\boldsymbol{\theta}, \rho)$ is unique for any $\boldsymbol{\theta}$, by Danskin's Theorem (Bernhard & Rapaport, 1995), we have

$$
\begin{aligned}
\left\| \nabla \Phi_\rho^k(\boldsymbol{\theta}_1) - \nabla \Phi_\rho^k(\boldsymbol{\theta}_2) \right\| &\le \| \nabla_{\boldsymbol{\theta}} R_{emp}(f_{\boldsymbol{\theta}_1}, P) - \nabla_{\boldsymbol{\theta}} R_{emp}(f_{\boldsymbol{\theta}_2}, P) \| \\
&\quad + \lambda \left\| \nabla_{\boldsymbol{\theta}} \boldsymbol{F}^k(\boldsymbol{\theta}_1)^\top \boldsymbol{u}_k^*(\boldsymbol{\theta}_1, \rho) - \nabla_{\boldsymbol{\theta}} \boldsymbol{F}^k(\boldsymbol{\theta}_2)^\top \boldsymbol{u}_k^*(\boldsymbol{\theta}_2, \rho) \right\| \\
&\le L_1 \| \boldsymbol{\theta}_1 - \boldsymbol{\theta}_2 \| + \lambda \left\| \nabla_{\boldsymbol{\theta}} \boldsymbol{F}^k(\boldsymbol{\theta}_1)^\top (\boldsymbol{u}_k^*(\boldsymbol{\theta}_1, \rho) - \boldsymbol{u}_k^*(\boldsymbol{\theta}_2, \rho)) \right\| \\
&\quad + \lambda \left\| \boldsymbol{u}_k^*(\boldsymbol{\theta}_2, \rho)^\top \left( \nabla_{\boldsymbol{\theta}} \boldsymbol{F}^k(\boldsymbol{\theta}_1) - \nabla_{\boldsymbol{\theta}} \boldsymbol{F}^k(\boldsymbol{\theta}_2) \right) \right\| \\
&\le L_1 \| \boldsymbol{\theta}_1 - \boldsymbol{\theta}_2 \| + \frac{\lambda L_{12}^2}{\rho} \| \boldsymbol{\theta}_1 - \boldsymbol{\theta}_2 \| + \lambda L_{11} \| \boldsymbol{\theta}_1 - \boldsymbol{\theta}_2 \| \\
&= \left( L_1 + \frac{\lambda L_{12}^2}{\rho} + \lambda L_{11} \right) \| \boldsymbol{\theta}_1 - \boldsymbol{\theta}_2 \| ,
\end{aligned}
\tag{81}
$$

which implies our conclusion. $\qquad \square$

We present the following lemma to state the descent property of the obtained iterates via Algorithm 1. Let us define

$$
\hat{\phi}^k(\boldsymbol{\theta}, \boldsymbol{u}) = R_{emp}(f_{\boldsymbol{\theta}}, P) + \lambda \boldsymbol{u}^\top \boldsymbol{F}^k
\tag{82}
$$

As we have assume that unbiased estimators $\hat{R}_{emp}(f_{\boldsymbol{\theta}}, P)$ and $\hat{\boldsymbol{F}}^k(\boldsymbol{\theta})$ have bounded variance, then according to

$$
\begin{aligned}
\mathbb{E}\left[ \left( \hat{\phi}^k(\boldsymbol{\theta}, \boldsymbol{u}) - \phi^k(\boldsymbol{\theta}, \boldsymbol{u}) \right)^2 \right] &\le 2\mathbb{E}\left[ \left( \hat{R}_{emp}(f_{\boldsymbol{\theta}}, P) - R_{emp}(f_{\boldsymbol{\theta}}, P)^2 \right) \right] \\
&\quad + 2\lambda^2 \mathbb{E}\left[ \left\| \hat{\boldsymbol{F}}^k(\boldsymbol{\theta}) - \boldsymbol{F}^k(\boldsymbol{\theta}) \right\|^2 \right] ,
\end{aligned}
\tag{83}
$$

w.o.l.g. we assume that

$$
\max \left\{ \mathbb{E}\left[ \left( \hat{\phi}^k(\boldsymbol{\theta}, \boldsymbol{u}) - \phi^k(\boldsymbol{\theta}, \boldsymbol{u}) \right)^2 \right], \mathbb{E}\left[ \left\| \hat{\boldsymbol{F}}^k(\boldsymbol{\theta}) - \boldsymbol{F}^k(\boldsymbol{\theta}) \right\|^2 \right] \right\} \le \sigma^2.
\tag{84}
$$

**Lemma 3.** *Let $L_1 + \lambda\left(L_{11} + L_{12}^2/\rho\right) = \tilde{L}$, if we have the estimation such that $\mathbb{E}[\hat{\phi}^k(\boldsymbol{\theta}, \boldsymbol{u})] = \phi^k(\boldsymbol{\theta}, \boldsymbol{u})$, $\mathbb{E}[(\hat{\phi}^k(\boldsymbol{\theta}, \boldsymbol{u}) - \phi^k(\boldsymbol{\theta}, \boldsymbol{u}))^2] \leq \sigma^2$ then*

$$
\begin{aligned}
\mathbb{E}\left[\sum_{k=1}^{K_y} \hat{p}_k \Phi_\rho^k(\boldsymbol{\theta}(t+1))\right] &\leq \mathbb{E}\left[\sum_{k=1}^{K_y} \hat{p}_k \Phi_\rho^k(\boldsymbol{\theta}(t))\right] - \left(\frac{\eta_{\boldsymbol{\theta}}}{2} - \tilde{L}\eta_{\boldsymbol{\theta}}^2\right) \mathbb{E}\left[\left\|\sum_{k=1}^{K_y} \hat{p}_k \nabla \Phi_\rho^k(\boldsymbol{\theta}(t))\right\|^2\right] \\
&\quad + \sum_{k=1}^{K_y} \left(\frac{\tilde{L}K_y L_{12}^2 \hat{p}_k \eta_{\boldsymbol{\theta}}^2}{\rho} + \frac{(\sigma^2 + L_{11})L_{12}^2 \eta_{\boldsymbol{\theta}}}{2\rho^2}\right) \hat{p}_k \mathbb{E}\left[\left\|\boldsymbol{F}^k(\boldsymbol{\theta}(t)) - \boldsymbol{F}_t^k\right\|^2\right] \\
&\quad + \frac{\tilde{L}\eta_{\boldsymbol{\theta}}^2 \sigma^2}{2}.
\end{aligned}
\tag{85}
$$

*Proof.* Due to the $\tilde{L}$-smoothness of $\Phi_\rho^k(\cdot)$ for any $k$ and $\rho$ we have

$$
\begin{aligned}
\sum_{k=1}^{K_y} \hat{p}_k \Phi_\rho^k(\boldsymbol{\theta}(t+1)) &\leq \sum_{k=1}^{K_y} \hat{p}_k \Phi_\rho^k(\boldsymbol{\theta}(t)) + \left\langle \sum_{k=1}^{K_y} \hat{p}_k \nabla \Phi_\rho^k(\boldsymbol{\theta}(t)), \boldsymbol{\theta}(t+1) - \boldsymbol{\theta}(t)\right\rangle + \frac{\tilde{L}}{2}\|\boldsymbol{\theta}(t+1) - \boldsymbol{\theta}(t)\|^2 \\
&= \sum_{k=1}^{K_y} \hat{p}_k \Phi_\rho^k(\boldsymbol{\theta}(t)) - \eta_{\boldsymbol{\theta}} \left\langle \sum_{k=1}^{K_y} \hat{p}_k \nabla \Phi_\rho^k(\boldsymbol{\theta}(t)), \sum_{k=1}^{K_y} \hat{p}_k \nabla_{\boldsymbol{\theta}} \hat{\phi}_\rho^k(\boldsymbol{\theta}(t), \hat{\boldsymbol{u}}_k^*(\boldsymbol{\theta}(t), \rho))\right\rangle \\
&\quad + \frac{\tilde{L}\eta_{\boldsymbol{\theta}}^2}{2} \left\|\sum_{k=1}^{K_y} \hat{p}_k \nabla_{\boldsymbol{\theta}} \hat{\phi}_\rho^k(\boldsymbol{\theta}(t), \hat{\boldsymbol{u}}_k^*(\boldsymbol{\theta}(t), \rho))\right\|^2 \\
&\leq \sum_{k=1}^{K_y} \hat{p}_k \Phi_\rho^k(\boldsymbol{\theta}(t)) - \eta_{\boldsymbol{\theta}} \left\|\sum_{k=1}^{K_y} \hat{p}_k \nabla \Phi_\rho^k(\boldsymbol{\theta}(t))\right\|^2 \\
&\quad + \eta_{\boldsymbol{\theta}} \left\langle \sum_{k=1}^{K_y} \hat{p}_k \nabla \Phi_\rho^k(\boldsymbol{\theta}(t)), \sum_{k=1}^{K_y} \hat{p}_k \left(\nabla \Phi_\rho^k(\boldsymbol{\theta}(t)) - \nabla_{\boldsymbol{\theta}} \hat{\phi}_\rho^k(\boldsymbol{\theta}(t), \boldsymbol{u}_k^*(\boldsymbol{\theta}(t), \rho))\right)\right\rangle \\
&\quad + \eta_{\boldsymbol{\theta}} \left\langle \sum_{k=1}^{K_y} \hat{p}_k \nabla \Phi_\rho^k(\boldsymbol{\theta}(t)), \sum_{k=1}^{K_y} \hat{p}_k \left(\nabla_{\boldsymbol{\theta}} \hat{\phi}_\rho^k(\boldsymbol{\theta}(t), \boldsymbol{u}_k^*(\boldsymbol{\theta}(t), \rho)) - \nabla_{\boldsymbol{\theta}} \hat{\phi}_\rho^k(\boldsymbol{\theta}(t), \boldsymbol{u}_k(t))\right)\right\rangle \\
&\quad + \frac{\tilde{L}\eta_{\boldsymbol{\theta}}^2}{2} \left\|\sum_{k=1}^{K_y} \hat{p}_k \nabla_{\boldsymbol{\theta}} \hat{\phi}_\rho^k(\boldsymbol{\theta}(t), \boldsymbol{u}_k(t))\right\|^2.
\end{aligned}
\tag{86}
$$

On the other hand, using the fact that $\mathbb{E}\left[\hat{\phi}_\rho^k(\boldsymbol{\theta}, \boldsymbol{u})\right] = \phi_\rho^k(\boldsymbol{\theta}, \boldsymbol{u})$, $\mathbb{E}\left[\left(\hat{\phi}_\rho^k(\boldsymbol{\theta}, \boldsymbol{u}) - \phi_\rho^k(\boldsymbol{\theta}, \boldsymbol{u})\right)^2\right] \leq \sigma^2$, and Young's inequality

$$
\begin{aligned}
&\mathbb{E}\left[\left\|\sum_{k=1}^{K_y} \hat{p}_k \nabla_{\boldsymbol{\theta}} \hat{\phi}_\rho^k(\boldsymbol{\theta}(t), \boldsymbol{u}_k(t))\right\|^2\right] \\
&= \mathbb{E}\left[\left\|\sum_{k=1}^{K_y} \hat{p}_k \left(\nabla_{\boldsymbol{\theta}} \hat{\phi}_\rho^k(\boldsymbol{\theta}(t), \boldsymbol{u}_k(t)) - \nabla_{\boldsymbol{\theta}} \phi_\rho^k(\boldsymbol{\theta}(t), \boldsymbol{u}_k(t))\right) + \sum_{k=1}^{K_y} \hat{p}_k \nabla_{\boldsymbol{\theta}} \phi_\rho^k(\boldsymbol{\theta}(t), \boldsymbol{u}_k(t))\right\|^2\right] \\
&\leq \sigma^2 + \mathbb{E}\left[\left\|\sum_{k=1}^{K_y} \hat{p}_k \nabla_{\boldsymbol{\theta}} \phi_\rho^k(\boldsymbol{\theta}(t), \boldsymbol{u}_k(t))\right\|^2\right] \\
&\leq \sigma^2 + 2\mathbb{E}\left[\left\|\sum_{k=1}^{K_y} \hat{p}_k \nabla \Phi_\rho^k(\boldsymbol{\theta}(t))\right\|^2\right] + 2\mathbb{E}\left[\left\|\sum_{k=1}^{K_y} \hat{p}_k \left(\nabla \Phi_\rho^k(\boldsymbol{\theta}(t)) - \nabla_{\boldsymbol{\theta}} \phi_\rho^k(\boldsymbol{\theta}(t), \boldsymbol{u}_k(t))\right)\right\|^2\right].
\end{aligned}
\tag{87}
$$

Due to Danskin's theorem and (i), (ii) in Lemma 2 we have

$$
\mathbb{E}\left[\left\|\sum_{k=1}^{K_y}\hat{p}_k\left(\nabla\Phi_\rho^k(\boldsymbol{\theta}(t))-\nabla_{\boldsymbol{\theta}}\phi_\rho^k(\boldsymbol{\theta}(t),\boldsymbol{u}_k(t))\right)\right\|^2\right]\le\mathbb{E}\left[\left(\sum_{k=1}^{K_y}\hat{p}_kL_{12}\left\|\boldsymbol{u}_k^*(\boldsymbol{\theta}(t),\rho)-\boldsymbol{u}_k(t)\right\|\right)^2\right]
$$

$$
\le\mathbb{E}\left[\left(\sum_{k=1}^{K_y}\frac{\hat{p}_kL_{12}}{\rho}\left\|\boldsymbol{F}^k(\boldsymbol{\theta}(t))-\boldsymbol{F}_t^k\right\|\right)^2\right]\tag{88}
$$

$$
\le\frac{K_yL_{12}^2}{\rho^2}\sum_{k=1}^{K_y}\hat{p}_k^2\mathbb{E}\left[\left\|\boldsymbol{F}^k(\boldsymbol{\theta}(t))-\boldsymbol{F}_t^k\right\|^2\right].
$$

Finally, we have

$$
\mathbb{E}\left[\left\langle\sum_{k=1}^{K_y}\hat{p}_k\nabla\Phi_\rho^k(\boldsymbol{\theta}(t)),\sum_{k=1}^{K_y}\hat{p}_k\left(\nabla_{\boldsymbol{\theta}}\hat{\phi}_\rho^k(\boldsymbol{\theta}(t),\boldsymbol{u}_k^*(\boldsymbol{\theta}(t),\rho))-\nabla_{\boldsymbol{\theta}}\hat{\phi}_\rho^k(\boldsymbol{\theta}(t),\boldsymbol{u}_k(t))\right)\right\rangle\right]
$$

$$
=\sum_{k=1}^{K_y}\hat{p}_k\mathbb{E}\left[\left\langle\sum_{k=1}^{K_y}\hat{p}_k\nabla\Phi_\rho^k(\boldsymbol{\theta}(t)),\left(\nabla_{\boldsymbol{\theta}}\hat{\phi}_\rho^k(\boldsymbol{\theta}(t),\boldsymbol{u}_k^*(\boldsymbol{\theta}(t),\rho))-\nabla_{\boldsymbol{\theta}}\hat{\phi}_\rho^k(\boldsymbol{\theta}(t),\boldsymbol{u}_k(t))\right)\right\rangle\right]
$$

$$
\le\frac{1}{2}\mathbb{E}\left[\left\|\sum_{k=1}^{K_y}\hat{p}_k\nabla\Phi_\rho^k(\boldsymbol{\theta}(t))\right\|^2\right]+\frac{1}{2}\sum_{k=1}^{K_y}\hat{p}_k\mathbb{E}\left[\sum_{k=1}^{K_y}\left\|\nabla_{\boldsymbol{\theta}}\hat{\phi}_\rho^k(\boldsymbol{\theta}(t),\boldsymbol{u}_k^*(\boldsymbol{\theta}(t),\rho))-\nabla_{\boldsymbol{\theta}}\hat{\phi}_\rho^k(\boldsymbol{\theta}(t),\boldsymbol{u}_k(t))\right\|^2\right]
$$

$$
\le\frac{1}{2}\mathbb{E}\left[\left\|\sum_{k=1}^{K_y}\hat{p}_k\nabla\Phi_\rho^k(\boldsymbol{\theta}(t))\right\|^2\right]+\frac{1}{2}\sum_{k=1}^{K_y}\hat{p}_k\mathbb{E}\left[\left\|\nabla\hat{\boldsymbol{F}}^k(\boldsymbol{\theta}(t))\right\|^2\left\|\boldsymbol{u}_k(t)-\boldsymbol{u}_k^*(\boldsymbol{\theta}(t),\rho)\right\|^2\right]
$$

$$
\le\frac{1}{2}\mathbb{E}\left[\left\|\sum_{k=1}^{K_y}\hat{p}_k\nabla\Phi_\rho^k(\boldsymbol{\theta}(t))\right\|^2\right]+\frac{\left(\sigma^2+L_{11}\right)L_{12}^2}{2\rho^2}\sum_{k=1}^{K_y}\hat{p}_k\mathbb{E}\left[\left\|\boldsymbol{F}^k(\boldsymbol{\theta}(t))-\boldsymbol{F}_t^k\right\|^2\right].
$$

$$
\tag{89}
$$

Plugging the three above inequalities into (86) and taking expectation to the both sides of the equality, we get

$$
\mathbb{E}\left[\sum_{k=1}^{K_y}\hat{p}_k\Phi_\rho^k(\boldsymbol{\theta}(t+1))\right]\le\mathbb{E}\left[\sum_{k=1}^{K_y}\hat{p}_k\Phi_\rho^k(\boldsymbol{\theta}(t))\right]-\left(\frac{\eta_{\boldsymbol{\theta}}}{2}-\tilde{L}\eta_{\boldsymbol{\theta}}^2\right)\mathbb{E}\left[\left\|\sum_{k=1}^{K_y}\hat{p}_k\nabla\Phi_\rho^k(\boldsymbol{\theta}(t))\right\|^2\right]
$$

$$
+\sum_{k=1}^{K_y}\left(\frac{\tilde{L}K_yL_{12}^2\hat{p}_k\eta_{\boldsymbol{\theta}}^2}{\rho}+\frac{\left(\sigma^2+L_{11}\right)L_{12}^2\eta_{\boldsymbol{\theta}}}{2\rho^2}\right)\hat{p}_k\mathbb{E}\left[\left\|\boldsymbol{F}^k(\boldsymbol{\theta}(t))-\boldsymbol{F}_t^k\right\|^2\right]+\frac{\tilde{L}\eta_{\boldsymbol{\theta}}^2\sigma^2}{2}.
$$

$$
\tag{90}
$$

This completes the proof of our theorem. $\qquad\square$

Then we proceed to the next lemma to characterize the dynamic of $\mathbb{E}\left[\left\|\boldsymbol{F}^k(\boldsymbol{\theta}(t))-\boldsymbol{F}_t^k\right\|^2\right]$.

**Lemma 4.** *For the $\boldsymbol{F}_t^k$ defined in Algorithm 1, and let $\delta^k(t)=\mathbb{E}\left[\left\|\boldsymbol{F}^k(\boldsymbol{\theta}(t))-\boldsymbol{F}_t^k\right\|^2\right]$, and $\delta(t)=\sum_{k=1}^{K_y}\hat{p}_k\delta^k(t)$, by choosing $\gamma\le2/3$, we have*

$$
\delta(t+1)\le\left(1-\frac{\gamma}{2}+\frac{4\eta_{\boldsymbol{\theta}}^2L_{11}^2K_yL_{12}^2}{\gamma\rho^2}\right)\delta(t)+\left(2\gamma^2+\frac{2\eta_{\boldsymbol{\theta}}^2L_{11}^2}{\gamma}\right)\sigma^2
$$

$$
+\frac{4\eta_{\boldsymbol{\theta}}^2L_{11}^2}{\gamma}\mathbb{E}\left[\left\|\sum_{k=1}^{K_y}\hat{p}_k\nabla\Phi_\rho^k(\boldsymbol{\theta}(t))\right\|^2\right].
$$

$$
\tag{91}
$$

*Proof.* W.o.l.g., we fix the $k$ during our proof. According to $\mathbb{E}\left[\hat{\boldsymbol{F}}^k(\boldsymbol{\theta}(t))\right]=\boldsymbol{F}^k(\boldsymbol{\theta}(t))$ and $\mathbb{E}\left[\left\|\hat{\boldsymbol{F}}^k(\boldsymbol{\theta}(t))-\boldsymbol{F}^k(\boldsymbol{\theta}(t))\right\|^2\right]\le\sigma^2$ we have

$$
\mathbb{E}\left[\left\|\boldsymbol{F}_{t+1}^k-\boldsymbol{F}^k(\boldsymbol{\theta}(t))\right\|^2\right]\le(1-\gamma)^2\delta^k(t)+\gamma^2\sigma^2\le(1-\gamma)\delta^k(t)+\gamma^2\sigma^2,\tag{92}
$$

where we use the fact $\gamma < 1$. Then due to the update rule of $\boldsymbol{F}_t^k$, Young's inequality and the above inequality,

$$
\begin{aligned}
\delta^k(t+1) &= \mathbb{E}\left[\left\|\boldsymbol{F}_{t+1}^k - \boldsymbol{F}^k(\boldsymbol{\theta}(t)) + \boldsymbol{F}^k(\boldsymbol{\theta}(t)) - \boldsymbol{F}^k(\boldsymbol{\theta}(t+1))\right\|^2\right] \\
&\leq \left(1 + \frac{\gamma}{2-2\gamma}\right)\mathbb{E}\left[\left\|\boldsymbol{F}_{t+1}^k - \boldsymbol{F}^k(\boldsymbol{\theta}(t))\right\|^2\right] + \left(1 + \frac{2-2\gamma}{\gamma}\right)\mathbb{E}\left[\left\|\boldsymbol{F}^k(\boldsymbol{\theta}(t+1)) - \boldsymbol{F}^k(\boldsymbol{\theta}(t))\right\|^2\right] \quad (93) \\
&\leq \left(1 - \frac{\gamma}{2}\right)\delta^k(t) + 2\gamma^2\sigma^2 + \frac{2}{\gamma}\mathbb{E}\left[\left\|\boldsymbol{F}^k(\boldsymbol{\theta}(t+1)) - \boldsymbol{F}^k(\boldsymbol{\theta}(t))\right\|^2\right].
\end{aligned}
$$

On the other hand, due to the $L_{11}$-continuity of $\boldsymbol{F}^k(\cdot)$ and (87), (88) we see

$$
\begin{aligned}
\mathbb{E}\left[\left\|\boldsymbol{F}^k(\boldsymbol{\theta}(t+1)) - \boldsymbol{F}^k(\boldsymbol{\theta}(t))\right\|^2\right] &\leq L_{11}^2\mathbb{E}\left[\|\boldsymbol{\theta}(t+1) - \boldsymbol{\theta}(t)\|^2\right] \\
&= \eta_{\boldsymbol{\theta}}^2 L_{11}^2\mathbb{E}\left[\left\|\sum_{k=1}^{K_y}\hat{p}_k\nabla_{\boldsymbol{\theta}}\hat{\phi}_\rho^k(\boldsymbol{\theta}(t),\boldsymbol{u}_k(t))\right\|^2\right] \\
&\leq \eta_{\boldsymbol{\theta}}^2 L_{11}^2\sigma^2 + 2\eta_{\boldsymbol{\theta}}^2 L_{11}^2\mathbb{E}\left[\left\|\sum_{k=1}^{K_y}\hat{p}_k\nabla\Phi_\rho^k(\boldsymbol{\theta}(t))\right\|^2\right] \\
&\quad + \frac{2\eta_{\boldsymbol{\theta}}^2 L_{11}^2 K_y L_{12}^2}{\rho^2}\sum_{k=1}^{K_y}\hat{p}_k^2\delta^k(t).
\end{aligned} \quad (94)
$$

Plugging this into the above inequality and weighted summing over $k$ (by $\hat{p}_k$) we get

$$
\begin{aligned}
\sum_{k=1}^{K_y}\hat{p}_k\delta^k(t+1) &\leq \left(1 - \frac{\gamma}{2} + \frac{4\eta_{\boldsymbol{\theta}}^2 L_{11}^2 K_y L_{12}^2}{\gamma\rho^2}\right)\sum_{k=1}^{K_y}\hat{p}_k\delta^k(t) + \left(2\gamma^2 + \frac{2\eta_{\boldsymbol{\theta}}^2 L_{11}^2}{\gamma}\right)\sigma^2 \\
&\quad + \frac{4\eta_{\boldsymbol{\theta}}^2 L_{11}^2}{\gamma}\mathbb{E}\left[\left\|\sum_{k=1}^{K_y}\hat{p}_k\nabla\Phi_\rho^k(\boldsymbol{\theta}(t))\right\|^2\right]
\end{aligned} \quad (95)
$$

which completes the our proof. $\qquad\square$

Now we are ready to state the convergence rate of the nonconvex-concave optimization problem.

**Theorem 5.** *Under Assumption 1 and 2, if $\hat{R}_{emp}(f_{\boldsymbol{\theta}}, P)$ and $\hat{\boldsymbol{F}}^k(\boldsymbol{\theta})$ are all unbiased estimators with bounded variance, $\boldsymbol{\theta}(t)$ is updated by Algorithm 1 with $\eta_{\boldsymbol{\theta}} = \mathcal{O}\left(T^{-\frac{3}{5}}\right)$ and $\gamma = T^{-\frac{2}{5}}$, then*

$$
\min_{1\leq t\leq T}\mathbb{E}\left[\left\|\sum_{k=1}^{K_y}\hat{p}_k\nabla\Phi_\rho^k(\boldsymbol{\theta}(t))\right\|^2\right] \leq \mathcal{O}\left(T^{-\frac{2}{5}}\right). \quad (14)
$$

*Besides that, for any $\boldsymbol{\theta}(t)$ and $\rho$, we have $|\sum_{k=1}^{K_y}\hat{p}_k(\Phi_\rho^k(\boldsymbol{\theta}(t)) - \Phi^k(\boldsymbol{\theta}(t)))| \leq \lambda\rho(1/me + 2\log m)$.*

*Proof.* First, note that $m = K_z^2$, for any $\boldsymbol{\theta}$,

$$
\Phi_\rho^k(\boldsymbol{\theta}) \leq M - \lambda\rho K_z\inf_x\{x\log(K_z^2 x)\} = M + \frac{\lambda\rho}{K_z e}. \quad (96)
$$

Due to the value of $\eta_{\boldsymbol{\theta}} \leq \frac{\gamma}{2\sqrt{6K_y L_{11}L_{12}}}$ we have that

$$
\left(1 - \frac{\gamma}{2} + \frac{4\eta_{\boldsymbol{\theta}}^2 L_{11}^2 K_y L_{12}^2}{\gamma\rho^2}\right) \leq 1 - \frac{\gamma}{3}. \quad (97)
$$

Thus we have

$$
\begin{aligned}
\delta(t) &\leq \left(1 - \frac{\gamma}{3}\right)^t 4M^2 + \left(2\gamma^2 + \frac{2\eta_{\boldsymbol{\theta}}^2 L_{11}^2}{\gamma}\right)\sigma^2\sum_{j=0}^{t-1}\left(1 - \frac{\gamma}{3}\right)^j \\
&\quad + \frac{4\eta_{\boldsymbol{\theta}}^2 L_{11}^2}{\gamma}\sum_{j=0}^{t-1}\left(1 - \frac{\gamma}{3}\right)^{t-j-1}\mathbb{E}\left[\left\|\sum_{k=1}^{K_y}\hat{p}_k\nabla\Phi_\rho^k(\boldsymbol{\theta}(j))\right\|^2\right]
\end{aligned} \quad (98)
$$

from Lemma 4. Plugging this into (85) in Lemma 3 and summing up it over $t = 0, \cdots, T$, we have

$$
\begin{aligned}
\mathbb{E}\left[\sum_{k=1}^{K_y} \hat{p}_k \Phi_\rho^k(\boldsymbol{\theta}(T))\right] &\leq \mathbb{E}\left[\sum_{k=1}^{K_y} \hat{p}_k \Phi_\rho^k(\boldsymbol{\theta}(0))\right] - \left(\frac{\eta_{\boldsymbol{\theta}}}{2} - \tilde{L}\eta_{\boldsymbol{\theta}}^2\right) \sum_{t=0}^{T-1} \mathbb{E}\left[\left\|\sum_{k=1}^{K_y} \hat{p}_k \nabla\Phi_\rho^k(\boldsymbol{\theta}(t))\right\|^2\right] \\
&+ 4M^2 \left(\frac{\tilde{L}K_y L_{12}^2 \eta_{\boldsymbol{\theta}}^2}{\rho} + \frac{(\sigma^2 + L_{11}) L_{12}^2 \eta_{\boldsymbol{\theta}}}{2\rho^2}\right) \sum_{t=0}^{T-1} \left(1 - \frac{\gamma}{3}\right)^t \\
&+ \left[\frac{T\tilde{L}\eta_{\boldsymbol{\theta}}^2}{2} + \left(\frac{\tilde{L}K_y L_{12}^2 \eta_{\boldsymbol{\theta}}^2}{\rho} + \frac{(\sigma^2 + L_{11}) L_{12}^2 \eta_{\boldsymbol{\theta}}}{2\rho^2}\right)\left(2\gamma^2 + \frac{2\eta_{\boldsymbol{\theta}}^2 L_{11}}{\gamma}\right) \sum_{t=0}^{T-1} \sum_{j=0}^{t-1} \left(1 - \frac{\gamma}{3}\right)^j\right] \sigma^2 \\
&+ \frac{4\eta_{\boldsymbol{\theta}}^2 L_{11}}{\gamma}\left(\frac{\tilde{L}K_y L_{12}^2 \eta_{\boldsymbol{\theta}}^2}{\rho} + \frac{(\sigma^2 + L_{11}) L_{12}^2 \eta_{\boldsymbol{\theta}}}{2\rho^2}\right) \sum_{t=0}^{T-1} \sum_{j=0}^{T-1-t} \left(1 - \frac{\gamma}{3}\right)^j \mathbb{E}\left[\left\|\sum_{k=1}^{K_y} \hat{p}_k \nabla\Phi_\rho^k(\boldsymbol{\theta}(t))\right\|^2\right].
\end{aligned}
\tag{99}
$$

It can be verified that for any $t$, $\sum_{j=0}^{t-1}(1 - \gamma/3)^j \leq 3/\gamma$, and plugging this into the above inequality we get

$$
\begin{aligned}
\mathbb{E}\left[\sum_{k=1}^{K_y} \hat{p}_k \Phi_\rho^k(\boldsymbol{\theta}(T))\right] &\leq \mathbb{E}\left[\sum_{k=1}^{K_y} \hat{p}_k \Phi_\rho^k(\boldsymbol{\theta}(0))\right] - \frac{\eta_{\boldsymbol{\theta}}}{3} \sum_{t=0}^{T-1} \mathbb{E}\left[\left\|\sum_{k=1}^{K_y} \hat{p}_k \nabla\Phi_\rho^k(\boldsymbol{\theta}(t))\right\|^2\right] \\
&+ \frac{12\eta_{\boldsymbol{\theta}} M^2 \tilde{M}_\rho}{\gamma} + \left[\frac{T\tilde{L}\eta_{\boldsymbol{\theta}}^2}{2} + \eta_{\boldsymbol{\theta}} \tilde{M}_\rho \left(2\gamma^2 + \frac{2\eta_{\boldsymbol{\theta}}^2 L_{11}}{\gamma}\right) \frac{3T}{\gamma}\right] \sigma^2,
\end{aligned}
\tag{100}
$$

where $\tilde{M}_\rho = \frac{\tilde{L}K_y L_{12}^2}{\rho} + \frac{(\sigma^2 + L_{11}) L_{12}^2}{2\rho^2}$ and we use the fact $\eta_{\boldsymbol{\theta}} \leq \min\left\{\frac{1}{12\tilde{L}}, \frac{\gamma}{12\sqrt{\tilde{M}_\rho L_{11}}}\right\}$. Finally from (96) and $\gamma = T^{-\frac{2}{5}}$, $\eta_{\boldsymbol{\theta}} \leq T^{-\frac{3}{5}}$ we get that

$$
\begin{aligned}
\frac{1}{T} \sum_{t=0}^{T-1} \mathbb{E}\left[\left\|\sum_{k=1}^{K_y} \hat{p}_k \nabla\Phi_\rho^k(\boldsymbol{\theta}(t))\right\|^2\right] &\leq \frac{3}{\eta_{\boldsymbol{\theta}} T}\left(M + \frac{\rho}{K_z e}\right) + \frac{36 M^2 \tilde{M}_\rho}{\gamma T} + \frac{3\tilde{L}\eta_{\boldsymbol{\theta}} \sigma^2}{2} \\
&+ 18\gamma \tilde{M}_\rho \sigma^2 + \frac{18\eta_{\boldsymbol{\theta}}^2 L_{11} \tilde{M}_\rho \sigma^2}{\gamma^2} \\
&\leq 3T^{-\frac{2}{5}}\left(M + \frac{\rho}{K_z e}\right) + 36 M^2 \tilde{M}_\rho T^{-\frac{3}{5}} + \frac{3\tilde{L}T^{-\frac{3}{5}}\sigma^2}{2} + 18 T^{-\frac{2}{5}}\tilde{M}_\rho \sigma^2 + 18 T^{-\frac{2}{5}} L_{11} \tilde{M}_\rho \sigma^2 \\
&= \mathcal{O}\left(T^{-\frac{2}{5}}\right).
\end{aligned}
\tag{101}
$$

This completes our proof to the first conclusion. We highlight that the value of $\eta_{\boldsymbol{\theta}}$ satisfies

$$
\eta_{\boldsymbol{\theta}} = \min\left\{\frac{1}{12\tilde{L}}, \frac{T^{-\frac{2}{5}}}{12\sqrt{\tilde{M}_\rho} L_{11}}, T^{-\frac{3}{5}}, \frac{T^{-\frac{2}{5}}}{2\sqrt{6K_y} L_{11} L_{12}}\right\} = \mathcal{O}\left(T^{-\frac{3}{5}}\right).
\tag{102}
$$

To see the last conclusion, similar to (96) we have that

$$
\begin{aligned}
\Phi_\rho^k(\boldsymbol{\theta}) - \Phi^k(\boldsymbol{\theta}) &= \lambda \boldsymbol{F}^k(\boldsymbol{\theta})^\top \left(\boldsymbol{u}_k^*(\boldsymbol{\theta}) - \boldsymbol{u}_k^*(\boldsymbol{\theta}, \rho)\right) - \lambda\rho \boldsymbol{u}_k^*(\boldsymbol{\theta}, \rho)^\top \log\left(K_z^2 \boldsymbol{u}_k^*(\boldsymbol{\theta}, \rho)\right) \\
&\leq \lambda \boldsymbol{F}^k(\boldsymbol{\theta})^\top \left(\boldsymbol{u}_k^*(\boldsymbol{\theta}, \rho) - \boldsymbol{u}_k^*(\boldsymbol{\theta})\right) + \frac{\lambda\rho}{K_z e}
\end{aligned}
\tag{103}
$$

where $\boldsymbol{u}_k^*(\boldsymbol{\theta}) = \arg\max\{i : \Phi^k(\boldsymbol{\theta})(i)\}$. Due to Theorem 1 in (Epasto et al., 2020), we have

$$
\boldsymbol{F}^k(\boldsymbol{\theta})^\top \left(\boldsymbol{u}_k^*(\boldsymbol{\theta}) - \boldsymbol{u}_k^*(\boldsymbol{\theta}, \rho)\right) \leq 2\rho \log K_z.
\tag{104}
$$

Thus we can conclude

$$
\left|\Phi_\rho^k(\boldsymbol{\theta}) - \Phi^k(\boldsymbol{\theta})\right| \leq \lambda\rho\left(\frac{1}{K_z e} + 2\log K_z\right)
\tag{105}
$$

which implies our conclusion. $\qquad\square$

Table 4: Test accuracy (%) of linear model on the OOD test data of `Toy example`. The OOD test data are drawn from distributions with different $\sigma_{YZ}^{\text{test}}$. The results are the mean of five independent runs.

| Method / $\sigma_{YZ}^{\text{test}}$ | 0.00 | -0.20 | -0.40 | -0.60 | -0.80 | -0.99 |
|---|---|---|---|---|---|---|
| ERM | 88.5 | 83.0 | 68.0 | 53.5 | 33.5 | 27.2 |
| IRM | 96.0 | 90.5 | 91.0 | 90.5 | 91.0 | 90.5 |
| GroupDRO | 96.5 | 94.5 | 93.0 | 91.0 | 90.5 | 88.0 |
| Correlation | 92.5 | 87.5 | 79.0 | 69.0 | 43.0 | 39.5 |
| RCSV | **99.5** | **99.0** | **98.5** | **97.5** | **98.0** | **97.0** |
| RCSV$_{\text{U}}$ | 98.5 | 97.0 | 96.0 | 95.0 | 94.5 | 89.5 |

Table 5: Cosine-similarity $\langle \boldsymbol{\theta}_2, \boldsymbol{\mu}_2 \rangle / (\|\boldsymbol{\theta}_2\| \|\boldsymbol{\mu}_2\|)$ of linear models trained on different methods. Model with smaller cosine-similarity theoretically exhibits better OOD generalization ability.

| Methods | ERM | IRM | GroupDRO | Correlation | RCSV | RCSV$_{\text{U}}$ |
|---|---|---|---|---|---|---|
| $\frac{\langle \boldsymbol{\theta}_2, \boldsymbol{\mu}_2 \rangle}{\|\boldsymbol{\theta}_2\| \|\boldsymbol{\mu}_2\|}$ | 0.95 | 0.13 | 0.18 | 0.89 | **0.03** | 0.05 |

# E  MORE EXPERIMENTS

In this section, we conduct more experiments on a synthetic dataset and real-world dataset to further verify the effectiveness of our proposed methods.

## E.1  TOY EXAMPLE

In this section, we apply the proposed RCSV and RCSV$_{\text{U}}$ to a constructed toy example with spurious correlation.

**Data.** The data is constructed as the example in Appendix B. For two 5-dimensional vectors $\boldsymbol{\mu}_1, \boldsymbol{\mu}_2$, the training data $X$ follows normal distribution $\mathcal{N}((Y\boldsymbol{\mu}_1^\top, Z\boldsymbol{\mu}_2^\top)^\top, \boldsymbol{I}_{10})$ where $\boldsymbol{I}_{10}$ is a $10 \times 10$ identity matrix. The label $Y$ and spurious attributes $Z$ take value from $\{-1, 1\}$ and are all drawn from a standard binomial distribution (i.e., $P_Y(Y = 1) = P_Y(Y = -1) = 0.5$). As in (1), the spurious correlation coefficient $\sigma_{YZ}^{\text{train}}$ between $Y$ and $Z$ vary on different distribution. We generate 1000 (resp. 200) training (resp. test) samples. Concretely, the 1 is fixed as 0.99 for the unique training distribution, while there are 6 constructed test distributions respectively with $\sigma_{YZ}^{\text{test}}$ in $\{0.00, -0.20, -0.40, -0.60, -0.80, -0.99\}$. As can be seen, the spurious correlations in the test sets are opposite to the one in the training set. Thus, over-fitting the spurious correlation will mislead the trained model.

**Setup.** We use the linear model $f_{\boldsymbol{\theta}}(\boldsymbol{x}) = \boldsymbol{\theta}^\top \boldsymbol{x}$ and its prediction on $Y$ is $\text{sign}(f_{\boldsymbol{\theta}}(\boldsymbol{x}))$. We compare the proposed methods RCSV and RCSV$_{\text{U}}$ with the baseline methods as in the main body of this paper. The domain generalization methods can be applied with observed spurious attributes is because the data can be viewed as from two domains i.e., data drawn under conditions of $Y = Z$ and $Y \neq Z$. The loss function $\mathcal{L}(\cdot, \cdot)$ is cross entropy. The hyperparameters of baseline methods follow the ones in original papers. Our methods are trained by SGD with the used hyperparemeters deferred in Appendix G.4.

**Main Results.** In Table 4, we report the test accuracies of trained models evaluated on OOD data to see if all the aforementioned methods can break the spurious correlation. From the results, we have the following observations.

For all methods, the test accuracies are consistently improved with the decrease of the gap between $\sigma_{YZ}^{\text{train}} - \sigma_{YZ}^{\text{train}}$. This is explained as the decreased $\sigma_{YZ}^{\text{train}} - \sigma_{YZ}^{\text{test}}$ leads to smaller mismatches between training and test distributions, thus improving accuracy.

The models trained by the proposed two methods and domain generalization methods (IRM and GroupDRO) can break the spurious correlation (generalize on OOD test data), which verifies the

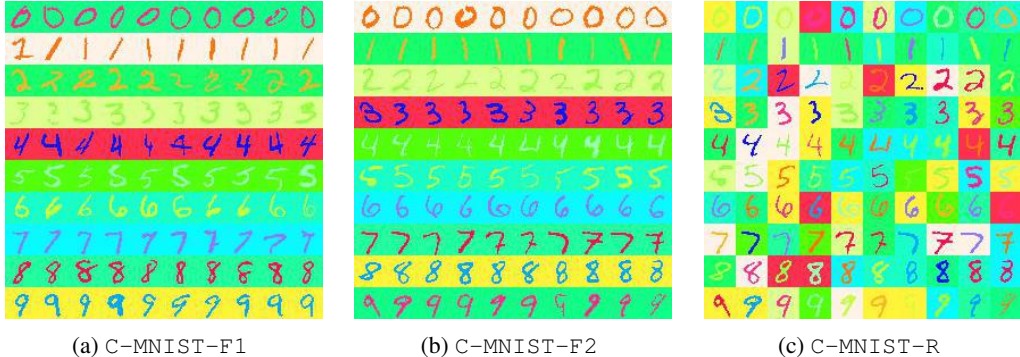

(a) C-MNIST-F1       (b) C-MNIST-F2       (c) C-MNIST-R

Figure 2: Images of three C-MNIST datasets with different spurious correlations. The first two have fixed spurious correlation between colors and the label of digits, while the spurious correlation in the last one is random.

Table 6: Test accuracy (%) of convolution neural networks trained on the different mixtures of C-MNIST-F1 and C-MNIST-R. The OOD test data either C-MNIST-R or C-MNIST-F2.

| Test set | C-MNIST-R | | | | | C-MNIST-F2 | | | | |
|---|---|---|---|---|---|---|---|---|---|---|
| Method/$\alpha$ | 0.80 | 0.85 | 0.90 | 0.95 | 0.99 | 0.80 | 0.85 | 0.90 | 0.95 | 0.99 |
| ERM | 96.1 | 95.4 | 93.3 | 87.5 | 63.8 | 95.6 | 95.2 | 92.8 | 82.6 | 52.8 |
| IRM | 91.2 | 90.5 | 90.4 | 87.9 | 76.6 | 94.3 | 93.5 | 91.1 | 87.2 | 63.1 |
| GroupDRO | 96.6 | 95.9 | 93.7 | 92.1 | 76.8 | 95.5 | 95.3 | 94.3 | 91.5 | 70.2 |
| Correlation | 82.6 | 79.6 | 79.5 | 72.6 | 38.5 | 82.6 | 78.4 | 68.2 | 35.6 | 25.2 |
| RCSV | **98.0** | **97.6** | **96.7** | **94.3** | **81.3** | **96.9** | **96.3** | **95.1** | **90.2** | **81.3** |
| RCSV$_\text{U}$ | 97.9 | 97.5 | 96.5 | 94.1 | 77.9 | 96.8 | 95.6 | 93.6 | 86.9 | 62.3 |

effectiveness of our methods. On the other hand, both RCSV and RCSV$_\text{U}$ beats the domain generalization methods which require the domain label (i.e., the spurious attributes $Z$ in the constructed dataset). Thus, the extra information required by RCSV (resp. RCSV$_\text{U}$) is equivalent (resp. less) compared with domain generalization methods.

On the other hand, let the last 5-dimensional parameters of the linear model be $\boldsymbol{\theta}_2$. By $X \sim \mathcal{N}((Y\boldsymbol{\mu}_1^\top, Z\boldsymbol{\mu}_2^\top)^\top, \boldsymbol{I}_{10})$, one can verify that the when $\boldsymbol{\theta}_2^\top \boldsymbol{\mu}_2 \approx 0$, the output of model $\boldsymbol{\theta}^\top X$ does not related to $Z$ with high probability. Then the model can break the spurious correlation.

To see this, in Table 5, we present the cosine-similarity $\langle \boldsymbol{\theta}_2, \boldsymbol{\mu}_2 \rangle / (\|\boldsymbol{\theta}_2\| \|\boldsymbol{\mu}_2\|)$ (the cosine-similarity is used to alleviate the interference caused by scales of the two vectors) of the models trained by methods in Table 4. The results show that the models trained by OOD generalizable methods have smaller cosine-similarities.

### E.2 COLORED-MNIST

In this section, we empirically verify the effectiveness of the proposed methods on a constructed real-world dataset Colored-MNIST.

**Data.** Our dataset is constructed on the MNIST (LeCun et al., 1998) which consists of 60,000 training data and 10,000 test data. Each data is a grey-scale hand-written digit from ten categories, i.e., 0 to 9. We construct our Colored-MNIST (C-MNIST) by inducing the spurious correlation in the training and test sets. Concretely, for each digit, we assign two colors as spurious attributes respectively for its foreground and background. The spurious correlation can be induced into such dataset by tying the relationship between the label of digits and the two colors.

We pick 20 specific colors, the first and the last 10 colors are respectively used as 10 categories of two spurious attributes, i.e., the colors of foreground and background of a digit. We consider datasets with two kinds of spurious correlations. The first is fixed spurious correlation, which means data from each *specific* category of digit is assigned two *specific* colors respectively for its foreground

and background. The other is random spurious correlation which means that for each data, two *randomly* sampled colors are respectively assigned to its foreground and background regardless of its category. We will construct two `C-MNIST` with different but fixed spurious correlations (abbrev. as `C-MNIST-F1` and `C-MNIST-F2`), and one `C-MNIST` with random spurious correlation (abbrev. as `C-MNIST-R`).

Some of the generated datasets are in Figure 2. As can be seen, the three versions of `C-MNIST` has different spurious correlations between the label of digits and the colors of foreground and background. Besides that, the spurious correlation in `C-MNIST-F1` and `C-MNIST-F2` are fixed while `C-MNIST-R` has randomized spurious correlation.

**Setup.** We construct various training sets based on the original 60,000 training samples of `MNIST`. Concretely, we choose $\alpha \in \{0.8, 0.85, 0.90, 0.95, 0.99\}$, then for each $\alpha$, we construct a training set with $\lfloor 60,000 \times \alpha \rfloor$ [4] samples are from `C-MNIST-F1` while the other $\lfloor 60,000 \times (1-\alpha) \rfloor$ are constructed as `C-MNIST-R`. We use two test sets which are respectively the 10,000 test samples constructed as `C-MNIST-F2` and `C-MNIST-R`. Obviously, the data from `C-MNIST-R` in the training set alleviates the misleading signal from the training set brought by `C-MNIST-F1` due to the spurious correlation between color and digit in it, and the existence of these data meets the Assumption 1.

Our model is a five-layer convolution neural network in (Devansh Arpit, 2019). The models are trained over the 5 aforementioned datasets with different $\alpha$ by the methods that appeared in the above section. One can verify that the training set can be viewed as a mixture of data from two domains, i.e., `C-MNIST-F1` and `C-MNIST-R`. Thus the domain generalization based methods IRM and GroupDRO can be applied here. The used loss function $\mathcal{L}(\cdot, \cdot)$ is cross-entropy, and detailed hyperparameters are presented in Appendix G.4.

**Main Results.** To see if the models trained by these methods can break the induced misleading spurious correlation, we report their test accuracies on the `C-MNIST-R` and `C-MNIST-F2`. The results are summarized in Table 6 with the following observations from it.

The test accuracies of all these methods increase with the decreased $\alpha$. This is a natural result since smaller $\alpha$ corresponds with more training samples from `C-MNIST-R` which alleviates the misleading signal from the data with spurious correlation in `C-MNIST-F1`. Thus models trained over the training set with smaller $\alpha$ exhibit improved generalization ability OOD data with correlation shift.

Similar to the results in Section 6, our RCSV (resp. RCSV$_\text{U}$) consistently improve the OOD generalization error, compared with the methods with (resp. without) observed spurious attributes. More surprisingly, RCSV$_\text{U}$ beats the methods with observed spurious attributes methods IRM and GroupDRO for a large $\alpha$. The observations again verify the efficacy of our proposed methods.

The model trained by the most commonly used method ERM on datasets with small $\alpha$ also generalizes on OOD data. Thus a relatively large number of data without spurious correlation in the training set also breaks the spurious correlation brought by other data.

Finally, we observe that the performance of models on `C-MNIST-R` consistently better than on `C-MNIST-F2`. This is due to there exist data drawn from `C-MNIST-R` in the training set, while the data from `C-MNIST-F2` does not appear in the training set.

# F    ABLATION STUDY

We have discussed in Section 6 that the reweighed sampling trick improves the OOD generalization. Thus, we explore the effect of such trick in this section.

We follow the settings in main part of this paper, expected for the reweighted sampling strategy is set as uniformly sampling, thus the methods ERMRS$_\text{YZ}$ and ERMRS$_\text{Y}$ become the ERM. The results are summarized in Tables 7, 8, and 9.

---

[4]$\lfloor \cdot \rfloor$ is the floor of a number

Table 7: Test accuracy (%) of ResNet50 on each group of CelebA and Waterbirds. The experiments are conducted without reweighted sampling trick.

| Dataset | Method / Group | D-F | D-M | B-F | B-M | Avg | Total | Worst | SA |
|---|---|---|---|---|---|---|---|---|---|
| CelebA | RCSV | 91.1 | 91.0 | 92.9 | 92.2 | **91.8** | 91.3 | **91.0** | √ |
| | IRM | 90.1 | 92.3 | 90.1 | 86.1 | 89.7 | 91.0 | 86.1 | √ |
| | GroupDRO | 90.3 | 93.0 | 94.3 | 87.2 | 91.2 | 91.8 | 87.2 | √ |
| | RCSV$_U$ | 94.0 | 98.6 | 91.1 | 60.0 | 85.9 | **95.1** | 60.0 | × |
| | Correlation | 94.1 | 99.3 | 82.1 | 35.7 | 77.8 | 94.0 | 35.7 | × |
| | ERM | 95.4 | 99.5 | 82.8 | 41.8 | 79.9 | 94.8 | 41.8 | × |

| Dataset | Method / Group | L-L | L-W | W-L | W-W | Avg | Total | Worst | SA |
|---|---|---|---|---|---|---|---|---|---|
| Waterbirds | RCSV | 97.1 | 86.0 | 86.4 | 93.3 | **90.7** | 91.2 | **86.0** | √ |
| | IRM | 98.6 | 90.6 | 79.4 | 90.8 | 89.9 | **92.5** | 79.4 | √ |
| | GroupDRO | 98.4 | 94.4 | 71.5 | 85.4 | 89.9 | 92.4 | 71.5 | √ |
| | RCSV$_U$ | 99.0 | 81.1 | 77.3 | 93.6 | 87.8 | 89.0 | 77.3 | × |
| | Correlation | 99.9 | 88.5 | 59.0 | 90.3 | 84.4 | 89.9 | 59.0 | × |
| | ERM | 99.6 | 88.5 | 58.1 | 92.5 | 84.7 | 90.6 | 58.1 | × |

Table 8: Test accuracy (%) of BERT on each group of MultiNLI. The experiments are conducted without reweighted sampling trick.

| Dataset | Method / Group | C-WN | C-N | E-WN | E-N | N-WN | N-N | Avg | Total | Worst | SA |
|---|---|---|---|---|---|---|---|---|---|---|---|
| MultiNLI | RCSV | 79.8 | 94.4 | 83.8 | 76.5 | 79.2 | 70.6 | 80.7 | 81.6 | **70.6** | √ |
| | IRM | 79.2 | 94.2 | 83.9 | 74.2 | 79.1 | 67.6 | 79.7 | 81.4 | 67.6 | √ |
| | GroupDRO | 80.4 | 94 | 82.4 | 76.2 | 80.8 | 70.3 | 80.7 | **81.8** | 70.3 | √ |
| | RCSV$_U$ | 80.1 | 94.2 | 83.6 | 80.1 | 78.2 | 67.4 | 80.6 | 81.3 | 67.4 | × |
| | Correlation | 73.1 | 91.2 | 76.3 | 64.5 | 77.9 | 62.4 | 74.2 | 77.8 | 62.4 | × |
| | ERM | 80.4 | 94.8 | 83.6 | 81.4 | 78.6 | 66.6 | **80.9** | 81.5 | 66.6 | × |

Table 9: Test accuracy (%) of BERT on each group of CivilComments. The experiments are conducted without reweighted sampling trick.

| Dateset | Method / Group | N-N | N-I | T-N | T-I | Avg | Total | Worst | SA |
|---|---|---|---|---|---|---|---|---|---|
| CivilComments | RCSV | 93.1 | 91.8 | 72.4 | 70.6 | **82.0** | 90.2 | **70.6** | √ |
| | IRM | 93.0 | 86.3 | 74.9 | 67.8 | 80.5 | 88.1 | 67.8 | √ |
| | GroupDRO | 92.9 | 88.3 | 77.9 | 65.1 | 81.1 | 88.8 | 65.1 | √ |
| | RCSV$_U$ | 97.4 | 94.3 | 69.0 | 62.9 | 80.9 | **92.7** | 62.9 | × |
| | Correlation | 97.1 | 92.3 | 66.5 | 61.7 | 79.4 | 91.6 | 61.7 | × |
| | ERM | 96.6 | 92.1 | 69.4 | 56.3 | 78.6 | 91.1 | 56.3 | × |

As can be seen from these tables, the OOD generalization performance of model drops for all these methods compared with the results in Section 6, especially for CelebA and Waterbirds, see the column of "Avg" and "Worst" in each table. We speculate this is because the reweighted sampling strategy enables the data in each group are equivalently appeared during training, this operation itself can break the spurious correlation in training data. Another evidence to support the degenerated OOD generalization is the improved test accuracies on the groups with similar spurious attributes in training data, e.g., the better performances on the groups D-F, D-M of CelebA and L-L, W-W of Waterbirds.

The other observation is that even without this trick, our methods improve the OOD generalization compared with other baseline methods due to their better mean and worst test accuracies.

Finally, the trade-off between the robustness over spurious attributes and in-distribution test accuracies is more clearly observed in these tables. This is from the comparisons between accuracy gap of data with same spurious attributes and total accuracy, which is in-distribution test accuracy for CelebA, MultiNLI, and CivilComments.

## G SETUP FOR EXPERIMENTS

### G.1 IMPLEMENTATION OF TWO PROPOSED ALGORITHMS

In this section, we present the detailed algorithm flows of the proposed RCSV and $\widehat{\mathrm{RCSV}}_\mathrm{U}$ in the main body of this paper. The critical part is their estimators to the $\boldsymbol{F}^k(\boldsymbol{\theta})$ defined in Section 5.

---

**Algorithm 2** Regularize training with $\widehat{\mathrm{CSV}}$ (RCSV).

---

**Input:** Training samples $\{(\boldsymbol{x}_i, y_i)\}_{i=1}^n$, number of labels $K_y$ and spurious attributes $K_z$, batch size $S$, learning rate $\eta_{\boldsymbol{\theta}}$, training iterations $T$, model $f_{\boldsymbol{\theta}}(\cdot)$ parameterized by $\boldsymbol{\theta}$. Initialized $\boldsymbol{\theta}_0, \{\boldsymbol{F}_0^k\}$. Positive regularization constant $\lambda$, surrogate constant $\rho$, and correction constant $\gamma$.

1: **for** $t = 0, \cdots, T$ **do**
2:     **Compute the estimator** $\hat{R}_{emp}(f_{\boldsymbol{\theta}(t)}, P)$**;**
3:     $\hat{R}_{emp}(f_{\boldsymbol{\theta}(t)}, P)$ is the empirical risk over a uniformly-drawn batch (size $S$) of data.
4:     **Compute the estimator** $\hat{\boldsymbol{F}}^k(\boldsymbol{\theta}(t))$**,** $k \in [K_y]$**;**
5:     Initialized $K_z$-dimensional vector $\hat{\mathcal{L}}^k = \boldsymbol{0}$, $k \in [K_y]$;
6:     Reweighted sample a mini-batch of data $\{(x_{t,i}, y_{t,i}, z_{t,i})\}$ with replacement, the probability of data satisfies $y_{t,i} = k$ and $z_{t,i} = z$ is $1/(K_y K_z n_{kz})$.
7:     Update $\hat{\mathcal{L}}^k(z)$ as the empirical risk over $\{(x_{t,i}, y_{t,i})\} \bigcap A_{kz}$, $k \in [K_y], z \in [K_z]$
8:     Compute $\hat{\boldsymbol{F}}^k(\boldsymbol{\theta}(t)) = K_y K_z \left(\hat{\mathcal{L}}^k(1) - \hat{\mathcal{L}}^k(1), \cdots, \hat{\mathcal{L}}^k(K_z) - \hat{\mathcal{L}}^k(K_z)\right)$, $k \in [K_y]$
9:     **Solve the maximization problem.**
10:     $\boldsymbol{F}_{t+1}^k = (1 - \gamma)\boldsymbol{F}_t^k + \gamma\hat{\boldsymbol{F}}^k(\boldsymbol{\theta}(t))$;
11:     $\boldsymbol{u}_k(t+1) = \mathrm{Softmax}(\boldsymbol{F}_{t+1}^k/\rho)$.
12:     **Update model parameters** $\boldsymbol{\theta}(t)$ **via SGD.**
13:     $\boldsymbol{\theta}(t+1) = \boldsymbol{\theta}(t) - \eta_{\boldsymbol{\theta}} \sum_{k=1}^{K_y} \hat{p}_k \nabla_{\boldsymbol{\theta}}(\hat{R}_{emp}(f_{\boldsymbol{\theta}(t)}, P) + \lambda\boldsymbol{u}_k(t+1)^\top \boldsymbol{F}_{t+1}^k)$.
14: **end for**

---

---

**Algorithm 3** Regularize training with $\widehat{\mathrm{CSV}}_\mathrm{U}$ (RCSV$_\mathrm{U}$).

---

**Input:** Training samples $\{(\boldsymbol{x}_i, y_i)\}_{i=1}^n$, number of labels $K_y$ and spurious attributes $K_z$, batch size $S$, learning rate $\eta_{\boldsymbol{\theta}}$, training iterations $T$, model $f_{\boldsymbol{\theta}}(\cdot)$ parameterized by $\boldsymbol{\theta}$. Initialized $\boldsymbol{\theta}_0, \{\boldsymbol{F}_0^k\}$. Positive regularization constant $\lambda$, surrogate constant $\rho$, and correction constant $\gamma$.

1: **for** $t = 0, \cdots, T$ **do**
2:     **Compute the estimator** $\hat{R}_{emp}(f_{\boldsymbol{\theta}(t)}, P)$**;**
3:     $\hat{R}_{emp}(f_{\boldsymbol{\theta}(t)}, P)$ is the empirical risk over a uniformly-drawn batch (size $S$) of data.
4:     **Compute the estimator** $\hat{\boldsymbol{F}}^k(\boldsymbol{\theta}(t))$**,** $k = 1, \cdots, K_y$**;**
5:     Initialized $|A_k|^2$-dimensional vector $\hat{\boldsymbol{F}}^k(\boldsymbol{\theta}(t)) = \boldsymbol{0}$, $k =\in [K_y]$;
6:     Reweighted sample a mini-batch of data $\{(x_{t,i}, y_{t,i})\}$ with replacement, the probability of data satisfies $y_{t,i} = k$ is $1/(K_y n_k)$.
7:     Update $\hat{\boldsymbol{F}}^k(j)$ with $\mathcal{L}(f_{\boldsymbol{\theta}}(x_{t,i}), y_{t,i})$ if $(x_{t,i}, y_{t,i})$ is the $j$-th data in $A_k$, $i \in [K_y], j \in [K_z]$.
8:     **Solve the maximization problem.**
9:     $\boldsymbol{F}_{t+1}^k = (1 - \gamma)\boldsymbol{F}_t^k + \gamma\hat{\boldsymbol{F}}^k(\boldsymbol{\theta}(t))$;
10:     $\boldsymbol{u}_k(t+1) = \mathrm{Softmax}(\boldsymbol{F}_{t+1}^k/\rho)$.
11:     **Update model parameters** $\boldsymbol{\theta}(t)$ **via SGD.**
12:     $\boldsymbol{\theta}(t+1) = \boldsymbol{\theta}(t) - \eta_{\boldsymbol{\theta}} \sum_{k=1}^{K_y} \hat{p}_k \nabla_{\boldsymbol{\theta}}(\hat{R}_{emp}(f_{\boldsymbol{\theta}(t)}, P) + \lambda\boldsymbol{u}_k(t+1)^\top \boldsymbol{F}_{t+1}^k)$.
13: **end for**

---

## G.2 DATASET

In this section, we give more details on the datasets appeared in the main part of this paper.

**CelebA.** This is a celebrity face dataset (Liu et al., 2015) with 162770 training samples and 20362 test samples. For each sample, the hair color {Dark, Blond} is class label, while the gender {Female, Male} is spurious attributes. For both training and test datasets, each of them can be divided into 4 groups, i.e., "Dark-Female" (D-F), "Dark-Male" (D-M), "Blond-Female" (B-F), "Blond-Male" (B-M). The numbers of samples in training and test dataset from the 4 groups are respectively {71629, 9767}, {66874, 7535}, {22880, 2880}, {1387, 180}. Our goal is to train a model that correctly recognizes the hair color of celebrities independent of their gender. One can verify that the most difficult group of data to be generalized on is B-M, due to its extremely small proportion in males in the training set.

**Waterbirds.** This is a synthetic real-world dataset in (Sagawa et al., 2019) with 4795 training samples and 6993 test samples, which is constructed based on combining photograph of bird from the Caltech-UCSD Birds-200-2011 (CUB) dataset (Wah et al., 2011) with image backgrounds from the Places (Zhou et al., 2017). For each image, its class label is from {Waterbird, Landbird}, and each bird is placed on spurious attributes: background from {Land background, Water background}. As in CelebA, the datasets can be categorized into 4 groups, i.e., "Landbird-Land background" (L-L), "Landbird-Water background" (L-W), "Waterbird-Water background" (W-W), "Waterbird-Land background" (W-L). The training and test datasets are constructed with the numbers of samples in each group are respectively {3498, 2255} (L-L), {184, 2255} (L-W), {56, 642} (W-W), {1057, 642} (W-L). As can be seen, the spurious correlations in the training and test sets are quite different. In the training set, most landbirds are on the land, and most waterbirds are on the water. But in the test set, waterbirds and landbirds are uniformly assigned on the two backgrounds. Thus, we are desired to train a model that breaks the spurious correlation between bird and background. The proportion of 4 groups in the training set informs that the most difficult of them to be generalized on are L-W and W-L.

**MultiNLI.** This is a dataset for natural language inference (Williams et al., 2018) with 206175 training samples and 123712 test samples. The dataset is consists of pair of sentences, and our goal is to recognize that whether the second sentence is entailed by, neutral with, or contradicts to the first sentence. It was explored in Gururangan et al. (2018) that there exists spurious correlation in the dataset such as the contradiction can be related to the presence of the negation words *nobody, no, never,* and *nothing*. Thus we set such presence as spurious attribute and the dataset can be categorized into 6 groups, i.e., "Contradiction-Without Negation" (C-WN), "Contradiction-Negation" (C-N), "Entailment-Without Negation" (E-WN), "Entailment-Negation" (E-N), "Neutrality-Without Negation" (N-WN), "Neutrality-Negation" (N-N). Our goal is learning a model that makes prediction independent with the presence of negation. The numbers of samples in training and test dataset from the 6 groups are respectively {57498, 34597}, {11158, 6655}, {67376, 40496}, {1521, 886}, {66630, 39930}, {1992, 1146}.

**CivilComents.** This is a dataset consists of collected online comments (Borkan et al., 2019). The dataset has 269038 training data and 133782 test data. Our goal is to recognize whether the comment is toxic or not. The toxicity can be spurious correlated with the annotation attributes such the presence of 8 certain demographic identities includes male, female, White, Black, LGBTQ, Muslim, Christian, and other religion. Thus we set the identity of any aforementioned words as the spurious attributes, and divided the dataset into 4 groups: "Nontoxic-Nonidentity" (N-N), "Nontoxic-Identity" (N-I), "Toxic-Nonidentity" (T-N), "Toxic-Identity" (T-I). The numbers of samples in training and test dataset from the 4 groups are respectively {148186, 72373}, {90337, 46185}, {12731, 6063}, {17784, 9161}. As can be seen, there exists a spurious correlation between the toxicity and the identity attribute in the training set due to the number of data in each group.

For all these datasets, from the number of data in each group, there exists dominated spurious correlation in CelebA and Waterbirds. But this does not happened in MultiNLI and CivilComments, especially for MultiNLI as the strong spurious correlation only exists in the group of "C-WN" v.s. "C-N". Thus for the MultiNLI and CivilComments, expected for the spurious feature, the model should extract other features to guarantee good performance.

## G.3 BENCHMARK ALGORITHMS

Empirical Risk minimization (ERM, Vapnik, 1999) pools together the data from all the domains and then minimizes the empirical loss to train the model.

Empirical Risk minimization with reweighted sampling (ERMRS, Idrissi et al., 2021) is similar to empirical risk minimization, but it reweight the sampling probability of each sample, and the weightes on each data is pre-defined.

Invariant Risk Minimization (IRM, Arjovsky et al., 2019) learns a feature representation such that the optimal classifiers on top of the representation is the same across the domains.

Group Distributionally Robust Optimization (GroupDRO, Sagawa et al., 2019) minimizes the worst-case loss over different domains.

(Correlation, Devansh Arpit, 2019) minimizes the intra-variance of data from the same category to break the spurious correlation.

## G.4 TRAINING DETAILS

As clarified in Section 6, the backbone models for image datasets (`CelebA`, `Waterbirds`) and textual datasets (`MultiNLI`, `CivilComments`) are respectively ResNet-50 (He et al., 2016) pre-trained on `ImageNet` (Deng et al., 2009) and pre-trained BERT Base model(Devlin et al., 2019).

The loss function $\mathcal{L}(\cdot, \cdot)$ is cross-entropy for all of these methods. The experiments on image datasets are conducted without learning rate decay while the results on textual datasets are obtained with linearly decayed learning decay via optimizer AdamW (Loshchilov & Hutter, 2018).

The hyperparameters of baseline methods follow the original one in (Gulrajani & Lopez-Paz, 2020; Sagawa et al., 2019; Devansh Arpit, 2019; Arjovsky et al., 2019; Idrissi et al., 2021). The hyperparameters of the proposed RCSV and RCSV$_\text{U}$ on `CelebA`, `Waterbirds`, `MultiNLI`, `CivilComments`, `Toy example` and `C-MNIST` respectively summarized in Table 10, 11, 12, 13, 14, and 15.

Table 10: Hyperparameters on `CelebA`.

| Algorithm | RCSV | RCSV$_\text{U}$ |
|---|---|---|
| Optimizer | Adam | Adam |
| Learning Rate | 1e-5 | 1e-5 |
| Batch Size | 256 | 256 |
| Weight Decay | 1e-4 | 1e-4 |
| Epoch | 50 | 50 |
| $\lambda$ | 5 | 1 |
| $\gamma$ | 0.9 | 0.9 |
| $\rho$ | 1e-4 | 1e-4 |

Table 11: Hyperparameters on `Waterbirds`.

| Algorithm | RCSV | RCSV$_\text{U}$ |
|---|---|---|
| Optimizer | Adam | Adam |
| Learning Rate | 1e-5 | 1e-5 |
| Batch Size | 64 | 64 |
| Weight Decay | 1e-4 | 1e-4 |
| Epoch | 300 | 300 |
| $\lambda$ | 0.1 | 0.1 |
| $\gamma$ | 0.9 | 0.9 |
| $\rho$ | 1e-4 | 1e-4 |

Table 12: Hyperparameters on `MultiNLI`.

| Algorithm | RCSV | RCSV$_\text{U}$ |
|---|---|---|
| Optimizer | AdamW | AdamW |
| Learning Rate | 1e-5 | 1e-5 |
| Batch Size | 32 | 32 |
| Weight Decay | 0 | 0 |
| Epoch | 3 | 3 |
| Drop Out | 0.1 | 0.1 |
| $\lambda$ | 0.1 | 0.1 |
| $\gamma$ | 0.9 | 0.9 |
| $\rho$ | 1e-4 | 1e-4 |

Table 13: Hyperparameters on `CivilComments`.

| Algorithm | RCSV | RCSV$_\text{U}$ |
|---|---|---|
| Optimizer | Adam | Adam |
| Learning Rate | 1e-5 | 1e-5 |
| Batch Size | 16 | 16 |
| Weight Decay | 0 | 0 |
| Epoch | 3 | 3 |
| Drop Out | 0.1 | 0.1 |
| $\lambda$ | 0.1 | 0.1 |
| $\gamma$ | 0.9 | 0.9 |
| $\rho$ | 1e-4 | 1e-4 |

Table 14: Hyperparameters on `Toy example`.

| Algorithm | RCSV | $RCSV_U$ |
|---|---|---|
| Optimizer | SGD | SGD |
| Learning Rate | 0.01 | 0.01 |
| Momentum | 0.9 | 0.9 |
| Batch Size | 32 | 32 |
| Weight Decay | 0 | 0 |
| Epoch | 100 | 100 |
| $\lambda$ | 1.0 | 5 |
| $\gamma$ | 0.9 | 0.9 |
| $\rho$ | 0.01 | 0.01 |

Table 15: Hyperparameters on `C-MNIST`.

| Algorithm | RCSV | $RCSV_U$ |
|---|---|---|
| Optimizer | Adam | Adam |
| Learning Rate | 0.001 | 0.001 |
| Momentum | / | / |
| Batch Size | 128 | 128 |
| Weight Decay | 1e-4 | 1e-4 |
| Epoch | 40 | 40 |
| $\lambda$ | 1.0 | 0.05 |
| $\gamma$ | 0.9 | 0.9 |
| $\rho$ | 0.01 | 0.01 |

