# OpenReview forum: "Breaking Correlation Shift via Conditional Invariant Regularizer"
_ICLR.cc/2023/Conference — ICLR 2023 poster_

### Official Review · Reviewer_Lqom · 2022-10-21

**Confidence:** 3
**Correctness:** 3
**Technical Novelty And Significance:** 4
**Empirical Novelty And Significance:** Not applicable
**Recommendation:** 8

**Clarity, Quality, Novelty And Reproducibility:**

This paper is written in a statistics style, and is clear and easy to follow. The proposed analysis and the idea of CSV (Conditional Spurious Variation) are novel and of good quality. Since the algorithm description is clear, I believe that the numerical results are reproducible. But it is clear that the main contribution of this paper is theory.

**Details Of Ethics Concerns:**

The paper studies machine learning theory, and there is no ethics concern.

**Strength And Weaknesses:**

Strength:
The idea of CSV-based regularization is promising and innovative.

Weaknesses
Below I list some possible mistakes or typos, most very easy to fix. I feel that "weakness" could be a word too big for these problems.
1. some terms may not be well known. I would suggest adding explanations. This includes: "domain labels"
2. paragraph 3 on page 3: what is the difference between "input's label" and "class label"?
3. I would suggest a comparison of the proposed condition with the condition in this paper:
Smale, Steve; Zhou, Ding-Xuan. Online learning with Markov sampling. Anal. Appl. (Singap.) 7 (2009), no. 1, 87--113.
For example, I see many works in the literature assuming the invariant of conditional distribution of labels instead. I am curious about the difference between this setting, and the setting used in the current paper.
4. Some small typos: "quantify" before Theorem 4 should be "quantifies";
5. Please define the big-O notation in Theorem 4. In particular, what is the limit process?
6. I hope that the author(s) can specify the definition of P, Q, X, Y, and Z, which I believe would greatly help the readers. In particular, I have the following questions directly related to the paper:
(a). Is "P" (and "Q") a distribution of (X,Y) or (X,Y,Z)?
(b). In general, X and Y are not independent. Are X and Z independent? Are Y and Z independent? Are (X,Y) and Z independent?
(c). If Z and Y are not independent, then Z has the prediction power on Y. Would this still fit the definition of "spurious correlation" in this paper?
7. In this paper the label set script-Y is discrete (Assumption 1). For the time being, let me assume the label set is just {0, 1}. Then the loss function script-L just assigns values for the set {(0,0), (0,1), (1, 0), (1,1)}. I think it does not make much sense to assume that L(u,y)=L(y) (the value depending only on y). I think it does not make much sense to assume in Assumption 2, that the parameter set Theta is disconnected and for any x, f_{theta}(x) depends on theta only through the connected component of the parameter set Theta. However, then there exist some x, y, theta1, and theta2 such that: theta1 and theta2 are path connected in the parameter space Theta (we skip the subtleties of path connected and connected spaces), f_{theta1}(x)=1, f_{theta2}=0, and L(1,y)!=L(0,y). But this would violate Assumption 2 because if we let theta travel along the path linking theta1 and theta2, at some stage f_{theta}(x) would jump from 1 to 0. Such a jump violates the Lipschitz continuity in Assumption 2, no matter how large L0 is.
So, Assumption 2 requires that the parameter space Theta is discrete and the distance of every pair of points uniformly bounded away from zero. As this is a very strong assumption for machine learning or statistics community, we suggest that this should be explicitly stated after Assumption 2. As a consequence, the Lipschitz smoothness of L in (6) is no longer necessary and should be removed.
8. Right after Theorem 4, I guess the expression "A decreases with B" is not clear: what is its precise meaning?
9. The definition (11) is confusing. It is obvious that when u runs through Delta_m, the maximum of u dot F is the maximum coordinate of the vector F. Why not directly use max-coordinate but employing Delta_m and writing (11) in the current complicated way?
10. Page 14, right after (17), how does one derive P_{X|Y}=Q_{X|Y} without assuming P_Z=Q_Z? Also in Example 1, all the numbers 0 should be -1.
11. I feel that the loss function in Example 1 is not designed properly. Indeed, L(1, -1)=3 while L(1, 1) = 4 is even larger. Also, the inequality (19) is derived based on the assumption that f(X)=1 almost surely, so the claim that (19) holds for any f is wrong. Even if we would like to cancel E[f(X)] with X|Y ~ Q or P, since the first term is conditioned on Y=1 and the second term is conditioned on Y=0, such cancellation is not doable in general.
12. Proof of Propsition 2: please provide section or page number when citing a book. In this proof however, since script-Y is discrete, one may simply choose A=Script-Y and there is no need to take supremum.
13. Equation (24) is wrong on the treatment of sup_{A in script-Y}.
14. Equation (25) is wrong. For example in the last equation, the left-hand side depends on y, while the right-hand side does not.
15. Proposition 2 is wrong. Obviously, since Q is in script-P, Q_Y=P_Y, so the only function w that achieves the minimum TV is a constant function w(y)=1, which makes the TV equal to zero.
16. In the proof of Proposition 1, the classifier does not fit the framework proposed in this paper. In particular, the first argument of the loss function script-L is defined on script-Y, a discrete space. However, the classifier here outputs a real number. I guess a reasonable solution is to change the definition of the loss function to accept real first argument (at the top of page 3). This change may help to resolve the issue on Assumption 2.
17. I think the first part of the proof of Theorem 1 does not work. We need to prove Q(Y|f(X))=P(Y|f(X)), but the current version of the proof still can not bridge to P(Y|f(X)). Please give explicit calculation that leads to this bridge.
18. I feel confused about the treatment around Equation (40). In particular, the set script-P of distributions only provide Q_{X|Y,Z}=P_{X|Y,Z}, and here it seems to me that one is using Q_{X,Z|Y}=P_{X,Z|Y}. Please provide more details for this proof.
19. My questions on Theorem 3 and its proof: is theta a random vector or a parameter? Seems neither way fits the proof. For the inequality "a", if all script-L stay close to M, then the log expectation term can not be bounded by O(1/n), is that right?
20. There is a general question I feel confused: is Z observed as data? If yes, then Z goes to learning process and we do not have the OOD problem. In the scenarios that the distribution P_{Z|Y} changes from training to testing, the boundary between X and Z may not be easy to specify, and therefore the CSV penalty is still not easy to define. On the other hand, I feel very difficult to imagine a case where Z is not observable but still needs treatment.

Due to the short review window, I have not checked the proofs of Appendix C.2 and later.

**Summary Of The Paper:**

This paper studies the machine learning problems with OOD data (that is, out-of-distribution data). The paper models OOD data by introducing a new variable called spurious attribute, of which the conditional distribution is different across training and testing data. The paper proposes to use CSV (conditional spurious variation) as regularization term, so that the training process prefers stable output function against the change of spurious attribute distribution. Theoretical analysis is provided for different aspects of the CSV regularization.


=====================


After the discussion session, I believe that most of my concerns and the issues raised in other reviews are resolved. I am convinced that this is a solid paper that merits acceptance.

**Summary Of The Review:**

This paper investigates an important machine learning problem of out-of-distribution data in a theoretical approach. The findings merit a publication. However, there are some mathematical problems -- probably just writing problems. I recommend a revision and would be more than happy to vote for a higher score after the above problem being resolved.

---

> ### Author Response · Authors · 2022-11-14
> **To Reviewer Lqom**
>
> We thank you for your reviews and address your concerns as follows.
>
> Q1: some terms may not be well known. I would suggest adding explanations. This includes: "domain labels”
>
> A1: This notation is from domain generalization in which data are from different environments i.e., domain, the domain label indicates which domain is the data from. We have added this explanation in Section 2.1 in the revised version.
>
> Q2: paragraph 3 on page 3: what is the difference between "input's label" and "class label"?
>
> A2: They are the same, we have unified the notation as a class label.
>
> Q3: I would suggest a comparison of the proposed condition with the condition in this paper: Smale, Steve; Zhou, Ding-Xuan. Online learning with Markov sampling. Anal. Appl. (Singap.) 7 (2009), no. 1, 87--113. For example, I see many works in the literature assuming the invariant of conditional distribution of labels instead. I am curious about the difference between this setting, and the setting used in the current paper.
>
> A3: After checking the suggested paper Smale et al., 2009, we think it studies a different problem. The paper mainly focuses on the online learning problem while the data stream is not i.i.d. Their goal is learning a model that performs well on these non-i.i.d. training data, instead of generalizing on some unseen data which are from distribution in different with training distribution. However, the problem of generalizing on unseen OOD data is mainly considered in this paper.
>
> As for the condition of invariant conditional label distribution $P_{Y\mid X}$, this is used to obviate the label shift. For example, under different environments, the data should have the label i.e., $P_{Y\mid X} = Q_{Y\mid X}$ for different $P$ and $Q$ with represents the probability under different environments.
>
> Q4: Some small typos: "quantify" before Theorem 4 should be "quantifies"
>
> A4: Thank you for pointing out this, we have fixed it.
>
> Q5: Please define the big-O notation in Theorem 4. In particular, what is the limit process?
>
> A5: The big-O notation is defined in Section section 2.2, and in Theorem 4, the limit process is $n\rightarrow \infty$.
>
> Q6: I hope that the author(s) can specify the definition of P, Q, X, Y, and Z, which I believe would greatly help the readers. In particular, I have the following questions directly related to the paper: (a). Is "P" (and "Q") a distribution of (X,Y) or (X,Y,Z)?
>
> A6: The $P$ without a subscript is the distribution of $P_{X, Y, Z}$ and we write it as $P$ for simplification. Thank you for pointing out this, we have made it clear in Section 2.2 in the revised version.
>
> Q7: In general, X and Y are not independent. Are X and Z independent? Are Y and Z independent? Are (X,Y) and Z independent?
>
> A7: $X$ and $Z$ are not independent, $Y$ and $Z$ are not independent, $(X, Y)$ and $Z$ are not independent.
>
> Q8: If Z and Y are not independent, then Z has the prediction power on Y. Would this still fit the definition of "spurious correlation" in this paper?
>
> A8: As claimed in paragraph 3 on page3, the spurious correlation means the feature $Z$ is not casual to the label, but it does not necessary that $Z$ is independent of the label. For example, in Figure 1, the celebrity’s gender (label Y) is not a cause of the celebrity’s hair color (spurious feature $Z$), while the correlation between gender and hair color may vary from training to test data e.g., most males have dark hair in the training set and vice-versa in the test set.
>
> Q9: “In this paper the label set script-Y is discrete (Assumption 1). For the time being, let me assume the label set is just {0, 1}…”
>
> A7: Thank you for pointing out this, in Assumption 2, the loss function $L(f_{\theta}(X), Y)$ should be $R^{|Y|}\times |Y|\rightarrow R^{+}$ with $f_{\theta}(X)$ takes values in $ R^{|Y|}$. Then the discontinuous problem you mentioned is addressed in Section 2.2 by refining the $L(f_{\theta}(X), Y)$. This is a typical setting in a machine-learning community. For example logistic regression the loss on data $(x_{i}, y_{i})$ is $-\log\left(\exp{f_{\theta}(x_{i})(y_{i})} / \sum\limits_{y\in [K_{y}]}exp{f_{\theta}(x_{i})(y)}\right)$, where $\exp{f_{\theta}(x_{i})(y_{i})}$ is the $y_{i}$-th dimension of vector $\exp{f_{\theta}(x_{i})}$.
>
> Q8: Right after Theorem 4, I guess the expression "A decreases with B" is not clear: what is its precise meaning?
>
> A8: It should be stated as CSV is upper bounded by $\hat{CSV}$, it has been corrected in the revised version.

---

> > ### Author Response · Authors · 2022-11-14
> > **To Reviewer Lqom Part II**
> >
> > Q9: The definition (11) is confusing. It is obvious that when u runs through Delta_m, the maximum of u dot F is the maximum coordinate of the vector F. Why not directly use max-coordinate but employing Delta_m and writing (11) in the current complicated way?
> >
> > A9: Yes you are correct. However, the function $\max_{1\leq i\leq K}f_{i}(\theta)$ w.r.t. $\theta$ is usually discontinuous. Thus we should consider its surrogate as we did in this paper i.e., convert it into $u^{\top}F^{k}(\theta)$ for some $u$. This is a classical method to handle such a problem, more details can be referred to Section 5.2.4 in Bubeck et al., 2015.
> >
> > Q10: Page 14, right after (17), how does one derive P_{X|Y}=Q_{X|Y} without assuming P_Z=Q_Z? Also in Example 1, all the numbers 0 should be -1.
> >
> > A10: Right after (17), as in (2), we have verified that the difference among explored distributions only originates from the variation of $P_{Z\mid Y}$ and $P(Y)$ (label shift) which are all unrelated to $P_{X\mid Y}$, thus $ P_{X|Y}=Q_{X|Y}$. The typos in Example 1 are fixed in this paper.
> >
> > Q11: 11.	I feel that the loss function in Example 1 is not designed properly. Indeed, L(1, -1)=3 while L(1, 1) = 4 is even larger
> >
> > A11: It does not matter, we only need $1 = L(-1, -1) \leq L(1, -1) = 3$ and $4=L(1, 1) \leq L(-1, 1)=6$ to make the loss proper to find the right prediction.
> >
> > Q12: Also, the inequality (19) is derived based on the assumption that f(X)=1 almost surely, so the claim that (19) holds for any f is wrong.
> >
> > A13: (19) is a lower bound for any $f$ takes values in $\{-1, 1\}$, so it holds for any $f$, for example for $f=-1$, the loss difference becomes $5TV(P_{Y}, Q_{Y}) \geq TV(P_{Y}, Q_{Y})$.
> >
> > Q14: Proof of Propsition 2: please provide section or page number when citing a book. In this proof however, since script-Y is discrete, one may simply choose A=Script-Y and there is no need to take supremum.
> >
> > A14: Thank you for pointing out this, we have added it. In this proof, we directly adopt the definition of the total variation.
> >
> > Q15: Equation (24) is wrong on the treatment of sup_{A in script-Y}.
> >
> > A15: Thank you for noticing it, actually, there is a typo in the first equality in (24) where the sum should be inside of the absolute value from the definition of total-variance distance. After fixing it, the (24) is right on the treatment of $\sup_{A}$.
> >
> > Q16: Equation (25) is wrong. For example in the last equation, the left-hand side depends on y, while the right-hand side does not.
> >
> > A16: Please note from the definition of $w^*(y) = 1 / (|Y|P_{Y}(Y = y))$, so that $w^{*}(y)P_{Y}(Y = y)$ does not depend on the value of $y$. The (25) is obtained via the explanation right after it, please check it.
> >
> > Q17: Proposition 2 is wrong. Obviously, since Q is in script-P, Q_Y=P_Y, so the only function w that achieves the minimum TV is a constant function w(y)=1, which makes the TV equal to zero.
> >
> > A17: Proposition 2 gives the optimal reweighted label probability distribution under the criteria of minimax total variance distance with a series of distributions $Q$. In fact, the loss will never become zero, the loss is $\sup_{Q}TV(P^w_{Y}, Q_{Y})$ for fixed $P^w_{Y}$ so that we can simply choose another $Q$ in-different with $ P^w_{Y}$ to make the loss larger than zero.
> >
> > Q18: In the proof of Proposition 1, the classifier does not fit the framework proposed in this paper. In particular, the first argument of the loss function script-L is defined on script-Y, a discrete space. However, the classifier here outputs a real number. I guess a reasonable solution is to change the definition of the loss function to accept real first argument (at the top of page 3). This change may help to resolve the issue on Assumption 2.
> >
> > A18: Thank you for pointing this out, we have changed the definition of $L(\cdot, \cdot)$ as a function from $R^{K}\times Y\rightarrow R^{+}$ to fix both the problem in Assumption 2 and Proposition 1. Please check it in Section 2.2 in the revised version.
> >
> > Q19: I think the first part of the proof of Theorem 1 does not work. We need to prove Q(Y|f(X))=P(Y|f(X)), but the current version of the proof still can not bridge to P(Y|f(X)). Please give explicit calculation that leads to this bridge.
> >
> > A19: As we have claimed in equation (2), the difference in distribution $P_{X, Y}$ and $Q_{X, Y}$ only originates from the variety of distribution $P_{Z|Y}$, as we have shown $Q(Y|f(X))$ is independent of $Q_{Z | Y}$, similarly $P_{Y|f(X)}$ is independent of $P_{Z|Y}$. So $P_{Y|f(X)}$ should be invariant under different $P_{Z | Y}$, so that $P_{Y|f(X)} = P_{Y|f(X)}$. To make it more clear, $Q_{Y|f(X)}(y) = Q_{Y, f(X)}(y) / \int_{y}Q_{Y, f(X)}(y, f(X))dy$ whose variety of  $Q$ can only originates from the variety of $Q_{Z|Y}$, while we have proved $Q_{Y|f(X)}(y)$ is invariant over $Q_{Z|Y}$. Thus, it is invariant of $Q$.

---

> > > ### Author Response · Authors · 2022-11-14
> > > **To Reviewer Lqom Part III**
> > >
> > > Q20: I feel confused about the treatment around Equation (40). In particular, the set script-P of distributions only provide Q_{X|Y,Z}=P_{X|Y,Z}, and here it seems to me that one is using Q_{X,Z|Y}=P_{X,Z|Y}. Please provide more details for this proof.
> > >
> > > A20: In the integral term of (40), according to the conditional independence, $\int_{Z}Q_{X, Z|Y}(f(X, z|Y))dz = Q_{X|Y}(f(X) | Y)int_{Z} Q_{Z|Y}(z| Y)dz = Q_{X|Y}(f(X) | Y)$ which is independent of $Q_{Z|Y}$, thus $Q_{X, Y}$ is invariant over $Q$. We have added this part in the revised version.
> > >
> > > Q21: My questions on Theorem 3 and its proof: is theta a random vector or a parameter? Seems neither way fits the proof. For the inequality "a", if all script-L stay close to M, then the log expectation term can not be bounded by O(1/n), is that right?
> > >
> > > A21: $\theta$ here is the parameters obtained on the training set, thus it depends on training data and has randomness. Please notice that we assume $E[L(f_{\theta}(\tilde{x_{i}}), y_{i})] = 0$, without loss of generality, so that $L(f_{\theta}(\tilde{x_{i}}), y_{i})$ will not stay close to zero. This is a classical technique to derive informatic generalization bound. Please check Xu et al., 2017 for details.
> > >
> > > Q22: There is a general question I feel confused: is Z observed as data? If yes, then Z goes to learning process and we do not have the OOD problem. In the scenarios that the distribution P_{Z|Y} changes from training to testing, the boundary between X and Z may not be easy to specify, and therefore the CSV penalty is still not easy to define.
> > >
> > > A22: Usually $Z$ can not be observed vectorized data. For example, in Figure 1, the celebrity’s hair color is a spurious attribute $Z$. Usually, we do not have the pixel value of hair color, but we may have its label e.g., dark hair during training. In this paper, we mainly treat the spurious attributes as label, as claimed right after Assumption 2. But our results before section 4.2 can be generalized to $Z$ is not a label, so we do not impose this Assumption before section 4.2.
> > >
> > > Q23: On the other hand, I feel very difficult to imagine a case where Z is not observable but still needs treatment.
> > >
> > > A23: Let us back to Figure 1, to classify a celebrity’s gender with a celebrity’s hair color as a spurious attribute. The label of hair color may be unobservable (uncollected) in the training set. However, overfitting the correlation shift in the training set may deteriorate the performance of the model on OOD test data. For example, if most males have dark hair in the training set, and the model overfits such spurious correlation, that means dark hair becomes a feature to predict the celebrity is a male. Then dark hair may become an interference feature if the most male in the test data have blonde hair.
> > >
> > > Ref:
> > >
> > > Bubeck 2015, Convex Optimization.
> > >
> > > Xu et al., 2017, Information-theoretic analysis of generalization capability of learning algorithms.

---

> > > > ### Comment · Reviewer_Lqom · 2022-11-17
> > > > **I appreciate the detailed responses**
> > > >
> > > > I appreciate the careful response on its coverage of all the points I raised.
> > > >
> > > > I quote the "Q*/A*" notation from the author response. These indexes may be different from the original review.
> > > >
> > > > A9. I do not think continuity would be a problem. First, as specified below (11), "each dimension of F^k(θ) is Lipschitz continuous function with Lipschitz gradient," so their maximum would be Lipschitz continuous. Second, transforming the maximum to a convex combination could be done later before describing the algorithm if the algorithm needs C1 smoothness. But I think this is a minor issue and only suggest its fix in the opportunity when other changes are planned.
> > > >
> > > > A17. I would invite more comments on Proposition 2. Let's look at Sup_{Q in script-P}TV(P^w_Y, QY). Remember that whenever Q is in script-P, by definition, Q_Y=P_Y. Therefore the supremum in the curl brackets in (23) is trivially equal to TV(P^w_Y, P_Y). When one further takes the minimum outside the curl brackets, the minimum is also trivial. The definition of script-P in "Definition 1" is changed in the revised version, but I believe the change is only notational and does not alter my concern.
> > > >
> > > > A19. I still believe that an explicit calculation is necessary. At least this does not hurt.
> > > >
> > > > A21. Is there any short explanation (or cited long theorem) for the claim that L is sub-Gaussian?

---

> > > > > ### Author Response · Authors · 2022-11-18
> > > > > **Response to comments**
> > > > >
> > > > > Thank you for your response. I will reply to your questions in the order as follows:
> > > > >
> > > > > 1. The max of a class of Lipschitz-smooth functions is not necessary Lipschitz-smooth. For example consider the function $f(x) = |x| = \max\\{-x, x\\}$, its gradient is discontinuous on the point $x=0$. So we still need the technique of surrogating in this paper. More details about the discontinuity can be referred to Nouiehed et al., 2019.
> > > > >
> > > > > 2. Thank you for pointing out this. In Appendix A, as clarified at the beginning of this section, we discuss the OOD generalization with the label shift, i.e., $P_Y \neq Q_{Y}$. Thus the condition $P_{Y} = Q_{Y}$ in Appendix A (also includes Proposition 2) does not hold anymore. Thus in (23), $Q_{Y}$ is not necessary equal to $P_{Y}$.
> > > > >
> > > > > 3. Thank you for your suggestion, we will add these details to the revised version.
> > > > >
> > > > > 4. As the loss function $L$ is bounded, thus it is sub-Gaussian. Details can be checked in Example 3.6 in Duchi 2019.
> > > > >
> > > > > Ref:
> > > > >
> > > > > Nouiehed et al., 2019. Solving a Class of Non-Convex Min-Max Games Using Iterative First Order Methods.
> > > > >
> > > > > Duchi 2019. Lecture Notes for Statistics 311/Electrical Engineering 377

---

### Official Review · Reviewer_fPjQ · 2022-10-23

**Confidence:** 4
**Correctness:** 3
**Technical Novelty And Significance:** 3
**Empirical Novelty And Significance:** 3
**Recommendation:** 6

**Clarity, Quality, Novelty And Reproducibility:**

clarity:
overall the paper is well-written and easy to understand

Quality:
While the proposal in observable case is straight-forward (why not use direct ways of measuring conditional independence than via (3)?), the proposal in more pratical case of unoservable suprious correlation seems restrictive making the contribution less strong.

Minor:
Also I prefer avoiding associating spurious features with non-causal features. because one can have anti-causal learning that is not spurious.



**Strength And Weaknesses:**

Strength:
1. Overall, the paper is well-written with nice organization.

Weakness:
1. though the motivation, various justifications make sense, I feel the final proposal i..e, (10) is over-restrictive. For e.g., why not perform sparse feature selection etc. to eliminate the spurious features. For resultant predictions, clearly (10) is restrictive, while they will alleviate the issue of spurious correlations?
2. Even in simulations I think simple baselines that perform feature selection are very important for understanding the usefulness of the proposal. Also, since a plethora of diverse methodologies for feature selection exist, this may make the baseline very competitive.


**Summary Of The Paper:**

The paper considers the case of learning under presence of spurious attributes, which are correlate with labels, and make the train and test distributions different. It begins with a criteria, (3), for measuring independence of prediction function and the spurious attributes. Then replaces with an upper bound as the identity of such spurious will not be known (10). Simulations show improvement wrt. existing OOD baselines.

**Summary Of The Review:**

In view of the issues desribed above like over-restirctedness of (10) and missing comparisons with feature selection based methods, I tend to recommend a reject.
]

____
After the rebuttal and discussion, my issues are resolved.

---

> ### Author Response · Authors · 2022-11-14
> **To Reviewer fPjQ**
>
> We thank you for your reviews and address your concerns as follows.
>
> Q1: “I feel the final proposal i..e, (10) is over-restrictive. For e.g., why not perform sparse feature selection, etc. to eliminate the spurious features.”
>
> A1: We are not sure if the sparse feature selection here means the technique such as LASSO or sure independent screening. If so, to the best of our knowledge, there is no existing literature working on eliminating spurious features via variable selection. We think this is because sparse feature selection builds upon the assumption that the ground model is sparse, while in OOD generalization
> tasks such as image classification problems, useful features, and spurious features can both be high dimensional, as in the example in Proposition 1 with both $d_{1}$ and $d_{2}$ very large. Thus, sparse feature selection may not find the ground truth model in this case.
>
> Besides, for many OOD problems, designing the feature selection criteria is a problem, as spurious attributes may be more informative than the useful feature in the training data when fitting the class label. For example, in the example in Proposition 1, when $\sigma_{YZ}(P)\rightarrow 1$, the sparse feature selection method will not necessarily select feature in the first $d_{1}$-dimensions (which are useful features), as both spurious features and useful features are informative to predict the label $Y$ in the training set.
>
> Q2: For resultant predictions, clearly (10) is restrictive, while they will alleviate the issue of spurious correlations?
>
> A2: Our $CSV_{U}$ imposes the constraint that the model should have uniform performance on data with the same class label. It is not contradicted the ground truth model provided the ground truth model can classify the data perfectly. In this case, the ground truth model tends to have a uniformly good performance on data with the same class label and we add the term $CSV_{U}$ to encourage the trained model to share such a property. Our regularizer is not restrictive as the objective model satisfies the such constraint.
>
> Q3: Even in simulations I think simple baselines that perform feature selection are very important for understanding the usefulness of the proposal. Also, since a plethora of diverse methodologies for feature selection exists, this may make the baseline very competitive.
>
> A3: In the simulation i.e., toy example, we use the linear models, and we try to select the useful features i.e., the first 5 dimensions, thus all methods can be viewed as feature selection methods in this regime. The cosine similarity in table 5 indicates how well the selected features are close to the useful ones. To further address your concern, we consider two classical feature selection methods, i.e., regularizing training with $l_{1}$ (LASSO) and $l_{2}$ regularizers (ridge regression), the results are summarized in Table 4 in the revised version. Our method beats the two feature selection methods.
>
> We also add $l_{1}$ regularizer in the experiments of CelebA, Waterbirds, MNLI, CivilComments in Section 6 of the revised version, the results show that our methods beat these baselines as well.
>
> Q4: “While the proposal in observable case is straight-forward (why not use direct ways of measuring conditional independence than via (3)?)”
>
> A4: As claimed in Section 4.1, the proposed CSV (3) can be computed via training distribution so that we can regularize training with it to improve OOD generalization. More than that, the proposed CSV can directly upper bound the OOD generalization error as in Theorem 2.
>
> Q5: The proposal in more pratical case of unoservable suprious correlation seems restrictive making the contribution less strong.
>
> A5: The proposed $CSV_{U}$ is used to estimate CSV in absence of observed spurious attributes, which can be computed without any other information. When we have observed spurious attributes, we can estimate CSV via (7) as we claimed in the paper.
>
> Q6: Also I prefer avoiding associating spurious features with non-causal features. because one can have anti-causal learning that is not spurious.
>
> A6: Thank you for pointing out this. The expression that $Z$ is not causal to predict $Y$ does not mean we aim to find the causal to predict $Y$. We only chase for the stable predictors which are not necessarily causal.
>
> Q7: In view of the issues desribed above like over-restirctedness of (10) and missing comparisons with feature selection based methods, I tend to recommend a reject.
>
> A7: The restrictedness of (10) has been addressed in A5, and the comparison with feature selection methods has been discussed in A3.

---

> > ### Comment · Reviewer_fPjQ · 2022-11-18
> > **Over-restriction point**
> >
> > Thanks for your response. I think my point about (10) was misunderstood.
> >
> > What I was try to say is, suppose there are two very different samples x_1, x_2, both in the same class and have the same values for the spurious attributes. (10) will unnecessarily enforce their function values being same. This seems very over-restrictive to me. Basically enforcing invariance in ALL cases rather than where needed.
> >
> > This, in conjunction with the fact commented by other reviewers that so many similar ideas are published, makes me comfortable not changing my original decision.

---

> > > ### Author Response · Authors · 2022-11-18
> > > **Reply to over-restriction point**
> > >
> > > Thanks for your response. In fact, for the different samples $x_1, x_2$, both in the same class and have the same values for the spurious attributes, the objective (10) will not make them output the same value. However, our training objective is (11), where the objective $R_{emp}(f_{\theta}, P)$ (classification loss) will make model output the same value on such $x_{1}$ and $x_{2}$. The (10) in (11) is a regularizer to improve OOD generalization.
> > >
> > > The difference between the existing literature has been discussed in our responses to reviewers and the revised version.

---

### Official Review · Reviewer_bzT6 · 2022-10-26

**Confidence:** 4
**Correctness:** 3
**Technical Novelty And Significance:** 3
**Empirical Novelty And Significance:** 3
**Recommendation:** 8

**Clarity, Quality, Novelty And Reproducibility:**

The paper is mostly clear and high quality, although a number of theorem statements and proofs have issues. The main issue here is novelty, as the problem formulation has been given before. Some aspects of the paper are nice contributions, but probably require major reframing. Experiments and algorithms are described in nice detail, especially in appendices.

**Strength And Weaknesses:**

Post-Rebuttal:

I am raising my score to accept, but with some additional suggestions for the authors.

 + The authors did a better job of connecting their problem formulation to previous work in the revisions. I would encourage the authors to also note that the content of the analysis in Makar et al 2021 is very similar to the analytical approach of this paper, including the way that the OOD generalization bound is constructed. I do think that the approach in this paper greatly streamlines those arguments, and relaxes some assumptions.
 - The proof of Theorem 1 still seems unnecessarily complicated, especially given my comment before that (39) is vacuous since the LHS and RHS are symbolically equivalent. Note that if you show $P(f(X), Y) = Q(f(X), Y)$, this implies that $P(Y | f(X)) = Q(Y | f(X))$, so the first half of the proof is unnecessary given the second half.
 - Given the presentation in the rest of the paper, Theorem 3 is still extremely difficult to motivate and parse parse. It is not clear what "conditional independence does not contradict in-distribution generalization" means; is the goal to show that the generalization bound goes to zero as n goes to infinity? Something else? It is also not clear what, say $f_\theta \perp S_Z \mid S_Y$ means, given that $f_\theta$ is trained using $S_Z$. The framework for producing this generalization bound is also not introduced at all. Some framing would be useful here, as simply giving references is not sufficient. However, this theorem does not seem to be particularly crucial to the overall story of the paper. Since I would find the rest of the paper to be rather strong even if this section were deleted, I won't lower my score because of this, but if the authors want to continue to include this result, I would suggest heavy edits.

==========


Strengths:
 + The paper does address an important spurious correlation problem.
 + Some of the risk invariance arguments (once fixed up) could yield more streamlined versions of the arguments made in Makar et al 2021.
 + The CSV penalty seems potentially easier to implement (and has no tuning parameters) compared to the MMD penalties proposed in previous work, although the parallel mini-batching strategy with differing weights could potentially be difficult to implement in standard workflows.
 + Empirical performance on standard benchmarks is compelling.
 + The CSV_U regularizer when no spurious variables are visible is a major contribution.

Weaknesses:
 - The paper's development is extremely similar to [Makar et al 2021, AISTATS](https://proceedings.mlr.press/v151/makar22a.html), which is not cited. There is also additional concurrent work building on Makar et al that makes a more explicit connection to conditional independence regularization [Makar and D'Amour 2022](https://arxiv.org/abs/2209.09423). It would be absolutely necessary to contrast against Makar et al 2021, and show how developments in the current paper go beyond what was shown in Makar et al. For example, the correlation shift is identical, as is the goal to find a predictor with invariant risk, and there is also a similar argument about the generalization gap. There are some aspects of this paper that are new, e.g., the setting is slightly more general in that the risk invariance result does not require the existence of a sufficient statistic, and the CSV penalty here differs from the weighted MMD penalty suggested in Makar et al 2021. **IMO, the paper needs some major reframing in light of these similarities.**
 - Similarly, there needs to be more discussion of the relationship between the CSV penalty and the conditional MMD penalty given here and the conditional MMD penalty suggested in Veitch et al 2021 (cited in the paper).
 - The proof of Theorem 1 does not really make sense. For example, equation (39) is vacuous as written (the final expression is symbolically equivalent to the LHS). There is an obvious way to improve this. You want to show $P(f(X), Y) = Q(f(X), Y)$ under the given assumptions. This follows from $Q(f(X), Y) = P(Y) \int_{\mathcal Z} P(f(X) \mid Y, Z=z) dQ(z | Y) = P(Y)P(f(X) \mid Y) \int_{\mathcal Z} dQ(z | Y)$, where the first equality follows from the correlation shift assumption, and the second equality follows from the $f(X) \perp Z \mid Y$ assumption.
 - Lemma 1 is incorrect. Consider the counter-example: $U = W$, $V = -W$. Then $I(W; U+V) = I(W;0) = 0$ but $I(W; U) = H(W) \geq 0$ (and $I(W; V | U) = 0$). The change of variables from U+V to (U,V) is done incorrectly in the proof; the domain of integration would need to change.
 - I have difficulty parsing Theorem 3. It is not clear what the mutual information term involving $\theta$ means. The proof references a distribution over $\theta$, which does not make sense to me. Similarly, Theorem 3 relies on the incorrect Lemma 1. Also, $M$ is not defined.
 - The prose around Theorem 3 seems to argue that there is no tradeoff between robustness and in-distribution prediction performance, which is not the case in general. In fact, at best, the theorem would indicate that the generalization **gap** between the empirical risk and population risk is small, but says nothing about whether the minimized population risk subject to the conditional independence constraint is small (which is what I might interpret as "generalization capability").

**Summary Of The Paper:**

The authors propose an approach to training models that are robust to correlation shifts, where the correlation between a label and a spurious attribute changes between training and test distributions, but the generative model given label and spurious attribute and the label distribution does not change. The authors argue that under such correlation shifts, constraining the predictor to be conditionally independent of the spurious attribute given the class label yields a risk invariant predictor. They propose a regularizers that upper-bound the risk difference between distributions. They argue that independence constraints yield a favorable generalization gap. They show that such an objective can be optimized at a favorable rate.

**Summary Of The Review:**

Post-Rebuttal: The authors addressed my concerns sufficiently. See comments above in "strengths and weaknesses".

===============

I think the paper needs a major rewrite in light of prior work. Many of the technical results constitute a nice contribution, but the paper would need to be reframed to highlight them.

---

> ### Author Response · Authors · 2022-11-14
> **To Reviewer bzT6**
>
> We thank you for your reviews and address your concerns as follows.
>
> General Response: Thank you for pointing out many important references, and some mathematical issues. We have made revisions to address your concerns in the revised version. The revision can be summarized as follows:
>
> 1)    We add the comparison with Makar et al., 2021, Makar et al., 2022, and Veitch et al., 2021. Please see A1, A2, A3 for details.
>
> 2)    We have corrected Lemma 1 and Theorem 3. The revised Theorem 3 does not change the result we want to convey. Please see A5, A6, A7 for details.
>
> Q1: The comparison with Makar et al., 2021, AISTATS
>
> A1: Thank you for pointing out this important reference, we have cited this paper, and added the comparison in the revised version in Section 2.2. After checking this paper, we summarize the difference as follows:
>
> 1)	Their paper relies on a causal DAG and assumes the existence of a sufficient statistic $X^{*}$ such that Y only affects $X$ through $X^{*}$, and it can be recovered via $X$. But we do not need such causal DAG and such sufficient a statistic.
>
> 2)	They characterize the correlation shift as by imposing the invariance on $P(X\mid X^{*}, Z), P(X^{*}\mid Y)$, and $P(Y)$, while we only impose invariance on $P(X\mid Y, Z)$, and $P(Y)$.
>
> 3)	Their method is built upon reweighting strategy i.e., change of measure, to get the risk under unconfounded distribution. Besides that, the regularizer they used is to get unconfounded distribution. However, we do not aim to get such risk by change of measure, we directly regularize the training with the conditional invariance of model over spurious attributes.
>
> 4)	Our method can be implemented in absence of spurious attributes, while their can not.
>
> Q2: The comparison with Makar et al., 2022.
>
> A2: Thank you for pointing out this important reference, we have cited this paper, and added the comparison in the revised version after Definition 2. After checking this paper, we summarize the difference as follows:
>
> 1)	As in Makar et al., 2021, their results rely on a causal DAG, and the existence of a sufficient statistic.
>
> 2)	Though they prove that the similar result that conditional independence breaks the correlation shift as we do, their regularizer in (3) is only applied to spurious attributes with two classes, while we have no such constraint.
>
> 3)	The regularizer they proposed, as they said, “has some practical limitations, especially when training using mini-batches of data in stochastic gradient descent”. Thus, the training objective they used is similar to the one in Makar et al., 2021. However, our regularizer is independent of the number of spurious attributes, and we develop the algorithm with a provable convergence rate to solve the regularized mini-max optimization problem.
>
> 4)	Our method can be implemented in absence of spurious attributes, while theirs can not.
>
> Q3: Similarly, there needs to be more discussion of the relationship between the CSV penalty and the conditional MMD penalty given here and the conditional MMD penalty suggested in Veitch et al 2021 (cited in the paper).
>
> A3: As we have clarified in A1 and A2, the regularizers in Makar et al., 2021, and Makar et al., 2022 are used to get the probability weights under unconfounded distribution. In Veitch et al., 2021, there are two regularizers. For the marginal regularization, it measures the invariance of $f(X)\mid Z$ over $Z$, which has been discussed in the last paragraph on page 4. The other regularizer is conditional regularization, which measures the conditional invariance of $f(X)\mid Y, Z)$ over $Z$. The invariance is similar to the one our CSV regularizer aims to capture. However, compared with their regularizer, our improvements are three-fold 1): their probability measure only applies to spurious attributes with two classes. 2): Their metric is defined on a Reproducing Hilbert Kernel Space, which depends on the choice of reproducing kernel, thus affecting the performance of the model. 3): In contrast to theirs, our regularized training objective has a provable convergence rate.
>
> Thank you for pointing out this, we have added these comparisons in the revised version after Definition 2.
>
> Q4: The proof of Theorem 1 does not really make sense. For example, equation (39) is vacuous as written (the final expression is symbolically equivalent to the LHS). There is an obvious way to improve this. You want to show P(f(X),Y)=Q(f(X),Y) under the given assumptions.
>
> A4: The equation (39) is to prove our first conclusion that $Y\mid f(X)$ (which decides the prediction error) is invariant over distributions. We prove $P(f(X),Y)=Q(f(X),Y)$ as you said in equation (40).

---

> > ### Author Response · Authors · 2022-11-14
> > **To Reviewer bzT6 Part II**
> >
> > Q5: Lemma 1 is incorrect. Consider the counter-example: U=W, V=−W. Then I(W;U+V)=I(W;0)=0 but I(W;U)=H(W)≥0 (and I(W;V|U)=0). The change of variables from U+V to (U,V) is done incorrectly in the proof; the domain of integration would need to change.
> >
> > A5: Thank you for pointing out this, lemma is used to derive the equation (50) in Theorem 3. We notice this mistake and refined Lemma 1 and Theorem 3. The revised Theorem 3 does not change the result we want to convey, that is “the conditional independence is not in contradiction with in-distribution generalization”. Please check it in Theorem 3 in the revised version.
> >
> > Q6: I have difficulty parsing Theorem 3. It is not clear what the mutual information term involving θ means. The proof references a distribution over θ, which does not make sense to me. Similarly, Theorem 3 relies on the incorrect Lemma 1. Also, M is not defined.
> >
> > A6: Please check the revised Theorem 3 in the revised version. The $M$ is the upper bound of the loss function defined in Section 2.2
> >
> > Q7: The prose around Theorem 3 seems to argue that there is no tradeoff between robustness and in-distribution prediction performance, which is not the case in general In fact, at best, the theorem would indicate that the generalization gap between the empirical risk and population risk is small, but says nothing about whether the minimized population risk subject to the conditional independence constraint is small (which is what I might interpret as "generalization capability").
> >
> > A7: Theorem 3 does not mean there is no tradeoff between robustness and in-distribution performance. It provides an improved in-distributional generalization bound with the imposed conditional independence constraint. But the improved bound does not necessarily mean the conditional independence model has better in-distribution generalization error than the one obtained by ERM. However, it states that at least, capturing conditional independence will not result in an extremely large in-distribution generalization error. Plugging this observation into Theorem 2 indicates that the conditional independence model has guaranteed OOD generalization error.
> >
> > Reference:
> >
> > Veitch et al., 2021, Counterfactual Invariance to Spurious Correlations: Why and How to Pass Stress Tests.
> >
> > Makar et al., 2021, Causally Motivated Shortcut Removal Using Auxiliary Labels
> >
> > Makar et al., 2022, FAIRNESS AND ROBUSTNESS IN ANTI-CAUSAL PREDICTION

---

> ### Author Response · Authors · 2022-12-07
> **Thank you for your suggestion.**
>
> Thank you for your suggestion about the proof of Theorem 1 and the presentation of Theorem 3! We will simplify the proof of Theorem 1 and clearly formulate Theorem 3 according to your suggestion.

---

### Official Review · Reviewer_M8jr · 2022-10-27

**Confidence:** 4
**Correctness:** 4
**Technical Novelty And Significance:** 3
**Empirical Novelty And Significance:** 3
**Recommendation:** 6

**Clarity, Quality, Novelty And Reproducibility:**

Some of the writing in the paper could be improved. This work has ideas very close to existing ideas and the differences are not adequately discussed.

**Strength And Weaknesses:**

The paper studies an important problem which has gained widespread interest in recent years. I have a few concerns which I mention below.

The invariance assumption of the model being conditionally independent of the spurious attribute given the label, how is it different from Conditional domain adaptation [1] which matches the feature distribution given the labels and the domain. It seems like the same assumption is used here.

This work lacks important citations and comparisons. They state that they can even estimate their CSV without ground truth group values but there already exists approaches which work without the ground truth labels [2,3,4] but many of these works have either not been cited or compared with . It would be good to have these comparisons included and also say what is different about their approach.

The empirical results are marginal. This algorithm improves the accuracy by a small amount on worst groups but reduces the average weighted performance on all the datasets. Hence, it is not clear if they are actually getting rid of the spurious feature or just trading some of the majority group accuracy with the minority group.

It is also not clear how different this approach is from the group distributionally robust approaches which also try to reduce the gap between losses across different groups.

I am also wondering whether the authors needed to regularize their models as well to make this method work because the authors in [5] claim that if there is no regularization, these overparameterized methods fit well on all the groups in the training set and hence, there is no effect of distributionally robust optimization.

[1] Conditional Adversarial Domain Adaptation
[2] No Subclass Left Behind: Fine-Grained Robustness in Coarse-Grained Classification Problems
[3] Just Train Twice: Improving Group Robustness without Training Group Information
[4] Environment Inference for Invariant Learning
[5] DISTRIBUTIONALLY ROBUST NEURAL NETWORKS FOR GROUP SHIFTS: ON THE IMPORTANCE OF REGULARIZATION FOR WORST-CASE GENERALIZATION

**Summary Of The Paper:**

This work studies the problem of learning with spurious correlations whose correlation with labels can change between the training and test distributions. They propose to regularize the algorithm with a new regularizer called Conditional Spurious Variation (CSV) which essentially measures the difference in loss values over different groups. They show that the Out of distribution (OOD) error can be upper bounded by the in distribution error + CSV term. They also show improved performance on many benchmark datasets.


**Summary Of The Review:**


The paper is missing important comparisons and citations. The empirical performance is also incremental.

****************

I have read the authors' response and they have sufficiently addressed all my concerns regarding comparisons with prior work and empirical contributions. I have also changed my recommendation accordingly.

---

> ### Author Response · Authors · 2022-11-14
> **To Reviewer M8jr**
>
> We thank you for your reviews and address your concerns as follows.
> Q1: The invariance assumption of the model being conditionally independent of the spurious attribute given the label, how is it different from Conditional domain adaptation [1] which matches the feature distribution given the labels and the domain. It seems like the same assumption is used here.
> A1: Thank you for pointing out the reference [1], we have added this reference in the revised version. After checking this paper, we find their main idea is creating feature representation $f$ and classifier prediction $g$ that are uniformly unidentifiable under an adversarial trained domain discriminator. In this regime, the $f$ and $g$ together can be viewed as the output of model $f_{\theta}(X)$ in our paper. Thus they aim to capture a model that is invariant over the domain i.e., invariance of $f_{\theta}(X)\mid Z$ over $Z$. We have discussed such invariance in the last paragraph on page 4, which states that such invariance can not handle correlation shift in this paper.
>
> Q2: This work lacks important citations and comparisons. They state that they can even estimate their CSV without ground truth group values but there already exists approaches which work without the ground truth labels [2,3,4] but many of these works have either not been cited or compared with. It would be good to have these comparisons included and also say what is different about their approach.
>
> A2: Thank you for pointing out these references. Before clarifying the difference between our paper and these references, we should point out that we have compared with the recent SOTA methods [6, 7] to mitigate correlation shift in absence of observed spurious attributes, and our method consistently improves their methods. In a word, the differences are, 1): the spirit of [2,4] is predicting the label of spurious attributes which we do not need such a classifier, 2): [3] uses an extra validation set that has observed spurious attributes. The detailed differences are summarized as follows:
>
> Compared with [2], they first train a label classifier, then labeled the spurious attribute by the clustering algorithm, and finally apply the GroupDRO [5] under each group consisting of the data with the same predicted spurious attribute and class label. Their method clearly depends on the correctness of the predicted spurious attributes. While obtaining a high-quality spurious attributes classifier is hard in practice. Compared with their method, our RCSVU measures the intra-variation of the model over the same class and does not need to predict spurious attributes as they do. Empirically, our RCSVU beats their method in terms of worst-case group accuracy in both CelebA and Waterbirds, i.e., 76.9 v.s. 52.4 and 81.2 v.s 76.2. We have added this comparison in the revised version in Section 6 and Section 2.1.
>
> Compared with [3], we have clarified in the last paragraph of page 2, their method uses an annotation validation set with observable spurious attributes during training. Thus, comparing our method with theirs is unfair.
>
> Compared with [4], the spirit of their method is similar to [3], that is, learning a soft-spurious attribute classifier to predict the spurious attributes. Empirically, our RCSVU beats their method in terms of worst-case group accuracy in Waterbirds and CivilComments i.e., 81.2 v.s. 78.7 and 68.7 v.s. 67.0. We have added this comparison in the revised version in Section 6 and Section 2.1.
>
> Q3: The empirical results are marginal. This algorithm improves the accuracy by a small amount on worst groups but reduces the average weighted performance on all the datasets. Hence, it is not clear if they are actually getting rid of the spurious feature or just trading some of the majority group accuracy with the minority group.
>
> A3: To show our improvements over the worst-case group is not “trading some of the majority group accuracies with the minority group”, we report the averaged accuracies over groups, and our method consistently improves the baseline methods on all experiments. As for “average weighted performance”, that is “total” accuracy on the test data, our method beat other methods on Waterbirds and MultiNLI instead of “reduces the average weighted performance on all the datasets.”. Compared with total accuracy on the test set, we think the averaged accuracies over groups is a better criterion to check the performance of the model, as the latter does not depends on the number of data in each group. Finally, as we have clarified in the last paragraph in Section 6, all methods capture some spurious correlation, while our method can mitigate such overfitting.

---

> > ### Author Response · Authors · 2022-11-14
> > **To Reviewer M8jr Part II**
> >
> > Q4: It is also not clear how different this approach is from the group distributionally robust approaches which also try to reduce the gap between losses across different groups.
> >
> > A4: We have made a comparison with GroupDRO in the second paragraph on page 7, please check it. In a word, we split the goal of “perfect in-distribution test accuracy” and “robustness over different spurious attributes”, which makes our training process more stable.
> >
> > Q5: I am also wondering whether the authors needed to regularize their models as well to make this method work because the authors in [5] claim that if there is no regularization, these overparameterized methods fit well on all the groups in the training set and hence, there is no effect of distributionally robust optimization.
> >
> > A5: In [5], the authors use a large weight-decay regularizer to create a stable training process that avoids overfitting the spurious correlation. However, as we use an explicit regularizer to avoid overfitting, we do not need such a large weight-decay regularizer and has a stable training process, see the second paragraph on page 7 for more details.
> >
> > Reference: [1] Conditional Adversarial Domain Adaptation
> >
> > [2] No Subclass Left Behind: Fine-Grained Robustness in Coarse-Grained Classification Problems
> >
> > [3] Just Train Twice: Improving Group Robustness without Training Group Information
> >
> > [4] Environment Inference for Invariant Learning
> >
> > [5] DISTRIBUTIONALLY ROBUST NEURAL NETWORKS FOR GROUP SHIFTS: ON THE IMPORTANCE OF REGULARIZATION FOR WORST-CASE GENERALIZATION
> >
> > [6] Predicting with high correlation features.
> >
> > [7] Simple data balancing achieves competitive worst-group-accuracy.

---

### Author Response · Authors · 2022-11-14
**General Response**

We thank all reviewers for their valuable comments. We have revised the paper, according to their comments. All revised parts are marked as blue. The main revisions are summarized as follows:

1)	We have added the comparisons with some existing literature pointed out by Reviewers. The comparisons were conducted both theoretically and empirically.

2)	We have corrected some mathematical issues pointed out by reviewers.

The details of revisions are referred to the following official comments.

---

### Decision · Program_Chairs · 2023-01-20

**Decision:**

Accept: poster

**Justification For Why Not Higher Score:**

The idea is not totally novel and coincides with some of the existing work.

**Justification For Why Not Lower Score:**

There is a consensus among the expert reviewers.

**Metareview: Summary, Strengths And Weaknesses:**

This paper investigates the problem of learning in the OOD setting in which certain spurious correlations between inputs and labels can change between training and test data. To encourage the predictor to learn the stable, i.e., non-spurious features, the paper proposes a new regularizer called Conditional Spurious Variation (CSV) which constrains the predictor to be conditionally independent of the spurious attribute given the class label. Since the CSV requires knowledge of the spurious features in the data, which is not the case in practice, the paper thus considers an upper bound of such quantity instead. The paper provides competitive empirical results compared to existing methods.

While the proposed idea is natural, intuitive, and coincides with existing work in the literature as pointed out by Reviewer bzT6, there is a consensus among the expert reviewers that this paper provides substantial contributions such that it deserves acceptance. After looking at the paper, reviews, and discussions between reviewers and authors, I mostly agree with the reviewers, but would also highly encourage the authors to incorporate some of the suggestions made by the reviewers into the camera-ready version.

**Note From Pc:**

if the above contains the word "oral" or "spotlight" please see: "oral" presentation means -> notable-top-5% and "spotlight" means -> notable-top-25%. As stated in our emails, we are disassociating presentation type from AC recommendations

**Summary Of Ac-Reviewer Meeting:**

No meeting has been conducted.